# Understanding Adam Requires Better Rotation Dependent Assumptions

**Tianyue H. Zhang**[1,2,†]  **Lucas Maes**[1,2,†]  **Alan Milligan**[3]

**Alexia Jolicoeur-Martineau**[4]  **Ioannis Mitliagkas**[1,2,5,6]  **Damien Scieur**[2,4]

**Simon Lacoste-Julien**[1,2,4,5]  **Charles Guille-Escuret**[1,2]

[1] Mila, Quebec AI Institute  [2] Université de Montréal
[3] University of British Columbia  [4] Samsung SAIL Montreal
[5] Canada CIFAR AI Chair  [6] Archimedes Unit, Athena Research Center

## Abstract

Despite its widespread adoption, Adam's advantage over Stochastic Gradient Descent (SGD) lacks a comprehensive theoretical explanation. This paper investigates Adam's sensitivity to rotations of the parameter space. We observe that Adam's performance in training transformers degrades under random rotations of the parameter space, indicating a crucial sensitivity to the choice of basis in practice. This reveals that conventional rotation-invariant assumptions are insufficient to capture Adam's advantages theoretically. To better understand the rotation-dependent properties that benefit Adam, we also identify structured rotations that preserve or even enhance its empirical performance. We then examine the rotation-dependent assumptions in the literature and find that they fall short in explaining Adam's behaviour across various rotation types. In contrast, we verify the orthogonality of the update as a promising indicator of Adam's basis sensitivity, suggesting it may be the key quantity for developing rotation-dependent theoretical frameworks that better explain its empirical success.

## 1  Introduction

Large Language Models (LLMs) have demonstrated remarkable capabilities as their scale grows [Brown et al., 2020, Kaplan et al., 2020]. However, this unprecedented growth in model scale has led to a proportional increase in the economic [Dong and Xie, 2024, Sharir et al., 2020, Varoquaux et al., 2024] and environmental [Luccioni et al., 2023, 2019] costs associated with their training. Despite this clear motivation, Adaptive Moment Estimation (Adam) [Kingma and Ba, 2015] has persisted as the go-to optimizer especially for language models, with only minor modifications such as AdamW [Loshchilov and Hutter, 2019] becoming widely adopted since Adam's inception. This success has prompted extensive research to provide theoretical justification for Adam's performance. While the original convergence proof for Adam was later found to be flawed [Rubio, 2017], recent studies have proposed rigorous convergence proofs under plausible assumptions [Li et al., 2023b, Chen et al., 2019, Défossez et al., 2022].

However, these proofs do not elucidate Adam's advantages over SGD when training transformer models [Vaswani et al., 2017]. Numerous works attempted to explain Adam's superiority, employing

---

[†] Equal Contribution, correspondence to: `tianyue.zhang@mila.quebec`

39th Conference on Neural Information Processing Systems (NeurIPS 2025).

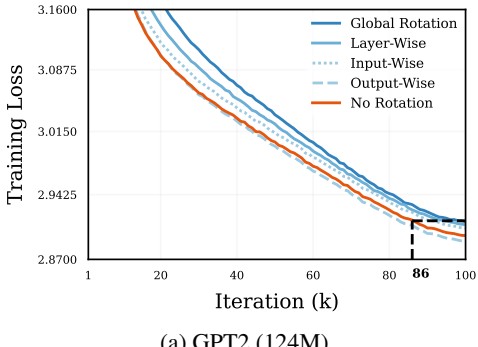 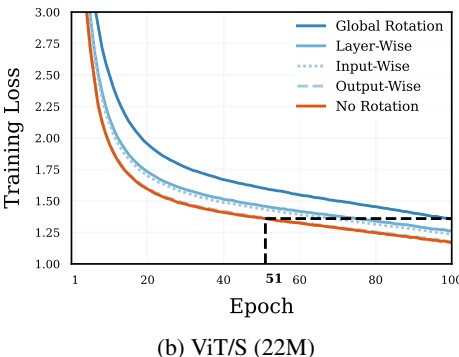

(a) GPT2 (124M)          (b) ViT/S (22M)

Figure 1: **Adam's performance degrades under certain random rotations of the parameter space, demonstrating its dependence on the standard basis.** (a) For GPT2, global rotations lead to a $16\%$ slowdown in training. (b) ViT experiences a more dramatic $96\%$ slowdown under global rotations. Performance is preserved under output-wise rotations but progressively worsens with input-wise, layer-wise, and global rotations, revealing Adam's increasing sensitivity to coordinate changes of broader scopes. Experimental details are provided in Section 3.1.

diverse assumptions and analytical frameworks [Zhou et al., 2024, Pan and Li, 2022, Zhang et al., 2024, Kunstner et al., 2024]. The heterogeneity of these approaches has led to a lack of consensus on which theoretical explanations most accurately capture the fundamental mechanisms underlying Adam's improved performance. For instance, Zhang et al. [2020] suggests it stems from enhanced robustness to heavy-tailed noise, while Kunstner et al. [2023] argues it plays no role.

This study focuses on a fundamental distinction between Adam and SGD: Adam's dependency on the coordinate system. SGD is rotation-equivariant, meaning if the loss landscape is rotated, the resulting optimization trajectories from SGD will be the same up to that rotation. In contrast, Adam produces substantially different trajectories. Although the sensitivity of adaptive methods to rotations of the parameter space is well established [Duchi et al., 2011], it remains unclear what properties of a rotation benefit or hinder performance, particularly in practical neural network training.

Our experimental investigation reveals that Adam's performance when training transformers empirically degrades when the objective function undergoes random rotations (Figure 1). This result demonstrates that Adam's effectiveness crucially depends on the canonical basis, challenging the adequacy of many existing theoretical frameworks used to analyze Adam's performance. Assumptions employed in the literature (Appendix F) are typically rotation-invariant, and thus the resulting frameworks are agnostic to the basis, preventing them from fully capturing or justifying Adam's empirical advantages.

Beyond theoretical motivations, recent studies have shown that applying rotations to the canonical basis can significantly enhance the performance of Adam and other optimizers [Vyas et al., 2025, Gupta et al., 2018, Jordan et al., 2024]. However, these rotations are often designed based on intuition and heuristics rather than systematically understanding their impact. By identifying the key properties that make a basis advantageous for Adam, we can develop more principled approaches to constructing optimal rotations that may outperform the existing rotation-based methods.

To understand the relationship between basis orientation and Adam's performance, we address two key questions:

> **Q1.** How do various types of rotations influence Adam's performance in practice?

We investigate **Q1** by conducting experiments in Section 3, examining Adam's convergence when rotating specific regions of the parameter space. We also identify some rotations that preserve or enhance Adam's performance. These findings provide a more nuanced picture of Adam's adaptive behaviour and its dependency on the basis.

Finally, as a rotation-invariant theoretical framework cannot fully capture Adam's advantage, we turn to rotation-dependent assumptions existing in the literature, and seek to answer:

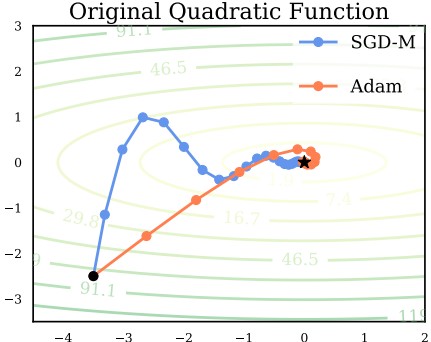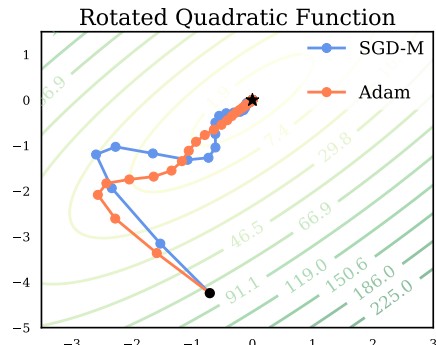

Figure 2: Trajectories of SGD-M and Adam on a quadratic under two different rotations. **SGD-M maintains the same trajectory up to rotation; Adam does not.**

---

**Q2.** What rotation-dependent assumptions adequately capture Adam's behaviour under rotations?

---

Section 4 examines three rotation-dependent assumptions used in literature in this context: $L_\infty$ bounded gradients, Hessian block-diagonality, and $L_\infty$-smoothness. Our analysis reveals that none of these conditions fully capture Adam's behaviour under rotations. Recently, Muon [Jordan et al., 2024] achieved strong performance by approximating orthogonalized gradients for each layer. Inspired by this, we measure the orthogonality of Adam's weight matrix updates for each layer (up to scalar multiplication) and find it closely aligns with performance across various rotations.

We summarize our key contributions:

1. **Analysis of Adam's sensitivity to various scopes of parameter space rotations in neural network training**: We conduct in Section 3.1 an empirical study demonstrating Adam's sensitivity to random parameter space rotations in practical training. We found a clear correlation between rotation scope (e.g., global, layer-wise) and performance, where a broader scope leads to greater degradation. In Section 3.2 we also employ a structured, SVD-based rotation inspired by Zhao et al. [2024] that improves Adam's convergence.

2. **Challenging existing rotation-dependent assumptions**: We assess the applicability of rotation-dependent properties in the literature by examining them jointly with our experimental study, and find that existing theoretical frameworks are not properly equipped to understand the beneficial properties of Adam.

3. **Verifying a better rotation-dependent quantity:** Given that SVD-based rotations improve performance, we analyze the singular values of the layer updates and find that their orthogonality is a strong predictor of Adam's performance across different bases. We also draw a connection to the Muon optimizer [Jordan et al., 2024], which approximates orthogonalized updates, and provide additional empirical support for its underlying intuition.

## 2 Preliminaries

Let $f : \mathbb{R}^d \to \mathbb{R}$ be the loss of a neural network with $d$ trainable parameters. Stochastic optimization algorithms approximate $\arg\min_{\mathbf{w} \in \mathbb{R}^d} f(\mathbf{w})$ by only accessing independent stochastic functions $f_B$ that depend on a stochastic minibatch $B$ following some data distribution $\mathcal{D}$ such that $\forall \mathbf{w} \in \mathbb{R}^d, \mathbb{E}_{B \sim \mathcal{D}}[f_B(\mathbf{w})] = f(\mathbf{w})$.

Our study examines the optimization process under **rotations of the parameter space**. More formally, let $SO(d)$ be the set of rotation matrices,

$$SO(d) = \left\{ \mathbf{R} \in \mathbb{R}^{d \times d} \colon \mathbf{R}^\top \mathbf{R} = \mathbf{R}\mathbf{R}^\top = \mathbf{I}, \det(\mathbf{R}) = 1 \right\}.$$

Instead of directly optimizing $f$, we consider its rotated counterpart $f^{(\mathbf{R})} : \mathbf{w} \to f(\mathbf{R}^\top \mathbf{w})$, $\mathbf{R} \in SO(d)$. This transformation rotates the coordinate system while preserving the geometry of the optimization landscape.

## 2.1 Rotational Equivariance of SGD

We say that an optimizer is **rotation equivariant** if, after a rotation of the parameter space, its trajectories are equally rotated.

**Definition 1** (Rotational equivariance). *Consider an optimization algorithm $\mathcal{A}$ applied to the function $f$, generating iterates $\mathbf{w}_{t+1} = \mathcal{A}(\{\mathbf{w}_i\}_{i=0,...,t}, f, t)$. We say that the optimization algorithm is **rotation equivariant** if it satisfies,*

$$\forall\, \mathbf{R} \in SO(d), \quad \mathbf{R}\mathbf{w}_{t+1} = \mathcal{A}(\{\mathbf{R}\mathbf{w}_i\}_{i=0,...,t}, f^{(\mathbf{R})}, t),$$

*where $f^{(\mathbf{R})} : \mathbf{w} \to f(\mathbf{R}^\top \mathbf{w})$ is the rotation of $f$.*

**Proposition 1.** *Stochastic Gradient Descent with momentum is rotation-equivariant.*

The rotation equivariance of SGD-M is a straightforward result as the gradient operator is rotation equivariant; we provide the proof in Appendix D for clarity. In contrast, Adam is not rotation equivariant, due to its element-wise division (Figure 2).

## 2.2 Training Neural Networks in Rotated Parameter Spaces

A crucial aspect of our study is the empirical evaluation of Adam's performance under parameter space rotations. Our approach (Figure 3) maintains the weights $\mathbf{w}_t$ in the standard basis and performs Adam's optimization steps in the rotated space. This allows us to leverage existing neural network frameworks for forward and backward propagation while examining Adam's behaviour under rotation.

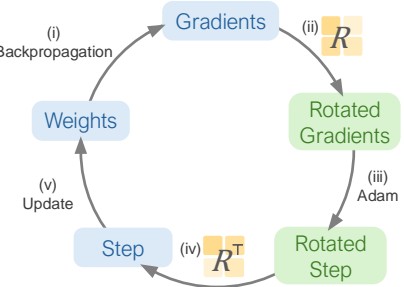

Figure 3: Methodology to train neural networks under parameter space rotations. (i) Forward and backward passes in the standard space to retrieve the gradients. (ii) The gradients are rotated using $\mathbf{R}$. (iii) Adam receives the rotated gradients and produces an update $\Delta\mathbf{w}^{(\mathbf{R})}$ in the rotated space. (iv) $\Delta\mathbf{w}^{(\mathbf{R})}$ is rotated back to the original space using $\mathbf{R}^\top$. (v) The parameters are updated with $\mathbf{R}^\top \Delta\mathbf{w}^{(\mathbf{R})}$.

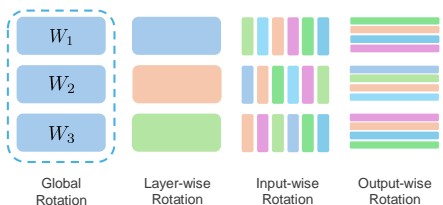

Figure 4: Illustration of different rotation scopes for a model with weights $\mathcal{W} \triangleq \{\mathbf{W}_1, \mathbf{W}_2, \mathbf{W}_3\}$. Global rotation rotates the entire parameter space at once, layer-wise only performs rotations within each layer subspace, and input-wise (resp. output-wise) rotates within the weights originating from a same input neuron (resp. leading to a same output neuron).

**Rotations in high dimension.** It is computationally intractable to operate with full $d \times d$ rotation matrices due to the size of modern neural networks. We employ a composite approach that combines block-diagonal rotations with strategic permutations to circumvent this limitation while preserving the essential characteristics of uniformly sampled rotations, effectively emulating the statistical properties of full-scale rotations. A detailed description and ablation studies are provided in Appendix A.

**Numerical considerations.** Neural network training is sensitive to numerical precision [Li et al., 2018, Wang et al., 2018, Sun et al., 2022], and it is crucial to ensure that rounding errors from rotations do not significantly confound the impact of the rotation. In particular, we apply rotations in single precision, and we refrain from using FlashAttention [Dao et al., 2022], which was found to increase numeric deviations [Golden et al., 2024]. We validate our methodology with ablations on SGD rotation equivalence, various rotation dimensions, and the use of FlashAttention in Appendix A.

# 3 Influence of Rotation on Adam's Efficiency

This section examines the effect of random rotations from neuron-wise to global (Section 3.1), showing that broader rotations degrade performance, while smaller-scale rotations have little to no impact. We then demonstrate that specific structured rotations can enhance Adam's performance.

## 3.1 Random Rotations

We first study the effect of four types of random rotations on Adam's performance (Figure 4): **Global** (entire parameter space), **Layer-wise** (per-layer subspaces; in transformers, keys/queries/values are treated separately), **Output-wise** (rotate weights where connections terminate at the same neuron in the subsequent layer), and **Input-wise** (rotate weights originating from the same neuron).

**Experimental setting.** We conduct experiments across three distinct settings spanning both transformer and non-transformer architectures, and language and vision tasks. Technical details and hyperparameters are provided in Appendix B.

- **Language modeling (GPT-2, Fig. 1a):** 124M-parameter decoder-only Transformer [Radford et al., 2019] trained on OpenWebText [Gokaslan and Cohen, 2019].
- **Image classification (ViT, Fig. 1b):** 22M-parameter Vision Transformer (ViT/S) [Dosovitskiy et al., 2021] evaluated on ImageNet-1K [Deng et al., 2009].
- **Image classification (ResNet, Fig. B):** ResNet-50 [He et al., 2016] on ImageNet-1K, where SGD often outperforms Adam [Keskar and Socher, 2017, Wilson et al., 2017].

**Results.** We make several key observations:

1. Adam's performance degrades under global rotations across all settings, confirming that the standard basis possesses advantageous properties.

2. The performance further degrades with broader rotation scopes. Layer-wise rotations, which preserve some basis structure, consistently outperform global rotations, highlighting the importance of local coordinate alignment.

3. ResNets exhibit minimal performance degradation under rotations. This reduced sensitivity suggests Adam obtains limited benefit from the standard basis structure in ResNets, which possibly explains its historically smaller marginal gain in training these networks.

4. Output-wise rotations show no degradation across all settings, with GPT2 even slightly improving. This suggests that Adam's adaptivity within output neurons is minimal, supporting recent approaches to reduce redundancy in Adam's second moments [Zhang et al., 2025].

Previous works [Zhang et al., 2024, 2025] have highlighted the heterogeneity across different parameter block types in Transformer architectures. In Appendix C, we restrict rotations to specific parameter types and study their individual impact on Adam's rotation sensitivity.

## 3.2 Investigating the Performance-Improving Rotations

Inspired by GaLore [Zhao et al., 2024], which uses low-rank Singular Value Decomposition (SVD) to compress optimizer states, we extend this concept to full-rank SVD to rotate the parameter space. Our approach decomposes the gradient matrix $\mathbf{G}$ of each layer into $\mathbf{G} = \mathbf{U}\mathbf{S}\mathbf{V}^\top$. This decomposition yields a natural rotation of the parameter space through the transformation $\mathbf{G} \rightarrow \mathbf{U}^\top\mathbf{G}\mathbf{V}$, which corresponds to an output-wise rotation (via $\mathbf{U}^\top$) and an input-wise rotation (via $\mathbf{V}$). We train a GPT2 model under the same conditions as in Section 3.1, but in this SVD-rotated space. We update the SVD decompositions every 250 steps.

Figure 5 shows that Adam's performance improves under SVD-based rotations, with a low computational overhead. These results highlight the potential of rotation-based approaches and motivate a more principled understanding of how basis orientation affects optimizer behaviour. Instead of relying on intuition or heuristics, we advocate for theory-driven design grounded in rotation sensitivity.

To further understand this behaviour, Figure 6 shows second moment distributions after training under various rotations. Global random rotations yield more concentrated second moments, implying less variation in effective learning rates and reduced adaptivity, which explains Adam's degraded

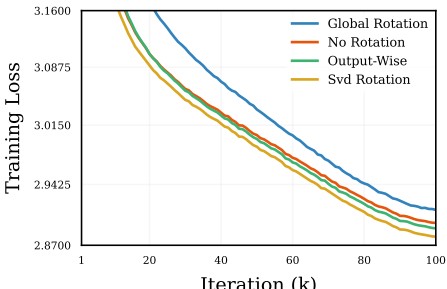

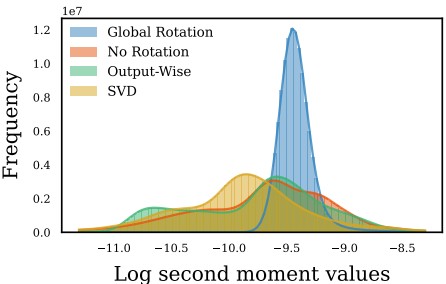

Figure 5: Performance of GPT2 trained with Adam in SVD-rotated space, without rotations, with random output-wise rotation and with random global rotation. **The rotations computed with SVD lead to sizeable improvement.**

Figure 6: Distribution of second-moment values for the final checkpoint of a GPT2 model trained with Adam in various rotated spaces. **Second moments are more concentrated under random rotations, indicating reduced adaptivity.**

performance. However, the benefits of SVD rotations are not apparent from second moments alone, suggesting a more subtle relationship between the parameter space and Adam's adaptive behaviour.

## 4 Examining Rotation Dependent Assumptions

We have established that Adam's performance depends heavily on the choice of basis. Rotation-invariant analyses yield identical guarantees for all bases, failing to capture performance gaps between rotations. A rotation-dependent assumption is necessary to explain Adam's practical advantage. In this section, we examine rotation-dependent assumption adequacy jointly with our GPT-2 experiments.

### 4.1 Adequacy of existing assumptions in theoretical frameworks

While rotation-invariant assumptions dominate optimization literature, some frameworks incorporate rotation-dependent properties. This subsection examines three existing assumptions and whether they adequately capture Adam's rotation dependency. In particular, we focuses on two aspects: (i) **Practical Feasibility.** The assumption must be realistic in practical settings. (ii) **Alignment with Adam's Performance.** An adequate property should have favourable constants under rotations that improve performance and break down (or have unfavourable constants) under rotations that hinder performance. An ideal theoretical convergence analysis should be based on realistic assumptions that relate the problem's characteristics to the optimizer's performance, where faster theoretical rates correspond to better practical performance.

$L_\infty$**-smoothness and** $(1,1)$**-norm.** $L_\infty$-smoothness was recently shown to guarantee the convergence of Adam, and presented as a potential key property of the basis [Xie et al., 2025, Balles et al., 2020]. We first remember its definition.

**Definition 2.** *A function $f$ is $C$-smooth wrt.* $\|\cdot\|_\infty$ *if* $\|\nabla f(\mathbf{x}) - \nabla f(\mathbf{y})\|_1 \leq C\|\mathbf{x} - \mathbf{y}\|_\infty \ \forall \ \mathbf{x}, \mathbf{y} \in \mathbb{R}^d$.

Given the challenges in directly estimating the $L_\infty$-smoothness constant, Xie et al. [2025] proposed using the (1,1)-norm of the Hessian as a surrogate, defined as:

$$\|\mathbf{H}\|_{(1,1)} := \sum_{m=1}^{M} \sum_{n=1}^{N} |\mathbf{H}_{mn}|,$$

where $\mathbf{H}_{mn}$ represents the element at the $m$-th row and $n$-th column of the Hessian matrix. Notably, they observed a degradation in their estimate of $\|\mathbf{H}\|_{(1,1)}$ under global random rotations. However, it remains unclear whether this degradation is a universal phenomenon for all rotations of the parameter space or if it specifically correlates with Adam's performance. To investigate this, we estimate the $(1,1)$-norm by averaging the $L_1$ norm of Hessian rows, sampled using the methodology described in Section 4.1. Figure 8 illustrates the change in $\|\mathbf{H}\|_{(1,1)}$ under global, SVD, and output-wise rotations.

Under global rotations, we confirm the $(1,1)$-norm degradation reported in [Xie et al., 2025], while SVD rotations improve it in line with Adam's performance gains, suggesting a link between this

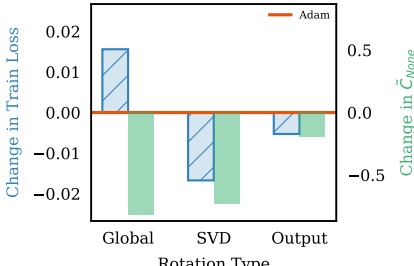
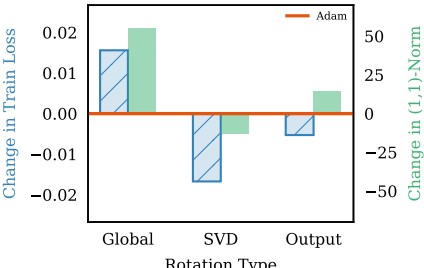

Figure 7: Empirical $L_\infty$ gradient bound $\tilde{C}$ over 1000 stochastic gradients at the last checkpoint for Global, SVD, and output-wise rotations, presented as differences from the non-rotated baseline. **The trend disagrees with Adam's performance, especially under global rotation.**

Figure 8: Estimated $(1,1)$-norm of the Hessian and final accuracy for Global, SVD, and output-wise rotations, presented as differences from the non-rotated baseline. **The $(1,1)$ norm correlates with Adam's performance on global and SVD rotations, but not on output-wise.**

$L_\infty$ **bounded gradients.** Reddi et al. [2018], Kingma and Ba [2015] assume a bound on the $L_\infty$ norm of stochastic gradients,

$$\forall\, \mathbf{w} \in \mathbb{R}^d,\ \|\nabla f_B(\mathbf{w})\|_\infty \le C \quad \text{almost surely.}$$

The constant $C$ depends on the basis, as the $L_\infty$ norm is not preserved under rotations. To evaluate this assumption, we compute the empirical bound on the rotated gradients,

$$\tilde{C}_{\mathbf{R}}(\mathbf{R}) := \max_{B_i} \|\nabla f_{B_i}^{(\mathbf{R})}(\mathbf{w_R})\|_\infty,$$

where $\mathbf{w_R}$ denotes the last checkpoint obtained by running Adam under rotation $\mathbf{R}$. The maximum is over 1000 stochastic minibatches $B_i$, across different rotations $\mathbf{R}$ (see Section 3). Figure 7 reveals that $\tilde{C}$ significantly decreases under random global rotations, predicting better performance, but we observe degradation in Adam's convergence. This discrepancy shows that the $L_\infty$ gradient bound fails to capture the beneficial properties of the basis for Adam.

geometric property and optimizer efficiency. Output-wise rotations yield slight performance gains but reduced $(1,1)$-norm, indicating that the metric does not capture all relevant factors. Overall, the $(1,1)$-norm shows promise as a rotation-sensitive indicator, especially under global and SVD rotations, but its limitations motivate the development of refined or complementary measures.

**Block-diagonality of the Hessian.** A common hypothesis for understanding Adam's behaviour is that the Hessian is well-approximated by a block-diagonal matrix [Zhang et al., 2024]. Then, random rotations likely disrupt block-diagonality and hinder convergence, while rotations within diagonal blocks preserve the structure, explaining the stable performance under output-wise rotations.

To examine this assumption's validity, we sample rows of the Hessian of $f^{(\mathbf{R})}$ at a checkpoint $\mathbf{w_R}$:

$$\mathbf{r_i}(\mathbf{R}) = \tfrac{1}{k}\sum_{j=1}^{k} \nabla^2 f_{B_j}^{(\mathbf{R})}(\mathbf{w_R})_{[i,:]} = \tfrac{1}{k}\sum_{j=1}^{k} \mathbf{e_i}^\top \mathbf{R} \nabla^2 f_{B_j}(\mathbf{w_R})\mathbf{R}^\top,$$

where $\mathbf{e_i}$ denotes the $i^{th}$ canonical basis vector. The vector $\mathbf{r_i}$ represents the average of the $i$-th row of the stochastic Hessian over $k$ minibatches. As $k$ increases, $\mathbf{r_i}$ converges to the true Hessian row. We set $k = 5000$ in the experiments, and use efficient Hessian-vector products [Dagréou et al., 2024].

We partition the indices of the Hessian row $\mathbf{r_i}$ corresponding to weight $w_i$ into three disjoint subsets:

$$\mathbf{r_i} = \mathbf{r_{i[I_N]}} + \mathbf{r_{i[I_L]}} + \mathbf{r_{i[I_{\not L}]}}, \quad \text{where}$$

- $I_N$ are indices of weights leading to the same output neuron as $\mathbf{w_i}$,
- $I_L$ are indices of other weights from the same layer,
- $I_{\not L}$ are indices of weights of other layers,

and $\mathbf{r_{i[I_N]}} = r_i$ (resp. $I_L$ and $I_{\not L}$) in the indices in $I_N$ (resp. $I_L$ and $I_{\not L}$) and zero elsewhere.

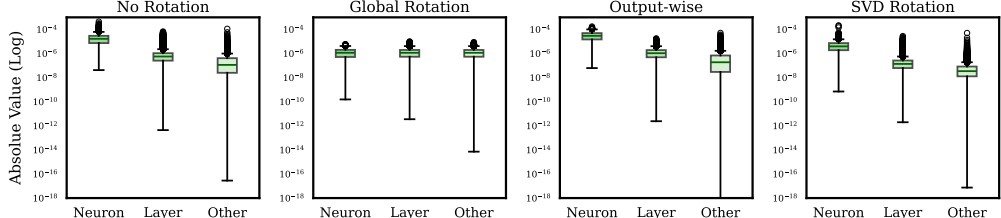

Figure 9: Distribution of Hessian values within output neuron, layer and non-layer in the second Transformer block attention projection layer. **In no rotation, values within the neuron are of magnitude higher than others, presuming a possible block-diagonal structure.** The structure is preserved in SVD and output-wise rotations, and lost in global rotation.

Figure 9 presents the distribution of absolute values for each subset. Our findings show that entries in $I_N$ and, to a lesser extent, $I_L$, are significantly larger than those from $I_{\not L}$, supporting an approximate block-diagonal Hessian structure.

Given this approximately block-diagonal Hessian structure, previous work [Zhang et al., 2024] proposes a strict block-diagonal approximation, assuming that the off-diagonal elements are negligible. We further investigate whether this simplification can accurately reflect the local geometry by assessing the practical implications of the block diagonal Hessian structure. We evaluate how each block contributes to gradient variations for a given small direction $\delta w$ via:

$$[\nabla f(\mathbf{w} + \delta\mathbf{w}) - \nabla f(\mathbf{w})]_i \approx \mathbf{e}_i^\top \nabla^2 f(\mathbf{w})\delta\mathbf{w} = \mathbf{r}_{\mathbf{i}[I_N]} \cdot \delta\mathbf{w} + \mathbf{r}_{\mathbf{i}[I_L]} \cdot \delta\mathbf{w} + \mathbf{r}_{\mathbf{i}[I_{\not L}]} \cdot \delta\mathbf{w}.$$

| $\delta\mathbf{w}$ direction | Random | Update |
|---|---|---|
| $\mathbf{r}_{\mathbf{i}[I_N]} \cdot \delta\mathbf{w}$ (Neuron) | $2.86 \times 10^{-5}$ | $-4.60 \times 10^{-10}$ |
| $\mathbf{r}_{\mathbf{i}[I_L]} \cdot \delta\mathbf{w}$ (Layer) | $-8.71 \times 10^{-6}$ | $1.30 \times 10^{-8}$ |
| $\mathbf{r}_{\mathbf{i}[I_{\not L}]} \cdot \delta\mathbf{w}$ (Other) | $\mathbf{1.48 \times 10^{-4}}$ | $\mathbf{2.02 \times 10^{-7}}$ |

Table 1: Contribution $\mathbf{r}_{\mathbf{i}[I]} \cdot \delta\mathbf{w}_{[I]}$ of Hessian values in block $I$ to the variation of the $i$-th gradient component in direction $\delta\mathbf{w}$. Averaged over multiple $\delta\mathbf{w}$, **off-diagonal blocks contribute significantly** in both random and update directions.

Table 1 quantifies these contributions in a random direction or update direction. Surprisingly, our results reveal that Hessian values outside the block are the primary drivers of gradient evolution, despite their smaller magnitude. This finding challenges the strict block-diagonal Hessian assumption in theoretical analyses. While the diagonal blocks contain larger values, their limited size compared to the full parameter space means that off-diagonal elements collectively play a crucial role in shaping the loss landscape's geometry. Neglecting off-diagonal elements is an oversimplification, making the approximation inadequate and potentially misleading downstream results.

### 4.2 Orthogonality of layer updates up to scalar factor

In Section 3.2 we find that SVD-based rotations improve Adam's performance. We also discuss in Appendix E connections with the recently proposed Muon optimizer [Jordan et al., 2024], which achieves strong performance by performing updates in the orthogonalized first moment direction.

**Scaled (semi-)orthogonality.** Since orthogonality is defined only for square matrices, we adopt a relaxed notion for rectangular matrices. Specifically, we say that a rectangular matrix is orthogonal if all its singular values are either 1 or 0 (this notion is commonly referred to as *semi-orthogonality* [Abadir and Magnus, 2005]). Moreover, we say that a matrix is a scaled orthogonal matrix if its eigenvalues are either $\alpha$ or 0, where $\alpha$ is the scaling parameter. To measure the scaled orthogonality of a matrix $\mathbf{A}$, we will use the coefficient of variation of its singular values $s_i$,

$$\mathrm{CV}(s_i) = \frac{\sigma_s}{\mu_s} = \min_\alpha \frac{1}{\mu_s} \sqrt{\frac{1}{n} \sum_i (s_i - \alpha)^2},$$

where $\mu_s$ and $\sigma_s$ are the mean and standard deviation of the $s_i$'s, respectively. We discuss in appendix C.4 other measures of scaled orthogonality of a matrix

**Scaled orthogonality of the layer update.** We resume training from a checkpoint for each rotation type for the next 500 steps to measure the orthogonality of the update $\mathbf{W}_{t+1}^{(l)} - \mathbf{W}_t^{(l)}$, where $\mathbf{W}_t^{(l)}$

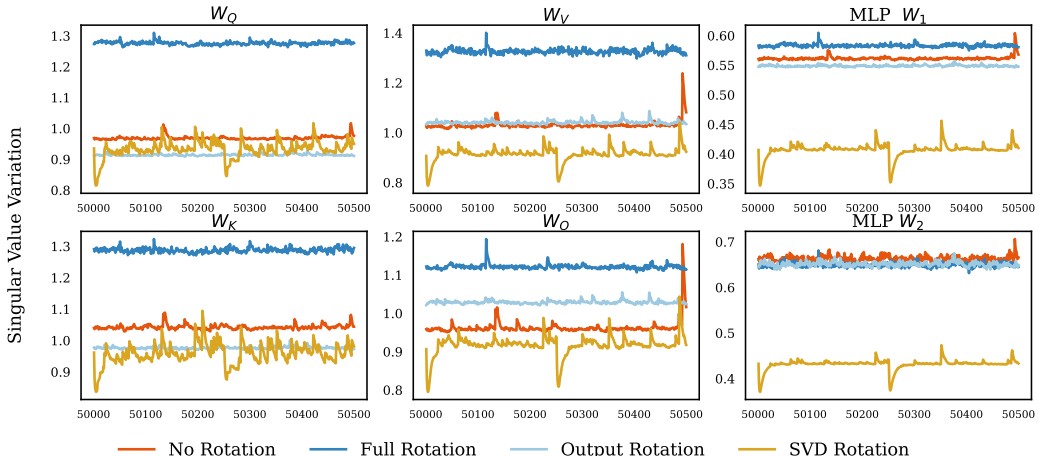

Figure 10: CV of singular values of layer updates over 500 steps, averaged over depth. SVD rotation consistently yields lower CV and more orthogonal layer updates, whereas full rotation shows the opposite. Downward spikes in the SVD rotation occur when the rotations are recomputed.

represents the weights of layer $l$ at time $t$. For simplicity, we omit the layer index $(l)$. To separate the effect of step size and weight decay, we measure the update as $\mathbf{A} = \mathbf{R}^{\top}\mathbf{M}_t^{(\mathbf{R})}/(\sqrt{\mathbf{V}_t^{(\mathbf{R})} + \epsilon})$.

**Observations.** Overall, we find that SVD consistently yields lower CV and more orthogonal updates under the coefficient of variation measure, and output-wise rotation behaves similarly to the no-rotation baseline. Full rotation consistently results in the least orthogonal updates. This ranking aligns clearly with the observed performance of Adam under rotations (Figure 11), making it a promising quantity to understand Adam's behaviour.

In Figure 10, we show the CV across time for each layer type, averaged over the depth. See Appendix C.4 for full results per layer and over time. Notably, CV drops right after the SVD rotation is recomputed every 250 steps. This offers insight into the frequency tradeoff of SVD rotations, where more frequent updates improve performance but introduce computational overhead. Overall, our analysis suggests that update orthogonality strongly correlates

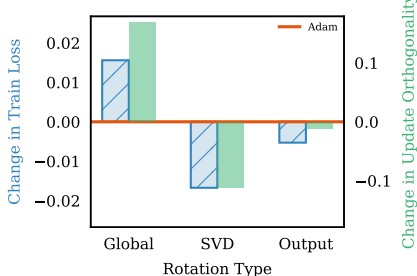

Figure 11: Each panel shows the difference in loss and average CV of singular values relative to the non-rotated baseline. **The orthogonality of layer updates closely aligns with performance under rotation.**

with optimizer performance, supporting the approaches of Muon [Jordan et al., 2024] and SOAP [Vyas et al., 2025] and opening new avenues for rotation-aware theoretical frameworks.

While a robust theoretical analysis of Muon's update rule remains open, attempts have been made to characterize in what context an orthogonal update is optimal. Bernstein [2025] argues that for linear layers, the orthogonalized update controls the scale at which the weight matrices can scale features. They argue this encourages stable optimization and can limit the need for normalization layers. As we discuss in Appendix E, after simplification and SVD rotation the Adam update simplifies to dividing the singular values by their magnitude. When rotated back, this results in the same update as Muon through a different mechanism, which is consistent with Muon's often superior performance.

## 5 Related Work

**Optimization under rotations.** Shampoo [Gupta et al., 2018] first demonstrated the benefits of optimizing under rotations by running AdaGrad under regularly updated rotations, identifying the singular values of the gradient matrices. More recently, SOAP [Vyas et al., 2025] improved Shampoo by applying it to Adam and appropriately updating the moments at each change of basis. Muon

[Jordan et al., 2024] concurrently explores a similar approach, but removes the need to explicitly store rotation matrices and instead orthogonalizes gradient matrices with a fast matrix iteration. While these methods show promising empirical improvements via heuristics, our work highlights the importance of developing better theoretical tools to understand their success.

On the theoretical side, we consider [Xie et al., 2025] to be the closest related study, showing that Adam converges more slowly with a randomly rotated loss landscape. They provide convergence analysis based on $L_\infty$ geometry, demonstrating that this yields a better empirical smoothness constant for GPT-2 models. While their work offers valuable theoretical insights, our study takes a more experimental stance. We aim to paint a comprehensive picture of Adam's behaviour under a spectrum of rotations, from random to structured transformations, and evaluate how existing rotation-invariant assumptions correlate with Adam's performance. Notably, Balles et al. [2020] also provides relevant insights through the lens of sign gradient descent.

**Understanding Adam.** Our work casts light on the critical interactions between Adam and the coordinate system, contributing to a growing body of research on Adam's behaviour and convergence. Recent works have attributed Adam's success to the heterogeneous block-diagonal structure of its Hessian [Zhang et al., 2024], though we find this assumption to be unrealistic. Others have improved convergence guarantees: Défossez et al. [2022] and Guo et al. [2022] offered simplified and novel derivations, Zhang et al. [2022] argued that vanilla Adam converges without modification, Zhou et al. [2024] provided a general convergence analysis for adaptive methods in non-convex settings, and Li et al. [2023b] proposed a convergence proof for Adam without relying on globally bounded gradients. Li et al. [2023a] developed a convergence analysis based on generalized smoothness conditions, and Hübler et al. [2023] proposed parameter-agnostic convergence results under these relaxed conditions. Finally, lower bounds for non-convex optimization were established by Arjevani et al. [2019], with Wang et al. [2023] addressing the gap between upper and lower bounds for Adam's iteration complexity.

**Adam's advantages over SGD.** Prior works have attempted to justify Adam's advantages over SGD. Zhang et al. [2020], Zhou et al. [2024] suggest SGD suffers more from heavy-tailed noise, with Adam converging faster when gradients are sparse. However, Kunstner et al. [2023] found that noise reduction through larger batch sizes benefits Adam but not SGD. Additionally, Kunstner et al. [2024] ties Adam's advantage in language models to ill-conditioning caused by heavy-tailed class imbalance, and Pan and Li [2022] to directional sharpness.

## 6 Discussion

**Limitations.** (i) Our purpose of the SVD rotation is not to introduce a new practical optimizer, but to demonstrate the existence of a more beneficial rotation and provide insights into the relationship between Adam and the standard basis. (ii) While our results reveal an alignment between the semi-orthogonality of layer updates and Adam's empirical performance, we do not offer theoretical guarantees, and further work is needed to formalize formalize this quantity and incorporate it into theoretical analysis. (iii) We relate our findings to the Muon optimizer and discuss the theoretical motivation for scaled orthogonality; however, there is still a lacking of rigours understanding of why Adam under SVD rotations produces more orthogonal updates than under the canonical basis, and why this quantity leads to improved performance. (iv) Finally, although the observed gap appears smaller in recent experiments, more evidence is required to confirm whether the superior performance of SGD on ResNets primarily comes from reduced sensitivity to rotations.

**Conclusion.** In this work, we have conducted a comprehensive investigation into Adam's sensitivity to rotations of the parameter space, revealing key insights into its optimization dynamics. We demonstrated that some rotations possess advantageous properties, opening new avenues for algorithmic contributions to adaptive algorithms. Our study demonstrates that Adam's performance is intricately tied to the choice of basis, a relationship that existing theoretical frameworks struggle to capture adequately. This investigation highlights the limitations of current rotation-invariant assumptions in explaining Adam's behaviour, and identifies update orthogonality as a promising theoretical tool. As the field evolves, we hope these findings will spark new avenues of research, potentially leading to more robust optimization algorithms and deepening our understanding of the fundamental principles underlying successful deep learning optimization.

## Acknowledgments and Disclosure of Funding

This research was partially supported by the Canada CIFAR AI Chair program (Mila) and Samsung Electronics Co., Ltd. Simon Lacoste-Julien is a CIFAR Associate Fellow in the Learning in Machines Brains program and acknowledges support by NSERC Discovery grant (RGPIN-2025-05123). We also acknowledge that this research was partly enabled by computing resources, software, and technical assistance provided by Mila and the Digital Research Alliance of Canada. Ioannis Mitliagkas acknowledges support by an NSERC Discovery grant (RGPIN-2019-06512). We thank Adam Ibrahim for his helpful comments and insights, and Ayoub Echchahed, Frederik Kunstner, Mark Schmidt, Pedram Khorsandi, Ryan d'Orazio, and Vitória Barin Pacela for their valuable feedback.

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

# Understanding Adam Requires Better Rotation Dependent Assumptions
# (Appendix)

# A   Sampling Random Rotations in High Dimension

This section explains our method of sampling random rotations for high-dimensional spaces and the implementation details.

## A.1   High-Dimensional Rotations

Even small modern machine learning models typically have millions of parameters. Consequently, storing a $d \times d$ rotation matrix is often intractable, let alone performing the dot product required to rotate the gradient vector. To address this issue, we sample a $n \times n$ rotation matrix $\mathbf{R}_n$ with $n \ll d$ uniformly (in the sense of the Haar measure) from the special orthogonal group $SO(n)$, and a random permutation $\pi$ of $0, \ldots, d-1$. For now, we assume $\frac{d}{n} \in \mathbb{N}$, see appendix A.3 for a general case. To rotate a gradient $g$, we compute:

$$g^{(\mathbf{R}_n, \pi)} := \pi^{-1} \circ \left( \left[ \bigoplus_{i=1}^{d/n} \mathbf{R}_n \right] (\pi \circ g) \right), \tag{1}$$

$$= \pi^{-1} \circ \begin{bmatrix} \mathbf{R}_n & & & \\ & \mathbf{R}_n & & 0 \\ & & \mathbf{R}_n & \\ & 0 & & \ddots \\ & & & & \mathbf{R}_n \end{bmatrix} (\pi \circ g), \tag{2}$$

where $\bigoplus$ denotes the direct sum operation, producing a block-diagonal matrix with $d/n$ blocks $\mathbf{R}_n$. This procedure effectively computes a rotation by blocks of size $n$ picked from a random partition of indices, constituting a valid rotation.

Intuitively, if $n$ is sufficiently large, we expect this procedure to approximate well the effect of random rotations sampled uniformly from $SO(d)$, due to the law of large numbers homogenizing geometric properties across coordinates. To confirm this intuition, we perform an ablation study in Figure 12, finding that the impact on Adam's performance saturates well below our operational values.

Our approximation reduces the memory cost from $O(d^2)$ to $O(n^2 + d)$, and the computational cost from $O(d^2)$ to $O(nd)$. Since batch matrix multiplications required for the rotation can be performed efficiently on modern GPUs, the final overhead of applying rotations is extremely small.

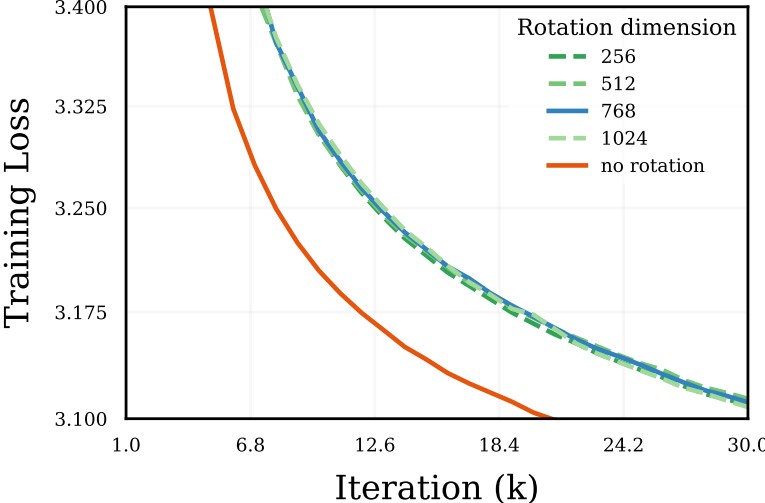

Figure 12: Training loss of GPT-2 when training with different rotation dimension $n$. The loss of performance is consistent across $n$ at our range.

## A.2 Reflections and Sampling From The Haar Measure

To sample $R_n$ uniformly from $SO(n)$ with respect to the Haar measure, we employ the QR decomposition trick [Mezzadri, 2007, Ozols, 2009], which samples from the Haar measure $\mu$ of the orthogonal group $O(n)$. Let us consider the projection $\pi : O(n) \to SO(n)$, such that $\pi(\mathbf{R})$ is $\mathbf{R}$ when $\mathbf{R} \in SO(n)$, and $\pi(\mathbf{R})$ simply multiplies the first column of $\mathbf{R}$ by $-1$ when $\mathbf{R} \in O(n) \setminus SO(n)$. The push forward of $\mu$ by $\pi$ is the Haar measure on $SO(n)$. Since Adam is reflection equivariant, rotating with $\pi(\mathbf{R})$ and with $\mathbf{R}$ will lead to identical performance for any $\mathbf{R} \in O(n)$. Thus we can omit to apply $\pi$, and simply sample from $\mu$ using the QR decomposition method.

Similarly, Adam is permutation equivariant; thus we omit to apply the inverse permutation before providing the rotated gradients to Adam, and to apply the permutation before rotating the update, as removing these two steps does not affect performance.

## A.3 Rotation Residual

Based on the type of rotation and the chosen dimension $n$, the number of blocks may not divide evenly, i.e., $\frac{d}{n} \notin \mathbb{N}$. To address this issue, we introduce an additional rotation matrix, which we refer to as the *residual* matrix, to complete the missing dimensions. More formally, let $d$ represent the dimensionality of the parameter space, and let $n$ denote the block dimensions of the rotation. We define $b \triangleq \lfloor \frac{d}{n} \rfloor$ as the number of complete blocks. The **residual** matrix $\mathcal{R}$ is then sampled from $SO(p)$, where $p \triangleq d - nb$. Therefore, eq. (1) becomes

$$g^{(\mathbf{R}_n, \mathcal{R}, \pi)} := \pi^{-1} \circ \left( [\mathbf{B} \oplus \mathcal{R}] \, (\pi \circ g) \right), \tag{3}$$

$$\tag{4}$$

$$= \pi^{-1} \circ \begin{bmatrix} \mathbf{B} & \\ & \mathcal{R} \end{bmatrix} (\pi \circ g), \tag{5}$$

$$= \pi^{-1} \circ \begin{bmatrix} \mathbf{R}_n & & & \\ & \mathbf{R}_n & & 0 \\ & & \ddots & \\ & 0 & & \mathbf{R}_n \\ & & & & \mathcal{R} \end{bmatrix} (\pi \circ g). \tag{6}$$

where $\mathbf{B} = \bigoplus_{i=1}^{b} \mathbf{R}_n$.

## A.4 Overall Validation and Impact of FlashAttention

In Figure 13, we present the training loss when training GPT-2 with SGD without rotations, with global random rotations using FlashAttention, and with global random rotations without FlashAttention. In particular, we confirm two important observations:

- Without FlashAttention (the setting we use for our experiments) the performances of SGD under global random rotation and under no rotations are identical. This validates that our experimental setting is behaving as expected.
- When we use FlashAttention with rotations, we observe a slight difference in performance. As explained in Section 2.2, this is due to FlashAttention amplifying numerical errors from the application of the rotation. Interestingly, likely due to a slight regularization effect, it it increases training performance.

# B  Experimental Details

This section provides additional details about the hyperparameters used for the architecture mentioned in the paper, as well as their optimizer and rotations.

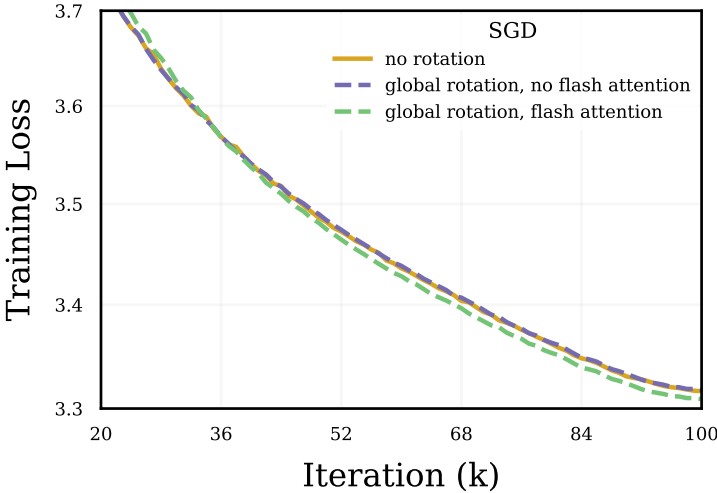

Figure 13: SGD performance when applying global random rotations, with and without FlashAttention.

## B.1 Rotations Design Choices

By default, for random rotations we fix the dimension of our rotation matrix $\mathbf{R}_n$ at 768 (which is the hidden dimension and thus makes residual rotations unnecessary for most rotation types). The matrix is sampled at the start of training, remains fixed throughout, and is shared across blocks the entire training process.

**SVD Rotation.**    Following [Zhao et al., 2024], this is the only rotation that is dynamic rather than static. Specifically, we compute the full-rank SVD decomposition of the gradient for each layer every 250 steps (recommended frequency in [Zhao et al., 2024]).

**Rotation in Transformers.**    By default, many implementations store the query, key, and value parameters within a single linear layer. Thus, we split them to treat them as separate layers, reflecting the fundamental differences in how their parameters are involved in forward computations. Additionally, PyTorch stores parameters as tensors in the shape (`output_dim`, `input_dim`), but embeddings are stored as lookup tables in the shape (`input_dim`, `output_dim`). For output neuron and input neuron rotations to behave intuitively, we thus transpose embedding layers before and after rotations.

## B.2 Architectures

**GPT2 (Transformer).**    We trained a GPT-2 model with 124M parameters on the OpenWebText dataset [Gokaslan and Cohen, 2019] using a configuration designed for efficient pretraining. The model architecture includes 12 layers, 12 attention heads, and a 768-dimensional embedding space, with no bias in LayerNorm or Linear layers. We employed the AdamW optimizer with a peak learning rate of $6 \times 10^{-4}$, $\beta_1 = 0.9$, $\beta_2 = 0.95$, and a weight decay of $0.1$, applying gradient clipping of $1.0$. Training ran for 100,000 iterations (or 30,000 for some smaller ablations), with learning cosine rate decay starting after a 2,000-iteration warm-up, decaying to a minimum of $6 \times 10^{-5}$. We used a sequence length of 1024 and micro batch size of 12 with gradient accumulation steps to simulate an effective batch size of 480 sequences. We additionally tried tuning $\beta_2$ by using values $0.9$ and $0.99$. We found that the base AdamW showed slightly better performance, but the globally rotated model's performance decreased with both of these values, meaning further tuning will not close the gap with the base model. All experiments were performed on four A100 80GB GPUs, leveraging mixed precision. Unless otherwise specified, all optimizer hyperparameters were shared across experiments and set to the default values specified in Karpathy [2022].

**ViT (Vision Transformer).**    We trained a Vision Transformer (ViT) model on the ImageNet-1K dataset [Deng et al., 2009] using the SimpleViT architecture [Beyer et al., 2022]. The model consists

of 12 layers, 6 attention heads, a hidden dimension of 384, and an MLP dimension of 1536, with a patch size of 16 and input image size of 224. The AdamW optimizer was employed with a learning rate of 0.001, $\beta_1 = 0.9$, $\beta_2 = 0.999$, $\epsilon = 10^{-8}$, and a weight decay of 0.1. We used a cosine learning rate schedule with 5 warm-up epochs. The training was conducted for 100 epochs with a batch size of 1024. All experiments were performed with mixed precision.

**ResNet-50 (CNN).** We trained a ResNet-50 model [He et al., 2015] on the ImageNet-1K dataset [Deng et al., 2009] using the AdamW optimizer. The optimizer was configured with a learning rate of 0.001, $\beta_1 = 0.9$, $\beta_2 = 0.999$, $\epsilon = 10^{-8}$, and a weight decay of 0.0001. We employed a cosine learning rate schedule with 5 warm-up epochs. The training ran for 100 epochs with a batch size of 256.

### B.3 Assumptions Estimation

We now outline how we computed empirical estimations of assumptions in Section 4.

$L_\infty$**-bounded gradient.** Algorithm 1 describes the process we use to estimate the bound constant $\tilde{C}$ of stochastic gradients under $L_\infty$ norm, as detailed in section 4.1.

---

**Algorithm 1** Empirical Gradient Bound Estimation for Adam

---

**Require:** $T$: total number of iterations (1000)
**Require:** $\mathbf{w_R}$: last checkpoint obtained by running Adam under rotation $\mathbf{R}$
  1: Initialize $\tilde{C} \leftarrow 0$                 ▷ Maximum infinity norm of gradients
  2: **for** $t \leftarrow 1$ **to** $T$ **do**
  3:     Sample a minibatch $B_i$                 ▷ Select one minibatch
  4:     $\mathbf{g}_{B_i} \leftarrow \nabla f_{B_i}^{(\mathbf{R})}(\mathbf{w_R})$          ▷ Compute gradient for minibatch
  5:     $\tilde{C}' \leftarrow \|\mathbf{g}_{B_i}\|_\infty$          ▷ Compute infinity norm of the gradient
  6:     $\tilde{C} \leftarrow \max(\tilde{C}, \tilde{C}')$        ▷ Update the maximum gradient bound
  7: **end for**
  8: **return** $\tilde{C}$                 ▷ Return the estimated gradient bound

---

$(1, 1)$**-Norm.** Using the Hessian rows sampled from GPT-2 checkpoints that were trained under various rotations in Section 4.1, we estimate $\frac{\|\mathbf{H}\|_{(1,1)}}{d}$ by averaging the $L_1$ norm of sampled rows. While this could induce a large variance from the sampling of rows, we find that variations of the $L_1$ norms from rotations are fairly homogeneous across rows.

## C   Additional Results

### C.1   Main Experiments

We provide additional results from our main line of experiments.

**ViT/S (ImageNet).** Figure 14 extends the results from Figure 1b with validation loss and accuracy

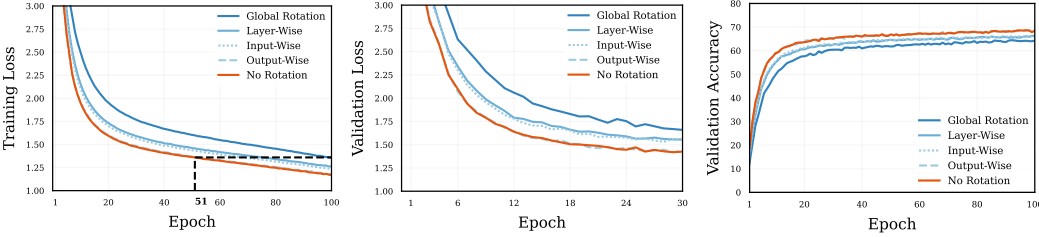

Figure 14: SimpleViT - Imagenet training loss, validation loss and top-1 validation accuracy

**ResNet50 (ImageNet).**   Figure 15 demonstrates that Adam maintains its performance well under rotational transformations for ResNets. This robustness to rotation implies that Adam gains little advantage from the standard basis structure in this setting. This finding aligns with the fact that SGD with extensive tuning can outperform Adam when training these networks.

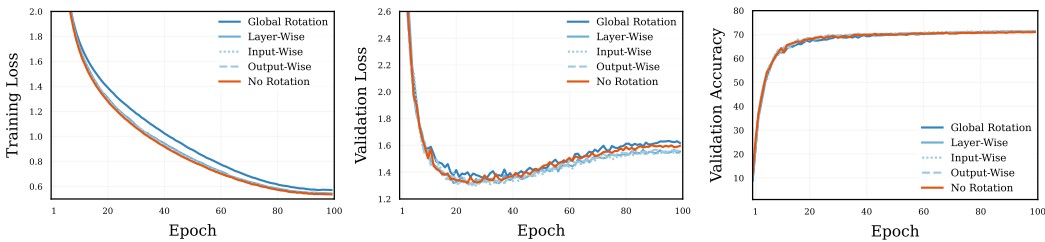

Figure 15: Training loss, validation loss and top 1 % validation accuracy, when training a ResNet-50 with Adam on ImageNet across different scopes of rotations.

## C.2   Architecture Aware Rotation.

We seek to identify whether certain transformer layer types are more sensitive to rotations than other, and contribute more to the overall performance degradation observed in Figure 1a when using layer-wise rotations.

Figure 16 shows the loss curves when rotating only one layer type at a time. We find that the performance degradation induced by layer-wise rotations is small for most layer types, seemingly balanced across these layers, with the exception of value and embedding layers.

Layerwise rotation of value layers seem to impact the loss more noticeably than with other layer types. In Figure 18, we find that reducing the scope of rotations to output neuron wise does not improve the performance when rotating value layers.

The biggest drop of performances is observed for embedding layers, which we conjecture to be linked to the discrepancy in frequency across tokens. Figure 17 shows indeed that when rotating the embedding layer by output neuron (i.e., within weights corresponding to a same token) the degradation becomes unnoticeable.

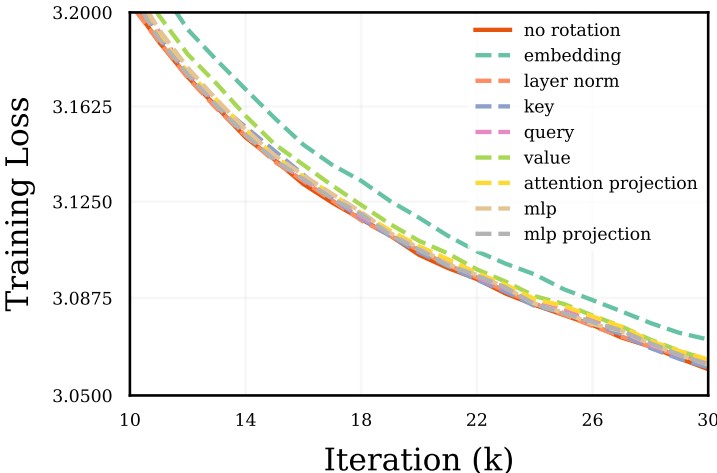

Figure 16: Layer-wise rotation applied to only specific layer types

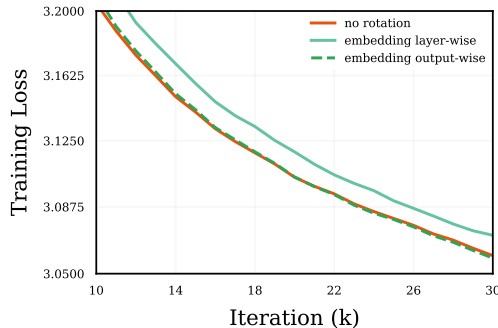

Figure 17: Layer-wise rotation and output-wise rotation on embedding layers only

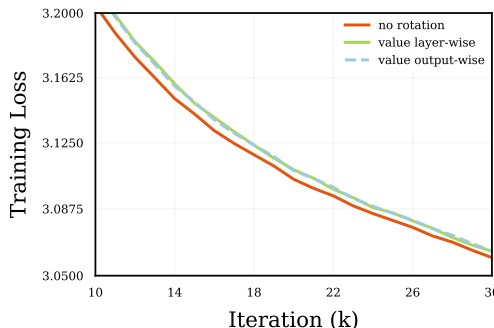

Figure 18: Layer-wise rotation and output-wise rotation on attention values only

### C.3 Hessian Rows

We use the end checkpoint of GPT-2 to sample rows from the Hessian in different rotated parameter spaces (see Section 4.1).

From Figure 19 to Figure 25, we present the same figure as in Figure 9, but for rows taken from different layer types, confirming that the behaviour we observed is consistent across parameter types. Except for embeddings, rows are always taken from the second Transformer block.

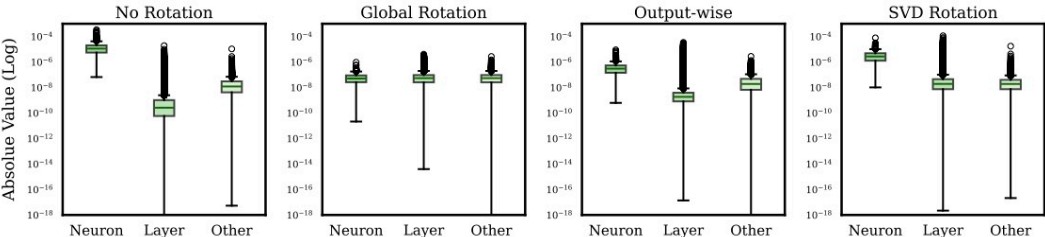

Figure 19: Hessian value distribution of a row in the embedding layer.

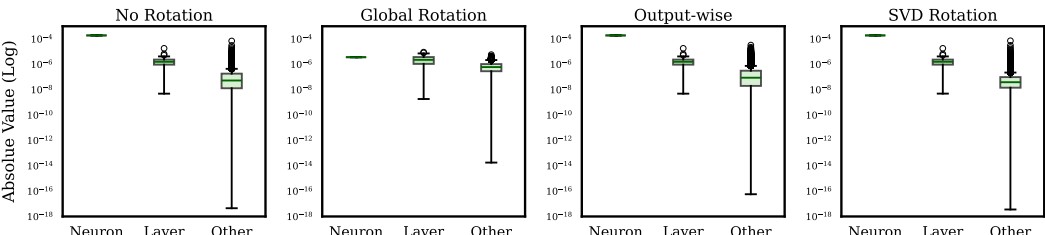

Figure 20: Hessian value distribution of a row in the second Transformer layer norm layer.

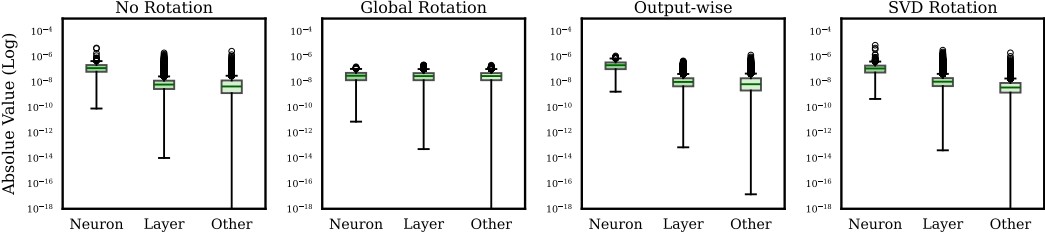

Figure 21: Hessian value distribution of a row in the second Transformer key layer.

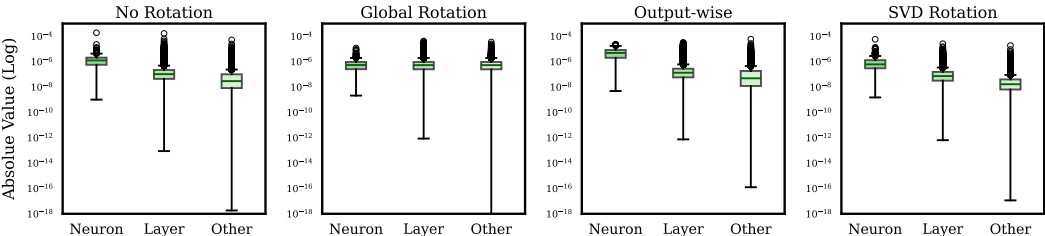

Figure 22: Hessian value distribution of a row in the second Transformer query layer.

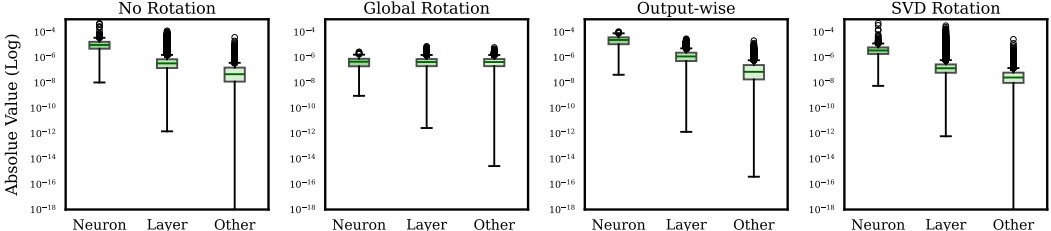

Figure 23: Hessian value distribution of a row in the second Transformer value layer.

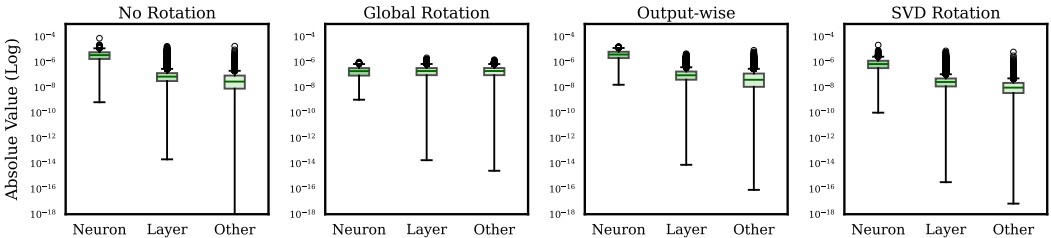

Figure 24: Hessian value distribution of a row in the second Transformer mlp layer.

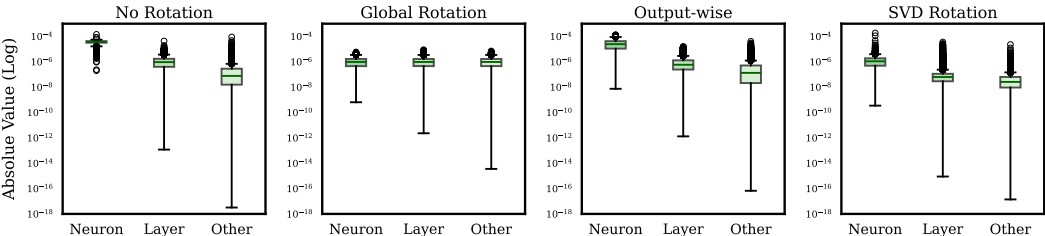

Figure 25: Hessian value distribution of a row in the second Transformer mlp projection layer.

Figure 26 shows a row in the attention projection layer of the 8-th transformer block, showing our observations seem also consistent across depth.

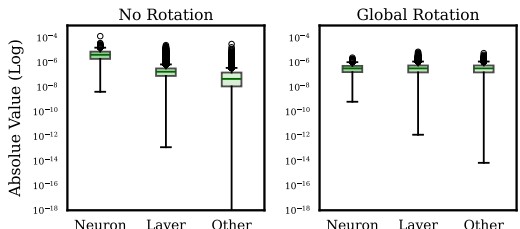

Figure 26: Hessian value distribution of a row in the eighth Transformer attention projection layer.

Figure 27 uses checkpoints trained with the same rotations as the one applied to the Hessian. We find the same behaviour for no rotations, global and output-wise, but we find that with the SVD-rotated checkpoints, there is increased variance in the Hessian values outside of the layer.

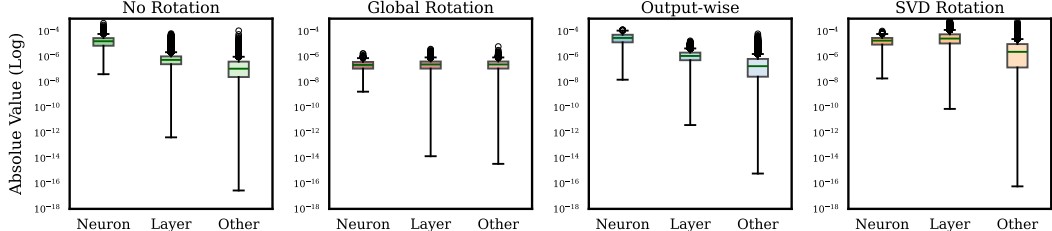

Figure 27: Hessian value distribution of a row in the second Transformer attention projection layer from checkpoints that are *trained with different rotations*.

## C.4 Update orthogonality

We aim to measure the orthogonality of the update. Since weight decay is applied directly in parameter space rather than to the gradient, the update rule (ignoring the bias corrections) becomes:

$$\mathbf{W}_{t+1} - \mathbf{W}_t = -\alpha_t \mathbf{R}^\top \frac{\mathbf{M}_t}{\sqrt{\mathbf{V}_t} + \epsilon} - \alpha_t \lambda \mathbf{W}_t, \tag{7}$$

where

$$\mathbf{M}_t = \beta_1 \mathbf{M}_{t-1} + (1 - \beta_1) \mathbf{R} \nabla \mathcal{L}(\mathbf{W}),$$
$$\mathbf{V}_t = \beta_2 \mathbf{V}_{t-1} + (1 - \beta_2) \left( \mathbf{R} \nabla \mathcal{L}(\mathbf{W}) \circ \mathbf{R} \nabla \mathcal{L}(\mathbf{W}) \right).$$

To isolate the effective update direction, we rewrite the expression as:

$$\frac{\mathbf{W}_{t+1} - (1 - \alpha_t \lambda) \mathbf{W}_t}{-\alpha_t} = \mathbf{R}^\top \frac{\mathbf{M}_t}{\sqrt{\mathbf{V}_t} + \epsilon} := \mathbf{A}. \tag{8}$$

**Coefficient of variation.** Suppose matrix $\mathbf{A}$ has singular values $\{s_i\}$. For a scale-invariant measure of orthogonality, we minimize the normalized root mean square deviation between singular values and a constant,

$$\min_\alpha \frac{1}{\mu} \sqrt{\frac{1}{n} \sum_i (s_i - \alpha)^2},$$

where $\mu_s = \frac{1}{n} \sum_i s_i$ is the mean of $\{s_i\}$. The normalization facilitates the comparison between data with different scales. The solution for this objective is the *coefficient of variation (CV)*, defined as

$$\mathrm{CV}(s_i) = \frac{\sigma_s}{\mu_s},$$

where $\sigma_s$ is the standard deviation of the singular values. CV captures the *relative dispersion* of the singular values and is invariant to uniform rescaling of $\mathbf{A}$. This makes it especially suitable for comparing updates or transformations that differ in magnitude but share underlying structure. Since an orthogonal matrix has all singular values equal to 1, a lower CV indicates that the singular values are more tightly clustered, suggesting that $\mathbf{A}$ is closer to being orthogonal up to a global scaling.

In Figure 10, we provided a mean variation of the singular values in the update for each layer type, averaged across the depth of the network using GPT-2. We now expand on this and show the variation for each individual weight matrix in the network. We display the coefficient of variation of the singular values of each linear layer's update. Each of the GPT-2 model's twelve encoder layer has six linear layers, four in the attention layer after splitting (to compute $\mathbf{Q}, \mathbf{K}, \mathbf{V}$ and the output projection $\mathbf{O}$) and two in the MLP block (labeled $\mathbf{W}_1$ and $\mathbf{W}_2$). We compute the SVD of each update, and compute the coefficient of variation of the singular values and show the results in Figure 28. For completeness, we also show this result at the beginning and end of training in Figure 29 and Figure 30 respectively. The trend is clearer and steadier at the end of training.

**Other metrics.** In our experimentation, we considered other metrics to measure orthogonality, with each finding similar results.

One alternative method is to measure how far the scaled singular values are from 1:

$$\frac{1}{n}\sum_{i=1}^{n}(\alpha s_i - 1)^2$$

where $\alpha$ is a scaling factor shared across the singular values $s_i$ of the update. For a set of singular values $\{s_1, \ldots, s_n\}$, the optimal scaling factor is

$$\alpha^* = \sum_{i=1}^{n}\frac{s_i}{\sum_{i=1}^{n}s_i^2}.$$

We display measures at the start, middle, and end of training in Figure 31 respectively. We found this to be strongly correlated with the coefficient of variation of $\{s_i\}$.

Similarly, another measure of semi-orthogonality is how close $\mathbf{A}\mathbf{A}^\top$ is to the identity matrix $\mathbf{I}$. The objective being

$$\min_{\alpha}\frac{1}{\mu}\sqrt{\frac{1}{n}\|\mathbf{A}\mathbf{A}^\top - \alpha\mathbf{I}\|_F^2} = \frac{\sigma_\lambda}{\mu_\lambda} = \mathrm{CV}(\lambda),$$

where $\lambda = s_i^2$ are the eigenvalues of $\mathbf{A}\mathbf{A}^\top$, and $\sigma_\lambda$ and $\mu_\lambda$ are the standard deviation and mean of $\lambda$.

To see this, since $\mathbf{A}\mathbf{A}^\top$ is symmetric, so it has an eigendecomposition by the Spectral theorem, i.e. $\mathbf{A}\mathbf{A}^\top = \mathbf{Q}\mathbf{\Lambda}\mathbf{Q}^{-1}$ where $\mathbf{Q}$ is an orthogonal matrix. Then,

$$\mathbf{Q}\mathbf{\Lambda}\mathbf{Q}^{-1} - \alpha\mathbf{I}$$
$$\mathbf{Q}\mathbf{\Lambda}\mathbf{Q}^{-1} - \alpha\mathbf{Q}\mathbf{Q}^{-1}$$
$$\mathbf{Q}\mathbf{\Lambda}\mathbf{Q}^{-1} - \alpha\mathbf{Q}\mathbf{Q}^{-1}$$
$$\mathbf{Q}(\mathbf{\Lambda} - \alpha\mathbf{I})\mathbf{Q}^{-1}$$

Then, using the circulant property of the trace, we have

$$\begin{aligned}\|\mathbf{Q}(\mathbf{\Lambda} - \alpha\mathbf{I})\mathbf{Q}^{-1}\|_F^2 &= \mathrm{Tr}\left((\mathbf{Q}(\mathbf{\Lambda} - \alpha\mathbf{I})\mathbf{Q}^{-1})^\top\mathbf{Q}(\mathbf{\Lambda} - \alpha\mathbf{I})\mathbf{Q}^{-1}\right)\\ &= \mathrm{Tr}\left(\mathbf{Q}(\mathbf{\Lambda} - \alpha\mathbf{I})^\top\mathbf{Q}^{-1}\mathbf{Q}(\mathbf{\Lambda} - \alpha\mathbf{I})\mathbf{Q}^{-1}\right)\\ &= \mathrm{Tr}\left(\mathbf{Q}(\mathbf{\Lambda} - \alpha\mathbf{I})^\top(\mathbf{\Lambda} - \alpha\mathbf{I})\mathbf{Q}^{-1}\right)\\ &= \mathrm{Tr}\left(\mathbf{Q}^{-1}\mathbf{Q}(\mathbf{\Lambda} - \alpha\mathbf{I})^\top(\mathbf{\Lambda} - \alpha\mathbf{I})\right)\\ &= \mathrm{Tr}\left((\mathbf{\Lambda} - \alpha\mathbf{I})^\top(\mathbf{\Lambda} - \alpha\mathbf{I})\right)\\ &= \sum_{i=1}^{n}(\lambda_i - \alpha)^2\\ &= n\sigma_\lambda^2 \qquad \text{Substituting optimal }\alpha^*\end{aligned}$$

The optimal $\alpha^*$ in this case is $\mu_\lambda$. The normalization by mean assures scale invariance, and yields a coefficient of variation of $\lambda$. We show this metric in Figure 32.

Instead, if the scaling factor is used on the symmetric matrix $\mathbf{A}\mathbf{A}^\top$, we have the objective

$$\min_{\alpha}\sqrt{\frac{1}{n}\|\alpha\mathbf{A}\mathbf{A}^\top - \mathbf{I}\|_F^2},$$

which is already scale invariant and yields the optimal scaling factor $\alpha^* = \sum_{i=1}^{n}\frac{\lambda_i}{\sum_{i=1}^{n}\lambda_i^2}$. We also plot this metric in Figure 33.

Additionally, Liu and Su [2025] use a metric inspired by the signal processing literature called the SVD entropy to study Muon updates, which is defined as follows.

$$H(s) = -\frac{1}{\log(n)} \sum_{i=1}^{n} \frac{s_i^2}{\sum_{j=1}^{n} s_j^2} \log \left( \frac{s_i^2}{\sum_{j=1}^{n} s_j^2} \right)$$

We also computed this metric, and it again strongly correlated with the coefficient of variation. We display measures at the start, middle, and end of training in Figure 34 and note that larger is better for this metric, in contrast to the others we consider. While all metrics showed roughly the same trends, we focus on the coefficient of variation of singular values for simplicity.

# D Optimization Algorithms with Rotations

We remind here the SGD-M algorithm in Algorithm 2, AdamW algorithm (pseudocode) in Algorithm 3, and provide a rotated version in Algorithm 4.

---

**Algorithm 2** SGD Momentum Optimization Algorithm

---

**Require:** $\alpha$: stepsize
**Require:** $\beta$: momentum parameter
**Require:** $\lambda$: weight decay coefficient
**Require:** $f(\boldsymbol{\theta})$: stochastic objective function with parameters $\boldsymbol{\theta}$
 1: Initialize $\boldsymbol{\theta}_0$, $t \leftarrow 0$
 2: **while** $\boldsymbol{\theta}_t$ not converged **do**
 3:      $t \leftarrow t + 1$
 4:      $\mathbf{g}_t \leftarrow \nabla_\theta f_t(\boldsymbol{\theta}_{t-1})$                   ▷ Get gradients w.r.t. stochastic objective at timestep $t$
 5:      $\boldsymbol{\theta}_t \leftarrow \boldsymbol{\theta}_{t-1} - \alpha\mathbf{g}_t + \beta(\boldsymbol{\theta}_{t-1} - \boldsymbol{\theta}_{t-2}) - \alpha\lambda\boldsymbol{\theta}_{t-1}$              ▷ Update parameters
 6: **end while**
 7: **return** $\boldsymbol{\theta}_t$                                         ▷ Return the final parameters

---

*Proof of section 2.1.* Using the notation of Section 2, we consider SGD with momentum with learning rate $\eta$, momentum parameter $\beta$, and fixed batches $B_t$:

$$\mathbf{w}_{t+1} = \mathcal{A}(\{\mathbf{w}_i\}, f, t) = \mathbf{w}_t - \eta\nabla f_{B_t}(\mathbf{w}_t) + \beta(\mathbf{w}_t - \mathbf{w}_{t-1}),$$

By the chain rule, $\nabla f_{B_t}^{(\mathbf{R})}(\mathbf{w}_t) = \mathbf{R}\nabla f_{B_t}(\mathbf{R}^\top\mathbf{w}_t)$, hence:

$$\begin{aligned}
\mathbf{w}_{t+1}^{(\mathbf{R})} &:= \mathcal{A}(\{\mathbf{R}\mathbf{w}_i\}_{i=0,\dots,t}, f^{(\mathbf{R})}, t) \\
&= \mathbf{R}\mathbf{w}_t - \eta\mathbf{R}\nabla f_{B_t}(\mathbf{R}^\top\mathbf{R}\mathbf{w}_t) + \beta(\mathbf{R}\mathbf{w}_t - \mathbf{R}\mathbf{w}_{t-1}) \\
&= \mathbf{R}\mathbf{w}_t - \eta\mathbf{R}\nabla f_{B_t}(\mathbf{w}_t) + \beta\mathbf{R}(\mathbf{w}_t - \mathbf{w}_{t-1}) \\
&= \mathbf{R}\mathbf{w}_{t+1}
\end{aligned}$$

matching Definition 1. □

While the SVD rotation we use in the AdamW algorithm can be represented as in Algorithm 4 mathematically for a specific choice in $R$, for clarity and to match our implementation, we write the SVD rotated AdamW in Algorithm 5. In our experiments, the SVD update frequency $F$ was set to 250 steps. We note that while the other rotations we study are written as matrix-vector products, the SVD rotation is written as a left and right matrix product on the gradient matrix. These can be shown to be mathematically equivalent, but we clarify given the standard practice of writing the gradient as a vector.

---

We consider the heavy ball formulation here [Polyak, 1964], but the same would hold for Nesterov Accelerated Gradient [Nesterov, 1983].

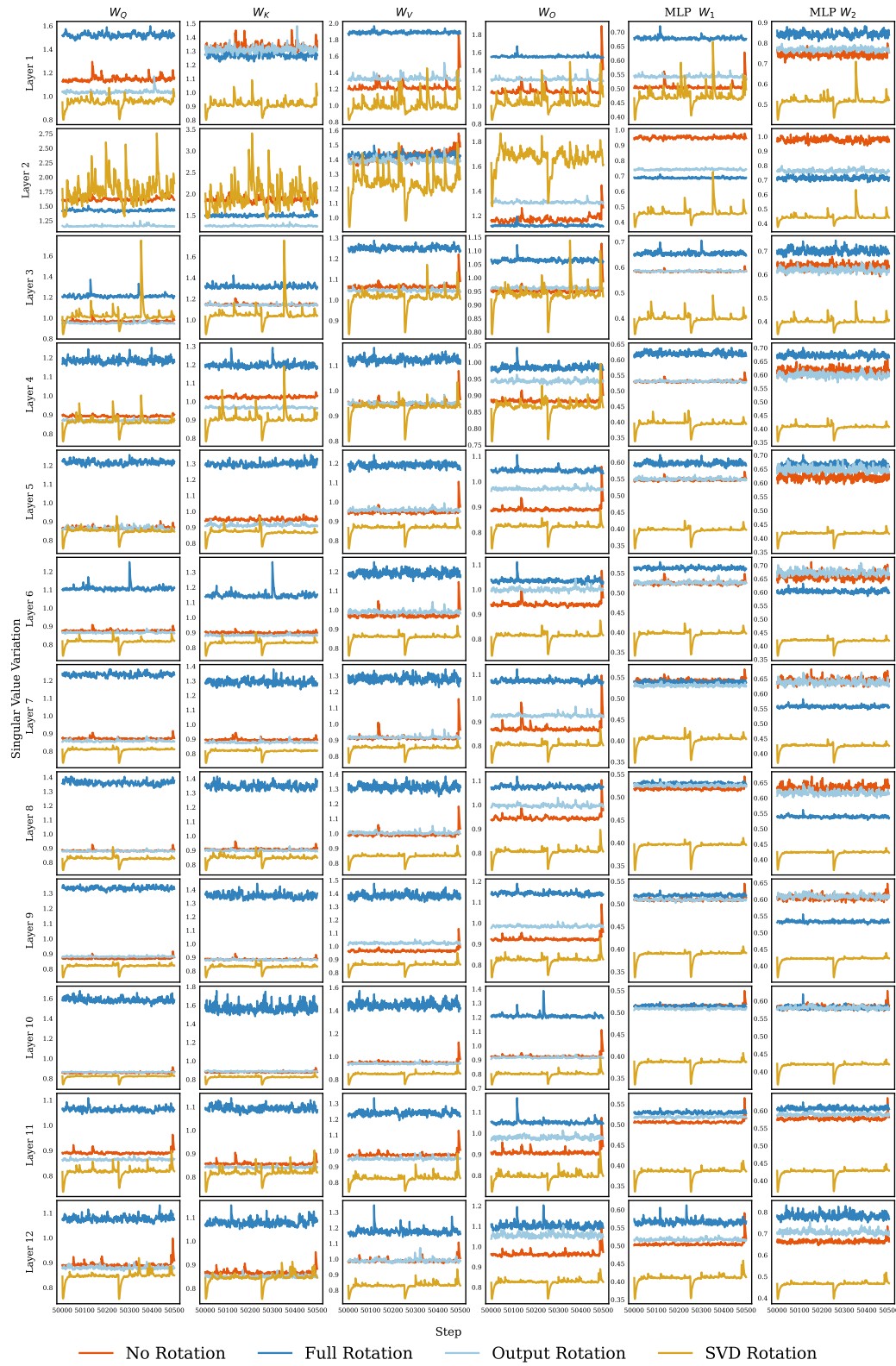

Figure 28: Singular value variation during training. We measure the variation of the singular values of each update of AdamW under various rotations for every linear layer in each GPT2 encoder block. While not universal, we find that a majority of the time, the SVD rotations leads to the lowest variation, while the global rotations leads to the highest. Additionally, we see that recomputing the SVD matrices (at 50000 and 50250 steps) leads to a downward spike in the variation.

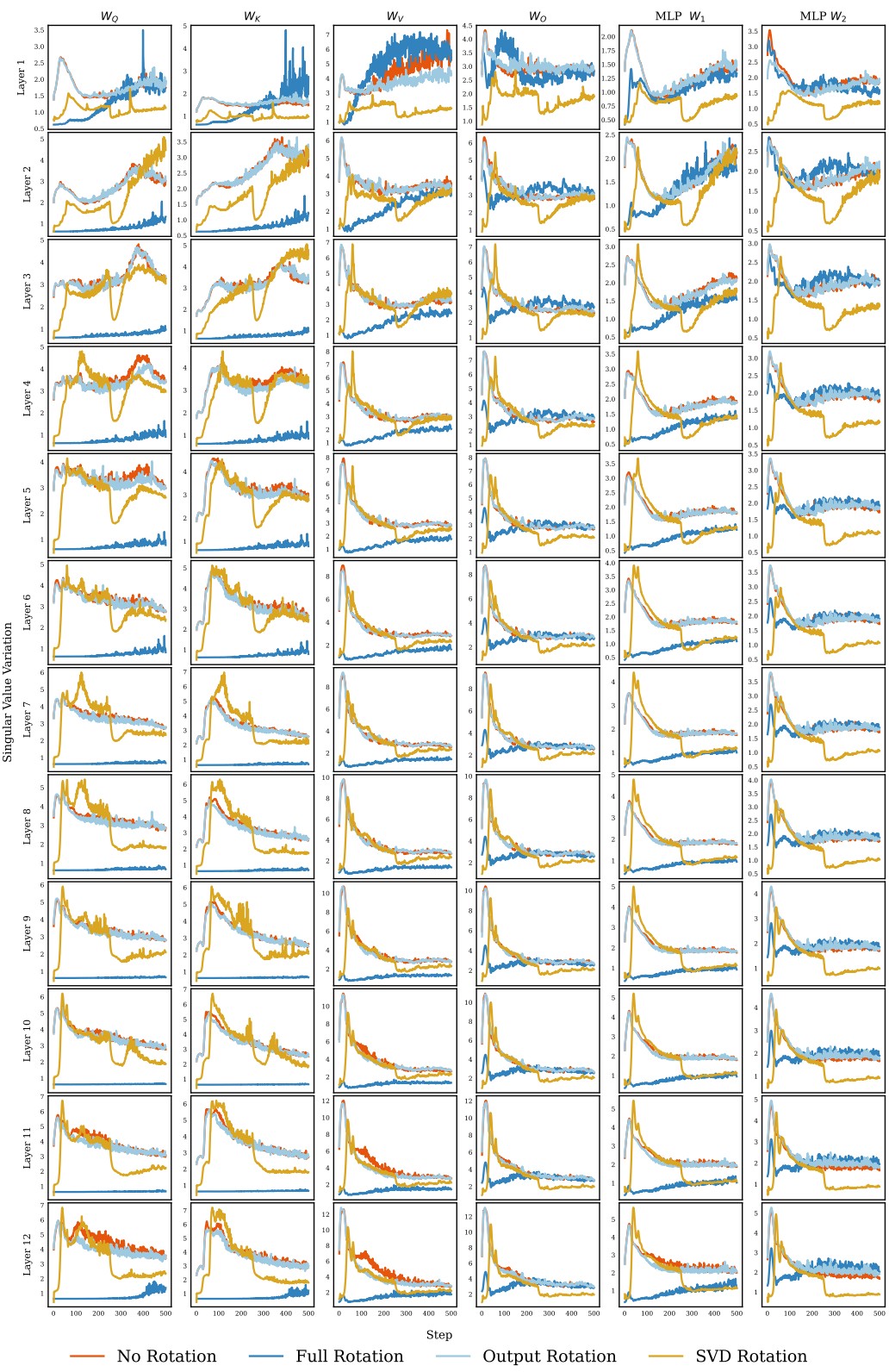

Figure 29: Singular value variation at the beginning of training. While the variation increases at initialization, we see it begin to trend downwards, notably after the SVD is recomputed.

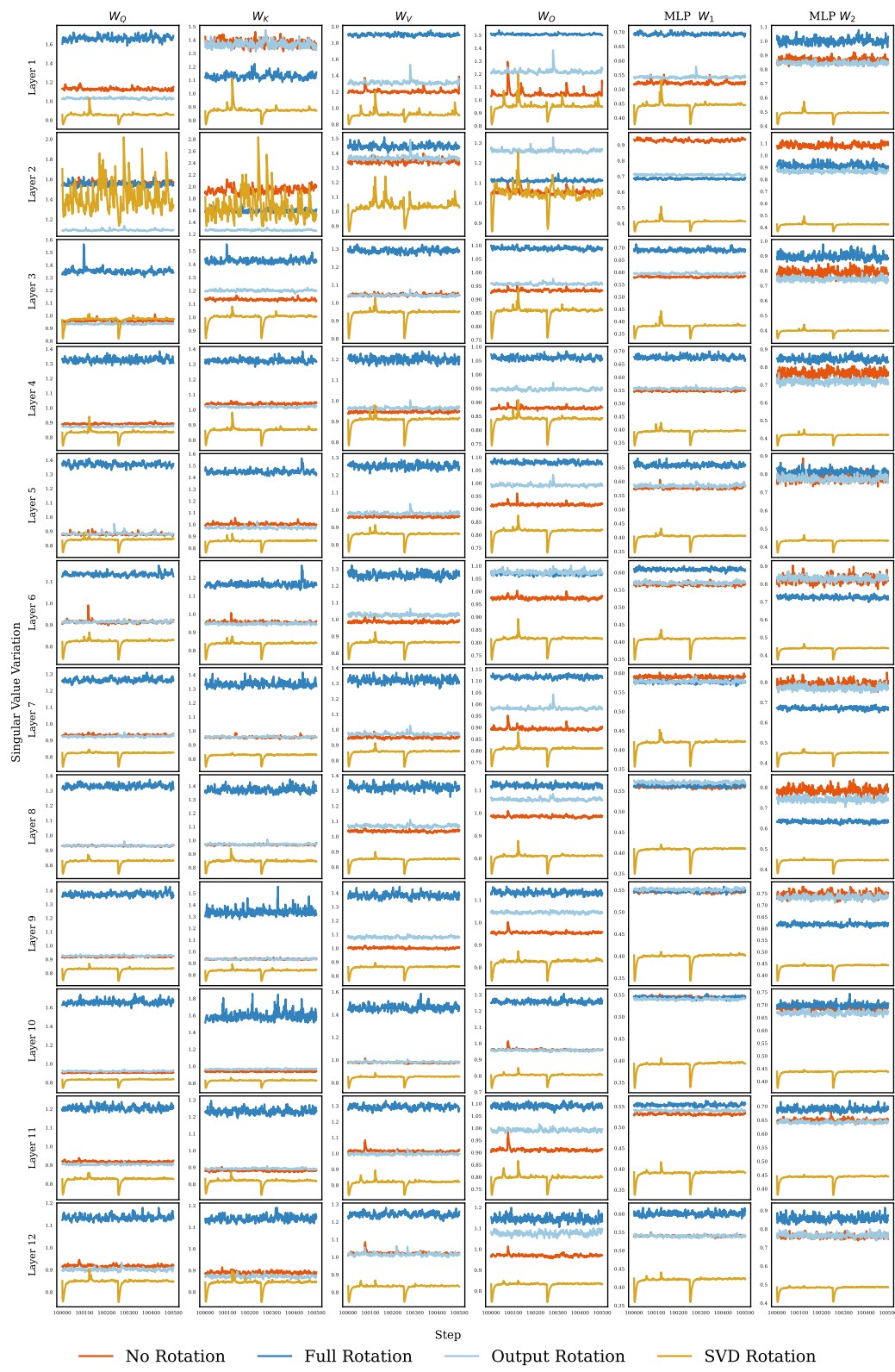

Figure 30: Singular value variation at the end of training. Except for a few layers, we see the variation hold relatively stably aside from when the SVD is recomputed.

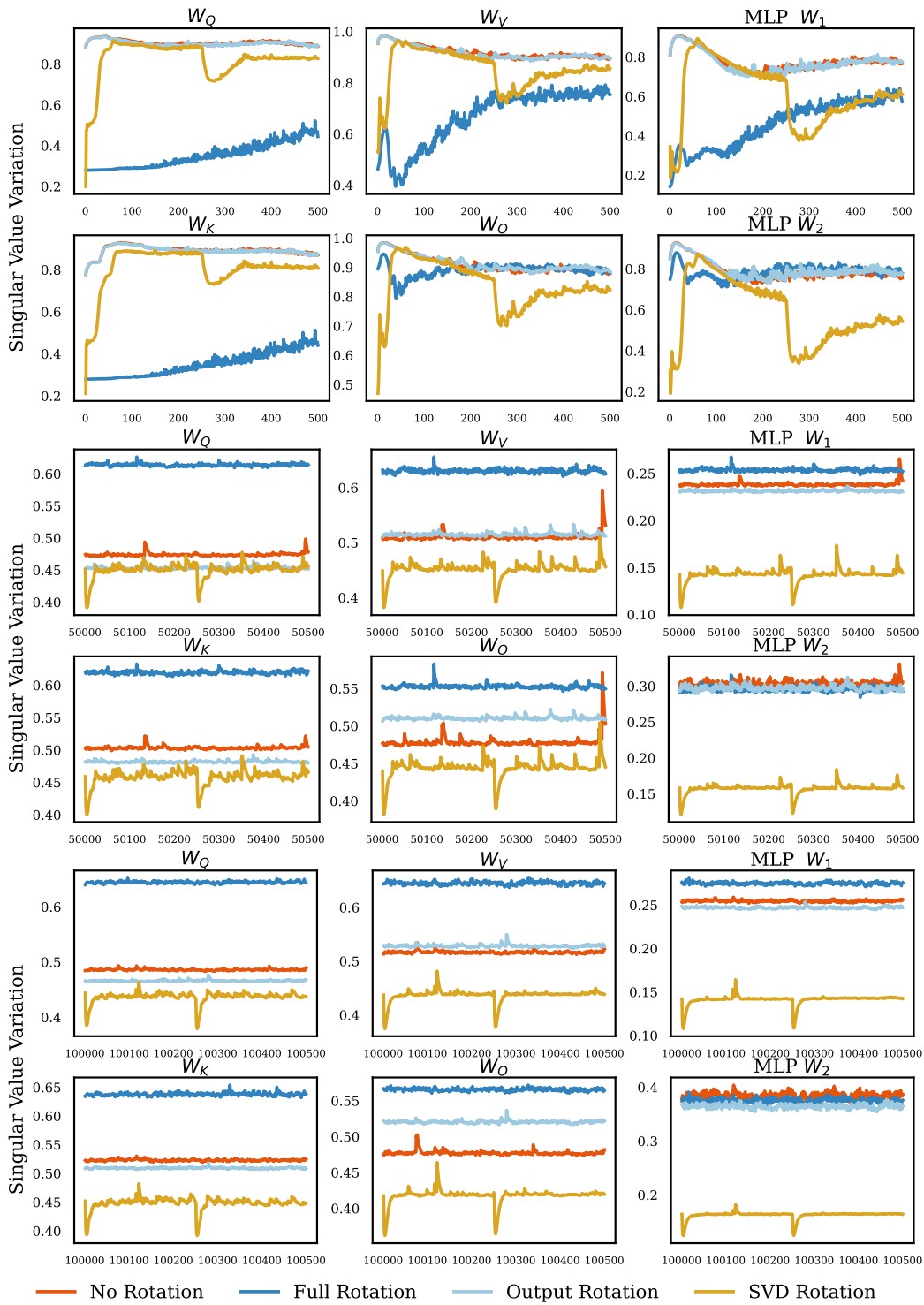

Figure 31: An alternative variation metric for singular values described in Appendix C.4 throughout training, averaged over network depth. We see similar results to the coefficient of variation.

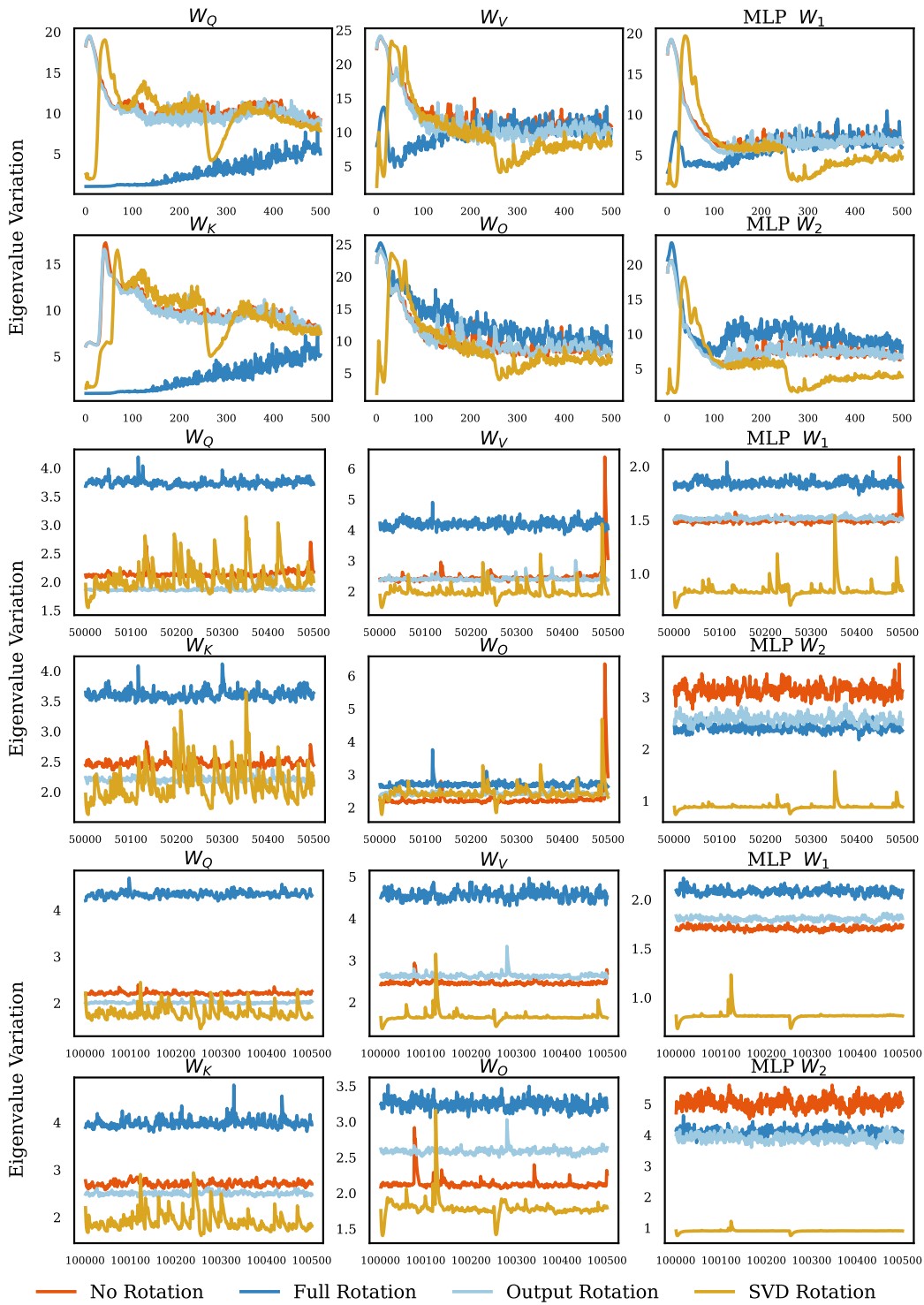

Figure 32: The coefficient of variation of the eigenvalues described in Appendix C.4 throughout training, averaged over network depth. We see similar results to the coefficient of variation.

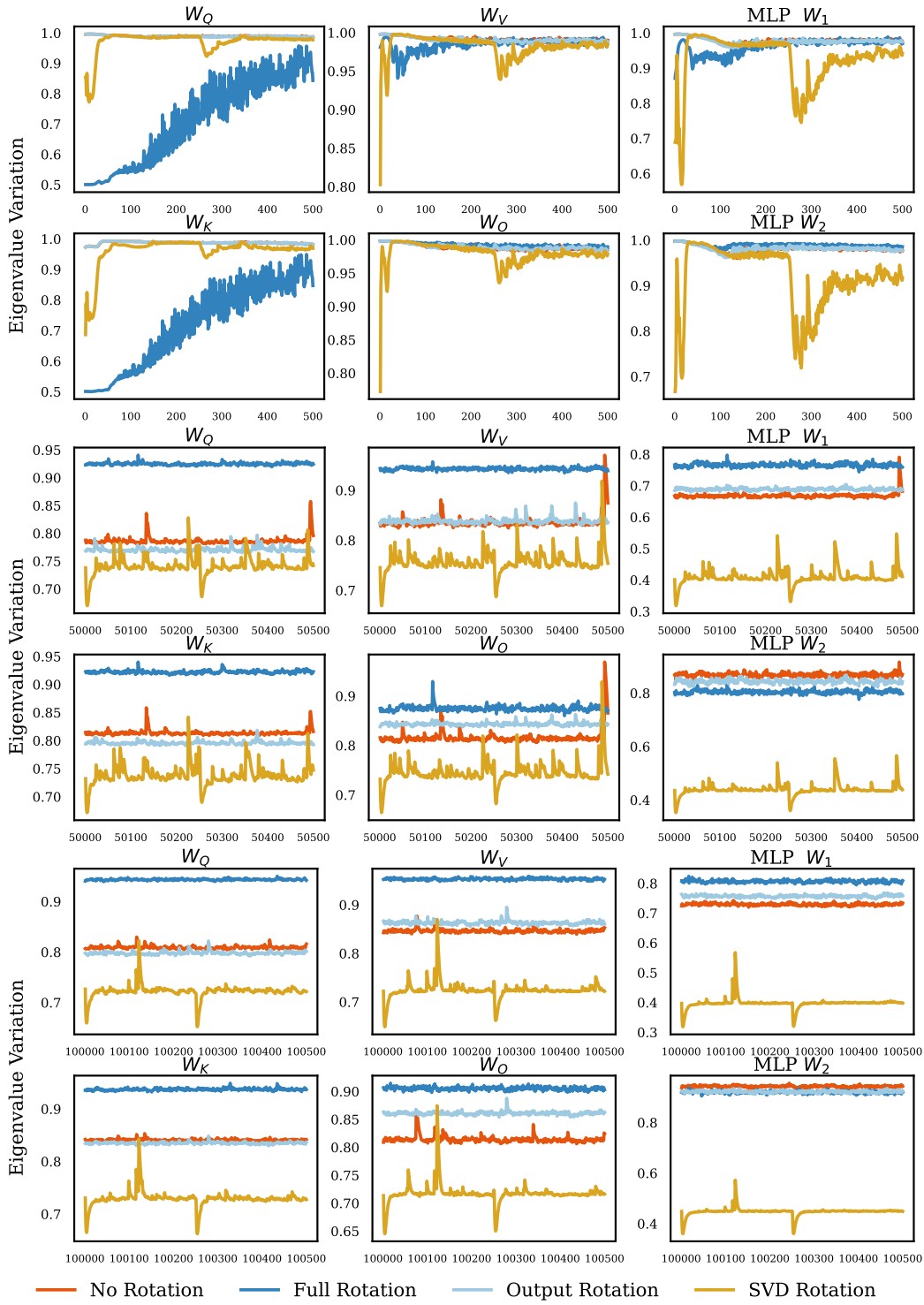

Figure 33: The variation eigenvalues of the scaled $\mathbf{A}\mathbf{A}^\top$ described in Appendix C.4 throughout training, averaged over network depth. We see similar results to the coefficient of variation.

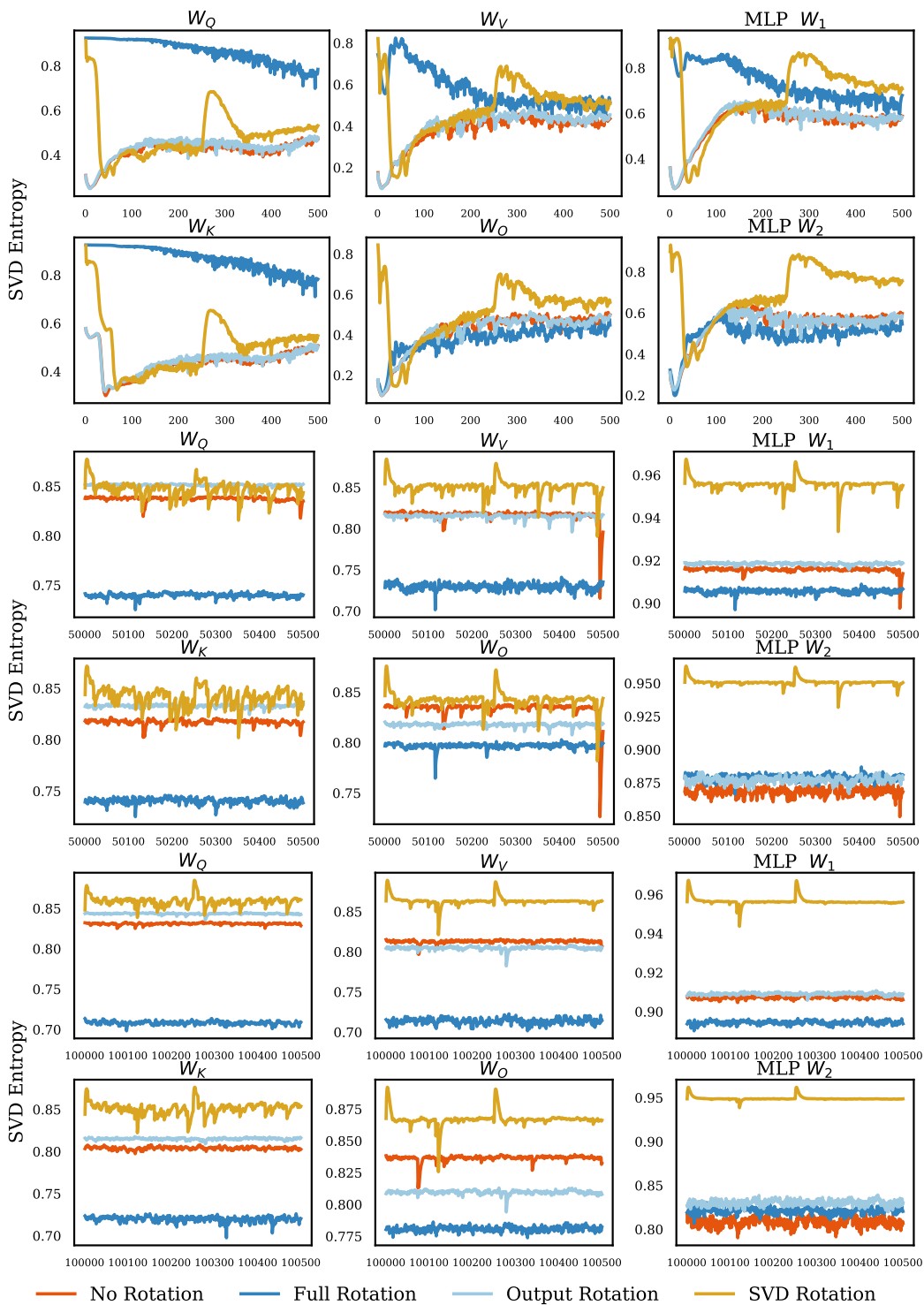

Figure 34: The SVD Entropy metric described in [Liu and Su, 2025] throughout training, averaged over network depth. We see similar results to the coefficient of variation, noting that higher is better for this metric.

**Algorithm 3** AdamW Optimization Algorithm
___

**Require:** $\alpha$: stepsize
**Require:** $\beta_1, \beta_2 \in [0, 1)$: exponential decay rates for moment estimates
**Require:** $\lambda$: weight decay coefficient
**Require:** $\epsilon$: small constant for numerical stability
**Require:** $f(\boldsymbol{\theta})$: stochastic objective function with parameters $\boldsymbol{\theta}$
1: Initialize $\boldsymbol{\theta}_0$, $\mathbf{m}_0 \leftarrow \mathbf{0}$, $\mathbf{v}_0 \leftarrow \mathbf{0}$, $t \leftarrow 0$
2: **while** $\boldsymbol{\theta}_t$ not converged **do**
3:      $t \leftarrow t + 1$
4:      $\mathbf{g}_t \leftarrow \nabla_{\boldsymbol{\theta}} f_t(\boldsymbol{\theta_{t-1}})$                $\triangleright$ Get gradients w.r.t. stochastic objective at timestep $t$
5:      $\mathbf{m}_t \leftarrow \beta_1 \mathbf{m}_{t-1} + (1 - \beta_1)\mathbf{g}_t$                $\triangleright$ Update biased first moment estimate
6:      $\mathbf{v}_t \leftarrow \beta_2 \mathbf{v}_{t-1} + (1 - \beta_2)\mathbf{g}_t^2$            $\triangleright$ Update biased second raw moment estimate
7:      $\hat{\mathbf{m}}_t \leftarrow \mathbf{m}_t/(1 - \beta_1^t)$               $\triangleright$ Compute bias-corrected first moment estimate
8:      $\hat{\mathbf{v}}_t \leftarrow \mathbf{v}_t/(1 - \beta_2^t)$          $\triangleright$ Compute bias-corrected second raw moment estimate
9:      $\boldsymbol{\theta_t} \leftarrow \boldsymbol{\theta_{t-1}} - \alpha \hat{\mathbf{m}}_t/(\sqrt{\hat{\mathbf{v}}_t} + \epsilon) - \alpha \lambda \boldsymbol{\theta_{t-1}}$             $\triangleright$ Update parameters
10: **end while**
11: **return** $\boldsymbol{\theta_t}$                              $\triangleright$ Return the final parameters

___

**Algorithm 4** AdamW Optimization Algorithm with Rotation
___

**Require:** $\alpha$: stepsize
**Require:** $\beta_1, \beta_2 \in [0, 1)$: exponential decay rates for moment estimates
**Require:** $\lambda$: weight decay coefficient
**Require:** $\epsilon$: small constant for numerical stability
**Require:** $f(\boldsymbol{\theta})$: stochastic objective function with parameters $\boldsymbol{\theta}$
1: Initialize $\boldsymbol{\theta}_0$, $\mathbf{m}_0 \leftarrow \mathbf{0}$, $\mathbf{v}_0 \leftarrow \mathbf{0}$, $t \leftarrow 0$
2: **while** $\boldsymbol{\theta}_t$ not converged **do**
3:      $t \leftarrow t + 1$
4:      $\mathbf{g}_t \leftarrow \nabla_{\boldsymbol{\theta}} f_t(\boldsymbol{\theta}_{t-1})$                $\triangleright$ Get gradients w.r.t. stochastic objective at timestep $t$
5:      $\tilde{\mathbf{g}}_t = \mathbf{R}\mathbf{g}_t$                            $\triangleright$ Apply rotation to gradients
6:      $\mathbf{m}_t \leftarrow \beta_1 \mathbf{m}_{t-1} + (1 - \beta_1)\tilde{\mathbf{g}}_t$             $\triangleright$ Update biased first moment estimate
7:      $\mathbf{v}_t \leftarrow \beta_2 \mathbf{v}_{t-1} + (1 - \beta_2)\tilde{\mathbf{g}}_t^2$         $\triangleright$ Update biased second raw moment estimate
8:      $\hat{\mathbf{m}}_t \leftarrow \mathbf{m}_t/(1 - \beta_1^t)$             $\triangleright$ Compute bias-corrected first moment estimate
9:      $\hat{\mathbf{v}}_t \leftarrow \mathbf{v}_t/(1 - \beta_2^t)$        $\triangleright$ Compute bias-corrected second raw moment estimate
10:     $\boldsymbol{\theta}_t \leftarrow \boldsymbol{\theta}_{t-1} - \alpha \mathbf{R}^{-1}(\hat{\mathbf{m}}_t/(\sqrt{\hat{\mathbf{v}}_t} + \epsilon)) - \alpha \lambda \boldsymbol{\theta}_{t-1}$      $\triangleright$ Update parameters
11: **end while**
12: **return** $\boldsymbol{\theta}_t$                             $\triangleright$ Return the final parameters

___

# E   SVD Rotations and Muon

Recently, Jordan et al. [2024] proposed Muon, an optimization algorithm for the internal linear layers of neural networks. This algorithm departed from various modifications of Adam on favour of using an "orthogonalized" matrix update. We write the Muon algorithm in Algorithm 6. We note that a simpler version of the algorithm is described in [Jordan et al., 2024], however, their implementation and the description in subsequent work is as described in Algorithm 6. We additionally make a notational switch to emphasize that Muon acts only on the matrices of internal layers, Jordan et al. [2024] recommends using a different update scheme (e.g., AdamW) on vector-valued parameters such as bias vectors or LayerNorm parameters (along with the embedding and prediction layer in transformers). We write parameters at time $t$ as $\boldsymbol{\Theta}_t$ and their gradients as $\mathbf{G}_t$.

Muon's normalization and orthogonalization step aims to drive the singular values of the update towards one. That is, if $\mathbf{B}_t'$ has the Singular Value Decomposition $\mathbf{U}\mathbf{S}\mathbf{V}^\top$, Muon aims to have the update approximate $\mathbf{U}\mathbf{V}^\top$. The approximation is computed through an iterative algorithm inspired by the Newton-Schulz method. We show that under simplifications, this is the same update recovered by SVD Rotated Adam(W). If we let $\beta_1 = \beta_2 = \epsilon = 0$ in Adam(W) and compute a single step update with rotation, the numerator and denominator terms become the singular values of the gradient matrix, which cancel, and the rotation back to the original basis leaves us with $\mathbf{U}\mathbf{V}^\top$. Mathematically,

**Algorithm 5** AdamW Optimization Algorithm with SVD Rotation

**Require:** $\alpha$: stepsize
**Require:** $\beta_1, \beta_2 \in [0, 1)$: exponential decay rates for moment estimates
**Require:** $\lambda$: weight decay coefficient
**Require:** $\epsilon$: small constant for numerical stability
**Require:** $F$: a frequency at which to update the SVD matrices
**Require:** $f(\boldsymbol{\theta})$: stochastic objective function with parameters $\boldsymbol{\theta}$
1: Initialize $\boldsymbol{\theta}_0, \mathbf{m}_0 \leftarrow \mathbf{0}, \mathbf{v}_0 \leftarrow \mathbf{0}, t \leftarrow 0$
2: **while** $\boldsymbol{\theta}_t$ not converged **do**
3:     $t \leftarrow t + 1$
4:     $\mathbf{g}_t \leftarrow \nabla_{\boldsymbol{\theta}} f_t(\boldsymbol{\theta}_{t-1})$         ▷ Get gradients w.r.t. stochastic objective at timestep $t$
5:     **if** $t \mod F = 0$ **then**
6:         $\mathbf{U}, \mathbf{S}, \mathbf{V}^{\top} \leftarrow \text{SVD}(\mathbf{g}_t)$     ▷ Calculate the Singular Value Decomposition of $\mathbf{g}_t$
7:     **end if**
8:     $\tilde{\mathbf{g}}_t = \mathbf{U}^{\top} \mathbf{g}_t \mathbf{V}$         ▷ Apply rotation to gradients
9:     $\mathbf{m}_t \leftarrow \beta_1 \mathbf{m}_{t-1} + (1 - \beta_1)\tilde{\mathbf{g}}_t$     ▷ Update biased first moment estimate
10:    $\mathbf{v}_t \leftarrow \beta_2 \mathbf{v}_{t-1} + (1 - \beta_2)\tilde{\mathbf{g}}_t{}^2$     ▷ Update biased second raw moment estimate
11:    $\hat{\mathbf{m}}_t \leftarrow \mathbf{m}_t/(1 - \beta_1^t)$     ▷ Compute bias-corrected first moment estimate
12:    $\hat{\mathbf{v}}_t \leftarrow \mathbf{v}_t/(1 - \beta_2^t)$     ▷ Compute bias-corrected second raw moment estimate
13:    $\boldsymbol{\theta}_t \leftarrow \boldsymbol{\theta}_{t-1} - \alpha \mathbf{U}(\hat{\mathbf{m}}_t/(\sqrt{\hat{\mathbf{v}}_t} + \epsilon))\mathbf{V}^{\top} - \alpha\lambda\boldsymbol{\theta}_{t-1}$     ▷ Update parameters
14: **end while**
15: **return** $\boldsymbol{\theta}_t$     ▷ Return the final parameters

---

**Algorithm 6** Muon Optimization Algorithm

**Require:** $\alpha$: stepsize
**Require:** $\mu$: Nesterov momentum parameter
**Require:** $\lambda$: weight decay coefficient
**Require:** $\epsilon$: small constant for numerical stability
**Require:** $f(\boldsymbol{\Theta})$: stochastic objective function with parameters $\boldsymbol{\Theta}$
1: Initialize $\boldsymbol{\Theta}_0, \mathbf{B}_0 \leftarrow \mathbf{0}, t \leftarrow 0$
2: **while** $\boldsymbol{\Theta}_t$ not converged **do**
3:     $t \leftarrow t + 1$
4:     $\mathbf{G}_t \leftarrow \nabla_{\boldsymbol{\Theta}} f_t(\boldsymbol{\Theta}_{t-1})$     ▷ Get gradients w.r.t. stochastic objective at timestep $t$
5:     $\mathbf{B}_t \leftarrow \mu\mathbf{B}_{t-1} + \mathbf{G}_t$     ▷ Update momentum buffer
6:     $\mathbf{B}_t' \leftarrow \mu\mathbf{B}_t + \mathbf{G}_t$     ▷ Apply Nesterov Momentum
7:     $\tilde{\mathbf{B}}_t \leftarrow \mathbf{B}_t'/(\|\mathbf{B}_t'\|_F + \epsilon)$     ▷ Normalize the update
8:     $\mathbf{O}_t \leftarrow \text{NewtonSchulz5}(\tilde{\mathbf{B}}_t)$     ▷ Approximately orthogonalize the update
9:     $\boldsymbol{\Theta}_t \leftarrow \boldsymbol{\Theta}_{t-1} - \alpha\mathbf{O}_t - \alpha\lambda\boldsymbol{\Theta}_{t-1}$     ▷ Update parameters
10: **end while**
11: **return** $\boldsymbol{\Theta}_t$     ▷ Return the final parameters

---

let $\mathbf{USV}^{\top}$ be the SVD of gradient $\mathbf{G}$. Then, the SVD Rotated Adam(W) numerator becomes $\mathbf{M} = \mathbf{U}^{\top}\mathbf{GV} = \mathbf{U}^{\top}\mathbf{USV}^{\top}\mathbf{V} = \mathbf{S}$. Similarly, the denominator is the entry-wise square of the rotated gradient, which leaves us with $\mathbf{V} = \mathbf{S}^2$. Then, computing the update $\mathbf{M}/\sqrt{\mathbf{V}}$ leaves us with $\mathbf{S}/\mathbf{S}$ which is the identity. The final step in the algorithm is to rotate this update back, which is done by $\mathbf{UIV}^{\top}$, leaving us with the Muon update.

While this setting is an oversimplification (the momentum parameters are often crucial for performance), it does offer an interesting connection between Adam's update in a different basis and more recent algorithms like Muon or Shampoo [Gupta et al., 2018].

## F Common Assumptions in First-Order Optimization Theory

We present a non-exhaustive summary of common assumptions used in theoretical works for first-order optimization, see Table 2. For each assumption, we indicate whether it is rotation invariant.

| Assumption | Rotation-Invariant |
|---|:---:|
| (Strong-) Convexity | ✓ |
| Polyak-Lojasiewicz [Polyak, 1963] | ✓ |
| Star-(Strong)-Convexity [Guille-Escuret et al., 2020] | ✓ |
| Quadratic Growth [Goujaud et al., 2022] | ✓ |
| L-Smoothness ($L_2$ norm) [Défossez et al., 2022, Zhou et al., 2024] | ✓ |
| Gradient Growth Condition [Zhang et al., 2022] | ✓ |
| Bounded Expected Gradient Squared Norm [Zou et al., 2019] | ✓ |
| $(L_0, L_1)$-Smoothness [Li et al., 2023b] | ✓ |
| Restricted Secant Inequality [Guille-Escuret et al., 2022] | ✓ |
| Error Bound [Luo and Tseng, 1993, Guille-Escuret et al., 2024] | ✓ |
| L-smoothness ($L_\infty$ norm)[Guo et al., 2022] | ✗ |
| Coordinate-wise $(L_0, L_1)$-Smoothness [Crawshaw et al., 2022] | ✗ |
| Coordinate-wise "Affine" Variance Noise [Li and Lin, 2024] | ✗ |
| Bounded Gradient ($L_\infty$) [Reddi et al., 2018] | ✗ |

Table 2: Common assumptions involved in first-order optimization algorithm, indicating whether they are rotation-invariant. Rotation-dependent assumptions are comparatively rare in the literature.

