# OpenReview forum: "Understanding Adam Requires Better Rotation Dependent Assumptions"
_NeurIPS.cc/2025/Conference — NeurIPS 2025 poster_

### Official Review · Reviewer_Jnyt · 2025-06-28

**Clarity:** 2
**Significance:** 1
**Originality:** 2
**Rating:** 2
**Confidence:** 4

**Summary:**

In this paper, the authors empirically compare the training loss of Adam when the parameters are conducted with random rotations with respect to different spaces, and report a degradation of performance when rotations are involved. Based on this observation, the authors design some experiments to examine some assumptions that are commonly adopted in recent theoretical studies regarding Adam. They found that these assumptions can not explain the degradation caused by random rotations. In contrast, they propose that the scaled(semi-) orthogonality, alignment with the degradation caused by random rotations.

**Questions:**

Could the authors explain why they introduce two rotation matrices $\mathbf{R}$ and $\mathbf{R}'$ before line 192 and line 220. I do not find any discussions regarding why two matrices are required. In addition, there might be a typo in the definition of $\mathbf r_i$. The current formula might indicate the $i$-th column instead of the $i$-th row.

**Ethical Concerns:**

["NO or VERY MINOR ethics concerns only"]

**Limitations:**

See weaknesses.

**Paper Formatting Concerns:**

No.

**Quality:**

2

**Strengths And Weaknesses:**

**Strengths:**

Based on the recent success of Muon, it is interesting to investigate optimizers from the perspective of the spectral distribution and the orthogonality of layers. The empirical observation that the training with Adam would be correlated with scaled(semi-) orthogonality proposed in this paper is somewhat insightful.

**Weakness:**
- I do not feel that one of the main conclusions that the rotation degrades the performance of Adam is convincing enough.
    - Compared to the training loss, people concern more about the generalization of the models obtained from the training. However, as demonstrated in Figure 13 and 14, there are no significant distinctions among different curves representing the validation loss and the validation accuracy.
    - Even when considering only the training loss, the current experimental results(Figure 1) are insufficient to fully support the claim. **It is evident that after a brief initial phase, all training loss curves exhibit nearly identical patterns, appearing almost parallel.** This indicates that the loss reduction rate is consistent across conditions, with or without rotation. Furthermore, the losses do not appear to have converged after 100 epochs. Should training be extended over a longer duration, the initial stage's instability would likely have a diminished impact on the entire training required, thereby dramatically reducing the claimed ratio of training iteration slowdown mentioned in the caption of Figure 1. In addition, I am also concerned about the visual interpretation of Figure 1(a) due to the chosen scale of the y-axis. Given the very narrow range of the y-axis (approximately 2.87 to 3.16), differences that appear visually distinct are, in absolute terms, less than 0.02. This narrow scaling potentially exaggerates what might otherwise be considered negligible differences in training loss, making them appear more substantial than they are.

- The foundational premise of discussions regarding Adam's assumptions, specifically that rotation degrades its performance, appears unconvincing. Consequently, this casts doubt on the validity of all conclusions and discussions presented in Section 4. However, even overlooking this fundamental issue, the basic logical framework of Section 4 itself lacks persuasiveness. Given my limited familiarity with the third assumption, my focus will primarily be on the first and second assumptions. Based on my understanding of theoretical studies concerning Adam (e.g., [1]), these assumptions are typically employed to provide theoretical convergence guarantees, reflecting worst-case behavior rather than an exact convergence rate. Furthermore, establishing an exact convergence rate for Adam remains exceedingly challenging; to my knowledge, no theoretical studies currently present such results, even under specific assumptions regarding data distributions. Therefore, to challenge certain theoretical assumptions solely based on observed loss convergence seems arbitrary, especially considering that theoretical results commonly accommodate discrepancies up to a constant factor, a phenomenon that exactly appears to be precisely reflected in your Figure 1.

- While the concept of scaled (semi-)orthogonality proposed in this paper offers some insight, its rigorous translation into a beneficial assumption for establishing a robust theoretical framework remains unclear. More importantly, the methodology for constructing such a theoretical framework based on this concept is not articulated. Therefore, this point, in its current presentation, may not suffice as the major contribution of the paper.

[1] Shuo Xie, Mohamad Amin Mohamadi, and Zhiyuan Li. Adam exploits $\ell_\infty$-geometry of
 loss landscape via coordinate-wise adaptivity. In ICLR, 2025

---

> ### Author Rebuttal · Authors · 2025-07-31
>
> We thank the referee for their consideration of our work. We would like to bring up some crucial points in response to the referee’s concerns:
>
> ---
>
> ### **W1: Concerns about rotations degrading Adam’s performance:**
>
> ### W1.1 Impact on validation:
> We thank the reviewer for making us realize our mistake with the figures: we made an error in the labels used for Figure 13, which led to the wrong Figures being loaded. Although Figure 13 (left) is correct, Figure 13 (mid) representing the validation loss is wrong, and Figure 13 (right) is just a duplicate of Figure 14 (right), which is for ResNet50, not ViT.
>
> We sincerely apologize for that mistake, which wrongly indicates that the differences in training loss were not extended to validation. Though openreview does not allow sharing figures, we will fix this error. The final validation accuracy for ViT is about 69% for original Adam, against 64% for Adam with global rotations. The original Adam reaches 64% validation accuracy in only 20/100 epochs, 5 times faster than with global rotations, an even wider gap than for training loss.
>
> To summarize the validation accuracy after fixing the error in Figure 13:
> * For GPT, we train for much less than a full epoch, and training samples are never seen multiple times. Therefore, the training loss has the same distribution as the validation loss and shows a clear difference in performance.
> * For ViT, we observe a significant difference in training and validation loss, and an even greater difference in validation accuracy.
> * For ResNet, we see no significant difference with rotations. This supports our claim that Adam’s advantages over SGD are tied to Adam leveraging the properties of the canonical basis. Indeed, Adam is not superior to SGD for training ResNet, and precisely does not seem to be affected by the basis in this setting. Adam is sensitive to the basis in the settings where it performs better than SGD (ViT and GPT).
>
> ### W1.2 Parallel training curves, non-convergence at the end, small loss delta:
> We provide evidence that these concerns raised by the referee are natural artifacts of empirical training of transformer models at this scale, well documented in this type of setting, and do not contradict a large degradation of performance under global rotations: In fact, they make optimizer performances appear visually smaller than they are.
>
> The training curves reported in the original blog post (Muon: an Optimizer for Hidden Layers in Neural Networks) by Keller Jordan (Figures 1 and 2) display the same ‘parallel curves’ that the referee is concerned about. Similarly, they also seem to have been stopped before converging, which, as explained below, is only in appearance. More strikingly, we invite the referee to consult Figure 2 of [2], which shows almost perfectly parallel lines, with a final difference between Muon and AdamW training losses of around 0.02. Given the clear success of Muon and the striking similarities with our experiment results, we hope the referee will reconsider the significance of these observations. We detail below the reasons for the artifacts noted by the referee.
>
> The apparent non-convergence after 100k iterations (the x-axis is in thousands of iterations, our training does not complete a single epoch of the dataset) is a typical artifact of learning rate decay in LLM training. A large portion of the loss improvement is driven by the decay of the LR (allowing more fine-grained representation), but the LR reaches 0 by the end of training. There is no way to pursue training any further, and keeping the LR constant before it reaches 0 would lead to a near-stagnation of the loss. A striking example can be seen in Figure 10 of [1]: although the training curves appear not to have converged at 200k steps, increasing the training duration to 400k steps leads to a near identical final training loss. The referee can also refer to Karpathy’s NanoGPT Github repository, which our experiments are based on, as it provides some insightful comparison, including a training curve of GPT2 (almost identical setting to ours – except we improved hyperparameter tuning) trained over 400k steps. The range of the y-axis in this baseline training is the same as ours, and the referee will notice that the final loss is 2.905, which is worse than ours, even though we trained for ‘only’ 100k steps. As such, our final training loss after 100k steps indicates a strong final loss that would be hard to significantly improve simply by training longer. The same phenomenon also leads to training curves having near-parallel lines due to the similar impact of the LR annealing.
>
> Additionally, evaluating the impact of a fixed shift in training loss can be misleading for LLMs, due to cross entropy and the naturally high noise in the data. For instance, from iterations 120k to 400k, Kartpathy’s GPT2 goes from loss ~3.0 to ~2.9 over 280k steps, and even this shift is largely driven by LR annealing. A shift of .02 at the same point of the LR schedule is thus significant. We refer to Figure 10 of [1] in which the optimizer introduced by the authors as superior to Adam achieves a .01 loss improvement (in an almost identical setting!) over AdamW, over 200k steps. We observe a degradation twice as large in half the training steps compared to the sizable gain claimed by the authors.
>
> As a side note, Xie et al. [3] concurrently made the same observation as us: “Adam on the original loss converges the fastest as expected. But convergence of Adam on a rotated loss is much slower [...]”
>
> [1] Sophia: A Scalable Stochastic Second Order Optimizer For Language Model Pretraining, H. Liu et al., 2024, arXiv 2305.14342 v4\
> [2] Muon is Scalable for LLM Training, J. Liu et al., 2025. arXiv 2502.16982 v1\
> [3] Shuo Xie, Mohamad Amin Mohamadi, and Zhiyuan Li. Adam exploits $L_\infty$-geometry of loss landscape via coordinate-wise adaptivity. In ICLR, 2025
>
>
> ### **W2: Logical framework of Section 4:**
>
> The referee brings an important point. The goal of Section 4 is not to reject theoretical assumptions or their usefulness. These assumptions have their purpose in theoretical frameworks. The goal of Section 4 is instead to assess whether existing assumptions that are rotation-dependent (and thus capable of discriminating bases) can explain the variations in performance of Adam observed across bases. The minimal requirement is that when varying only the basis, a basis with the worst property should lead to the worst performances and vice-versa.
>
> Interestingly, the paper cited by the referee, Xie et al. [3], precisely uses the same reasoning as us to support the validity of the $(1,1)$ norm, and, by proxy, of the $L_\infty$ smoothness that it lower bounds. See Table 1 of [3], which they use to justify the effectiveness of their quantity by observing that its values align with Adam’s performance in the canonical and randomly rotated bases. While we also observe this alignment for the canonical basis and global rotation, it breaks under a type of basis that [3] had not considered, obtained via output-wise rotations. For the same reason, they supported the effectiveness of their assumption by observing this alignment; our observation that this alignment does not generalize contradicts its effectiveness.
>
> Very importantly, this does not imply that metrics such as $L_\infty$ smoothness are not useful. It simply invalidates its ability to capture the properties of the basis that Adam leverages to achieve its advantages over SGD.
>
> ### **W3: Scaled (semi-)orthogonality as a beneficial theoretical assumption:**
>
> We agree with the referee that this aspect should be explicitly discussed in the main paper. We discuss the connection with Muon, but do not explicitly remind or point to Muon’s theoretical justification, which directly justifies scaled orthogonality as a powerful theoretical tool for studying the optimization of neural networks. We will update our submission to explicitly refer to the theoretical origins of Muon (and therefore the justification for scaled orthogonality), and remind them in an Appendix section. For a detailed overview of the theoretical justifications of Muon, we refer to Jeremy Bernstein’s blog post “Deriving the Muon optimizer”. To the best of our knowledge, there is no published paper explicitly on this topic, but we will cite the blog post accordingly.
>
> For a summary of the importance of scaled orthogonality in theoretical frameworks, we refer to our reply to [reviewer z9Dc Q2](https://openreview.net/forum?id=KD4wgunbhO&noteId=rNhxJeyPDz), who raised a similar concern. We will also add this to the appendix and refer to it in the main paper.
>
> ### **Q1: Difference between $R$ and $R’$:**
> The reason for the distinction is to be able to distinguish the basis that is used for measuring a quantity, and the basis that is used during training. For instance, if $R’=$None and $R$=Global, we would use the checkpoints from a normal Adam training run, and measure, such as the $(1,1)$-norm of the Hessian in a globally rotated basis. The initial purpose was to separate the impact of training on a basis from the impact of measuring on a basis. However, we found that the significance of the results obtained with $R!=R’$ was not well justified. Reporting results that have no clear interpretation or meaning was confusing, so we removed them. We thank the referee for this remark, and we will remove this separated notation by a single dependency on R since we only report with $R=R’$.
>
> ### Typo in $r_i$:
>
> The reviewer is correct, and we will fix this for clarity, though amusingly, with the Hessian being symmetric, its rows and columns are the same anyway (up to transpose).
>
> ---
> We thank the reviewer for helping us improve the paper. We kindly ask the reviewers to consider increasing their scores if they find our responses satisfactory. We remain available and happy to address any further questions during the discussion period.

---

> > ### Comment · Reviewer_Jnyt · 2025-08-05
> >
> > Thanks for the response. I believe the authors may have conflated some fundamental concepts—or at the very least, they have misunderstood my original question. Let's make it clear in the following.
> >
> > The paper's central claim asserts that rotation deteriorates Adam's performance, primarily characterized as reducing its convergence rate. However, as I explicitly noted in my original response, **I can only feel that all the training loss curves decay at almost the same speed since they exhibit almost parallel patterns.** I find nearly all of the authors' rebuttal content regarding this point either irrelevant or unconvincing. Why should I care about the parallel training loss curves for training Muon and AdamW? Are those authors also claiming that the parallel curve indicates significantly distinct training loss convergence? Please note that I'm not disputing the existence of parallel training loss curves. My fundamental question remains: how can parallel curves, which inherently suggest similar convergence rates, be construed as indicating markedly distinct convergence properties?
> >
> > I must reiterate that comparing final loss values after a predetermined number of iterations is fundamentally uninformative, since such discrepancies could entirely originate from variations in initialization and stochastic fluctuations during the initial training phase - precisely what your figures illustrate. In addition, I also want to point out that different optimizers (here considering rotated Adam variants as distinct optimizers) may naturally converge to different local minima with varying loss values, which serves as another reason why such a comparison is meaningless. That's why in the literature of non-convex optimization, people usually study the convergence of the gradient, instead of the objective function itself. (I'm not urging the authors to add experiments reporting the gradient.)
> >
> > Given my discussion above, I do not feel your claims regarding the training loss convergence are correct. I also do not feel that a difference on the scale of 0.02 presents any essential distinctions. I haven't read Muon's original blogs, but I guess its success does not originate from just achieving a slightly lower training/validation loss by 0.02; otherwise, any difference larger than 0.02 can be interpreted as 'significant distinction', following the authors' logic. Since you are claiming that your training results of GPT2 models directly reflect the results on the validation set. It is hard for me to see any convincing evidence from this set of experiments to support the central claims in terms of the generalization. While I acknowledge the 5\% accuracy improvement in your ViT experiments, this observation warrants more careful interpretation. If Adam's ability to utilize basis functions varies significantly across different architectures (ViT, GPT2, ResNet), can we truly attribute these effects to Adam's inherent properties rather than model-specific characteristics?
> >
> > In fact, the claimed conclusion that Adam's performance exhibits rotation sensitivity comparable to $\ell_\infty$ geometry is extremely surprising to me (though I remain skeptical of its validity). Beyond [1], numerous studies demonstrated connections between Adam and $\ell_\infty$  geometry. Notably, recent works [2, 3] rigorously demonstrate that the convergence point of Adam exhibits properties in terms of $\ell_\infty$ norm. And a following work [4] shows that Muons exhibit these types of properties, but in terms of matrix-spectral norm, which is seemingly more connected with the rotations and basis functions. At least from my perspective, I hope to obtain some convincing explanations or interesting insights from your empirical observations, instead of these observations themselves.
> >
> > [1]. Shuo Xie, Mohamad Amin Mohamadi, and Zhiyuan Li. Adam exploits $\ell_\infty$-geometry of loss landscape via coordinate-wise adaptivity. ICLR
> >
> > [2]. Shuo Xie, and Zhiyuan Li. Implicit Bias of AdamW: $\ell_\infty$-Norm Constrained Optimization. ICML
> >
> > [3]. Chenyang Zhang, Difan Zou, and YuanCao. The implicit bias of Adam on separable data. NeurIPS
> >
> > [4]. Chen Fan, Mark Schmidt, and Christos Thrampoulidis. Implicit Bias of Spectral Descent and Muon on Multiclass Separable Data. HiLD.

---

> > > ### Author Response · Authors · 2025-08-06
> > >
> > > Thank you for taking the time to engage in discussion.
> > >
> > > We hope the referee will again take the time of considering the following points, which are rarely considered in purely theoretical works but are very standard in real-world training:
> > >
> > > >I can only feel that all the training loss curves decay at almost the same speed since they exhibit almost parallel patterns
> > >
> > > >how can parallel curves, which inherently suggest similar convergence rates, be construed as indicating markedly distinct convergence properties?
> > >
> > > As mentioned in our rebuttal, this is a standard artifact when using cosine learning rate decay in stochastic training (and without which our experiments would not be realistic). In theoretical frameworks studying SGD, under assumptions of strong convexity, smoothness, and bounded gradient variance, it is standard to model the training with:
> > >
> > > $\mathbb{E}[||x_{n+1}-x^*||^2] \leq$
> > >
> > > $(1-c \lambda_{n+1}) \mathbb{E}[||x_n-x^*||^2]+\lambda_{n+1}^2 \sigma^2 $
> > >
> > > where $\sigma^2$ is the variance bound, $\lambda_{n+1}$ is the learning rate at step $n+1$, and $c$ is a constant (see e.g., [1]). The first term comes from learning from the signal, and the second from the noise. A similar phenomenon will affect all optimizers. But even ignoring that, **the above model will lead to similar observations, even when using a different constant c (i.e., a different convergence rate).** We provide the following matplotlib snippet of the above model with two different rates to show that the curves will have the same *apparent* parallelism due to cosine LR decay at the end of training:
> > >
> > > ```import numpy as np
> > > import matplotlib.pyplot as plt
> > >
> > > def calculate_y_iterative(n_steps, rate, noise):
> > >     n_values = np.arange(n_steps + 1)
> > >     z_values = np.zeros(n_steps + 1, dtype=float)
> > >     z_values[0] = 1.0
> > >     for n in range(1, n_steps + 1):
> > >         lr = (1 + np.cos(np.pi * n / n_steps))
> > >         z_values[n] = (1 - rate * lr) * z_values[n-1] + (noise * lr)**2
> > >     return n_values, z_values
> > >
> > > n, z = calculate_y_iterative(2000, 0.0005, 0.006)
> > > _, z2 = calculate_y_iterative(2000, 0.00055, 0.006)
> > > plt.figure(figsize=(10, 6))
> > > plt.plot(n, z, z2)
> > > plt.grid(True)
> > > plt.legend()
> > > plt.show()
> > > ```
> > > These two curves have very different rate constant c, but they appear parallel at the end nonetheless. The referee's reasoning is technically correct: the actual convergence of the loss at the end is dominated by the lr annealing, meaning the variations will be similar across optimizers. But this nearly independent additive term at the end of training is not relevant to the final performance, and hides the differences in the convergence rate of the relevant term with the fixed noise reduction from lr annealing.
> > >
> > > **All reasonable optimizers in realistic settings for LM training will display this behavior**. Disqualifying performance variations based on *apparent* parallelism of training curves is a standard that would imply that all optimizers are equivalent for training LM in realistic settings. **This simplifies the theoretical understanding of Adam's advantage by simply denying its existence.**
> > >
> > > [1] Non-Asymptotic Analysis of Stochastic Approximation Algorithms for Machine Learning, F. bach and E. Moulines, 2011.
> > >
> > > >I must reiterate that comparing final loss values after a predetermined number of iterations is fundamentally uninformative
> > >
> > > We agree, though for a different reason: the scaling of training loss in LM training is extremely dependent on the data distribution, and in particular its entropy, which lower bounds the best loss achievable (by the ground truth distribution). As the loss approaches its lower bound L*, improving it becomes more and more difficult. E.g., one can go from L* + 5 to L* + 4 simply by memorizing a few grammatical rules, but going from L* + 0.04 to L* + 0.02 might require to learn 100s of millions of niche facts in the extreme of the distribution's tail. Thus, a difference of e.g., 0.02, should not be considered *small* out of context.
> > >
> > > We would like to remind the referee that they brought up the absolute difference in loss value as a concern. Our comparison to other differences of loss values, with context (same architecture, data distribution, etc), was in response to that.
> > >
> > > > such discrepancies could entirely originate from variations in initialization and stochastic fluctuations during the initial training phase - precisely what your figures illustrate
> > >
> > > Our figures do not illustrate this. The variations related to stochastic minibatch sampling and stochastic initialization are typically significantly below these differences in this type of setting. We would not observe such consistent results across settings and even data modalities if it was not the case -- nonetheless we confirmed negligible variations when varying stochastic seed. As explained above, a same $\delta$ of loss is a lot more significant at low loss than high loss. So maintaining a gap in loss as it goes down means increasing the gap in model quality.

---

> > > > ### Author Response · Authors · 2025-08-06
> > > >
> > > > > That's why in the literature of non-convex optimization, people usually study the convergence of the gradient, instead of the objective function itself
> > > >
> > > > In realistic stochastic training scenarios (at least for LM with transformers), not only the gradients do not converge to 0, but they generally do not even go down in norm throughout training. They are so heavily dominated by noise, that there is no perceptible decrease. **A fundamental goal of our paper is to build theoretical frameworks that are applicable and relevant to real-world practice. We strongly believe this is impossible if we choose to discard all the properties of real-world training.**
> > > >
> > > > > I haven't read Muon's original blogs, but I guess its success does not originate from just achieving a slightly lower training/validation loss by 0.02; otherwise, any difference larger than 0.02 can be interpreted as 'significant distinction', following the authors' logic.
> > > >
> > > > Yes, it is the scale of loss improvement achieved by Muon in an almost identical settings to ours (because their authors used the same codebase as ours, nanoGPT). Again, any difference of loss must be interpreted in the context of the data distribution, and even the choice of tokenizer (in this case we used the same data and tokenizer as Muon). Looking at an absolute value like this provides zero information. If the loss was at L* + 0.01, this difference would not only be huge, but literally impossible to achieve in expectation.
> > > >
> > > > >If Adam's ability to utilize basis functions varies significantly across different architectures (ViT, GPT2, ResNet), can we truly attribute these effects to Adam's inherent properties rather than model-specific characteristics?
> > > >
> > > > As we remark, the setting in which Adam's performance is not significantly affected by the basis (ResNet) is precisely the setting in which Adam is not superior to SGD. Here, *model-specific characteristics* directly translate into properties of the loss landscapes. It is likely that the canonical basis doesn't have advantageous properties for the loss landscape of ResNets, but does for transformers. **We attribute these effects to Adam's inherent capability of leveraging properties of the canonical basis of its objective loss landscape, which are directly affected by the characteristics of the model**. Thus, it is not mutually exclusive: the model's characteristics induce the properties, Adam's properties take advantage of these properties -- if favorable.
> > > >
> > > > As an example, nesterov momentum accelerate GD's convergence on quadratic. But if the quadratic is perfectly conditioned (e.g. $f(x)=||x||_2^2$), it does not help. Does that mean that we should attribute the performance improvement of momentum on badly conditioned quadratics to model-specific characteristics (bad conditioning)? Or to Nesterov momentum's inherent properties? Obviously, the answer is both: the advantage should be attributed to Nesterov's momentum ability to alleviate bad conditioning when the model is such that the loss has bad conditioning.
> > > >
> > > > >the claimed conclusion that Adam's performance exhibits rotation sensitivity comparable to $l_\infty$
> > > >  geometry
> > > >
> > > > We are not sure of the referee's meaning here, nor what they find surprising given the cited literature? Our empirical findings show that the $l_\infty$ smoothness, approximated by the $(1,1)$ norm as in [Xie et al, 2025], does not align with Adam's performance when considering a wider range of bases, beyond global random rotation and canonical basis. On the contrary, our findings suggest that it is the orthogonality of the update, as suggested in Muon's theoretical justifications, that seems to be aligned with Adam's performances. Moreover, [Chen et al, 2025] which the referee mentioned only considers linear classification, while we consider deep transformer architectures, a drastic difference in setting.
> > > >
> > > > **General remark:** while we fully acknowledge the merits of optimization theory for ML conducted in controlled, simplified settings, such works have struggled to have impact and relevance on explaining and guiding real-world optimization of deep neural nets. **It seems the referee is concerned that our experiments and insights do not coincide with typical theoretical settings and predictions, in which gradients converge to zero, there is no (or little) noise in gradient sampling, etc. Our goal is to build theoretical frameworks closer to realistic settings, rather than bring empirical experiments closer to existing theory.** We hope that the referee will consider that our angle is of high relevance to at least a significant fraction of the NeurIPS community -- even though it is of lesser interest to them personally.

---

> > > > > ### Comment · Reviewer_Jnyt · 2025-08-06
> > > > >
> > > > > I respectfully disagree with the authors' conclusion that the parallel patterns in training loss curves are caused by cosine learning rate decay. After carefully reproducing the authors' code, I must point out: The so-called "parallel patterns" (though I question whether they are truly parallel) appear to be simply an artifact of the specific constant values chosen in their experiments. When I repeated the experiments with a fixed learning rate of 1, I observed nearly identical "parallel patterns". Moreover, while the curves may appear parallel at first glance, closer examination(or you can direct report) reveals that the offset between them strictly increases over time. This holds true for both cosine decay and fixed learning rate scenarios, indicating that for both cases, these curves are inherently not parallel. But what about your curves in Figure 1? Could you please also report the offset of these curves at 20 epochs, 40 epochs, etc.
> > > > >
> > > > > Additionally, if the learning rate's decay toward zero were truly responsible for these patterns (as implied by the authors' cosine LR formula), we should expect the curves' slopes to approach zero. Yet Figure 1(b) clearly shows straight-line trajectories with persistent positive slopes, except for a brief initial phase. This discrepancy raises serious questions: Is the learning rate in these experiments actually decaying to zero as claimed? Could the authors clarify this apparent contradiction between their claimed vanishing learning rates and empirical results? If your cherry-picked figures look unconvincing to me, how should I expect the precision and validation of your work?
> > > > >
> > > > > Furthermore, I would like to re-emphasize my point about non-convex optimization, the possibility that there might exist multiple local minima, and these minima themselves have different values. Given such a possibility, it might be meaningless to directly compare the training losses. While I do not understand what $L^\*$ indicates under this context, I would like to point out, if an optimizer can achieve a loss near $L^\*$, while another can only gradually achieve $L^\* + 0.01$, then I can only interpret $L^\* + 0.01$ as the convergence point for your second optimizer, and your loss convergence study should focus on its distinctions between $L^\* + 0.01$, the inherent convergence point. If you can ensure that your objective function possesses only one global minimum, then I can only conclude that the second optimizer can never converge. I understand there are some gaps between theory and applications, but given these inconsistencies, how can your observations be rigorously formulated to benefit the theoretical community? Please note that your statement of the purpose of this work is to "build theoretical frameworks".
> > > > >
> > > > > Please do not construe my questions about training loss convergence as suggesting all optimizers are equivalent for LLM training. Different optimizers inherently possess distinct properties that drive convergence toward different local minima - minima which may demonstrate varying performance despite having similar training loss values.
> > > > >
> > > > > Given this possibility and my earlier arguments, I recommend that the authors reconsider which results and figures warrant emphasis in their introduction. Regarding generalization, their current explanations remain unsatisfactory. A critical unanswered question persists: why can certain models effectively utilize basis functions while others cannot? This verification is crucial - without resolving this fundamental issue, how can we possibly establish a unified theoretical framework to study Adam, which is robust to different choices of models?

---

> ### Author Response · Authors · 2025-08-07
>
> Thank you again for staying engaged in the discussion.
>
> We respect the referee's disagreement and will continue the discussion by providing additional evidence, including the one requested by the referee, below. But before that, we would like to temporarily set aside our disagreement on the takeaways of our training curves, and make an argument at a higher level.
>
> The core goal of our research is to enable theoretical frameworks that are more relevant to practice, and that will be suitable to guide algorithmic design and understanding of real-world behaviour. **To this end, we argue that the criterion for determining whether the degradation of Adam performance by random rotations is significant should be the criterion of practitioners**. Regardless of what theory considers significant or not, **the differences in the training curves that we reported is considered significant by the practitioners community**, and is very impactful in real-world scenarios. We provided numerous evidence of that, including the fact that in a near-identical setting, the difference that prior works reported betweem Muon and Adam is very similar to the one we report between Adam and full-rot-adam -- and the significance of Muon advantages over Adam in practice is now clearly established. The Lion optimizer has become a popular choice and was published based on much smaller reported difference of performance compared to our curves, see e.g., Figure 11 of [2] for transformer training on text, which are eerily similar to ours, except the gap we observe is larger. Our work lies at the intersection of theory and practice: our goal is to build empirically relevant theoretical frameworks, and this naturally requires to acknowledge what practice considers relevant. How can we hope to build theoretical frameworks that are useful, if we choose in the process to ignore real-world impact in favor of what theory considers important?
>
> To be clear, we have absolute confidence that the claim that the degradation we observe from random rotations in transformer training is **empirically** significant is not controversial, and is supported by extensive prior work and evidence. Besides all the prior work we referred to that observed similar training patterns, and claimed significant differences even for smaller gaps, as additional anecdotical evidence of the community's opinion on the matter, the concurrent work of [Xie et al., 2025] which made a similar claim based on similar experiments, received an spotlight paper award at ICLR this year.
>
> With that said, we now address directly the referee's argument and provide requested information:
>
> >  Could you please also report the offset of these curves at 20 epochs, 40 epochs, etc.
>
> Absolutely, here is the gap between standard Adam and fully rotated Adam, at various number of steps (we remind the referee that there are no epochs here, we do not consume all of the tokens in OpenWebText, the x-axis is in thousands of iterations):
>
> 40k: 0.03594
>
> 60k: 0.02980
>
> 80k: 0.01844
>
> 100k: 0.01534
>
> As the referee can see, the curves are not quite parallel, the gap between them slowly decreases over time. This is expected in optimization: consider any two optimizers, no matter how different their rate of convergence are, if they both converge to a minima of same loss, the difference between their loss will necessarily go to zero asymptotically, and it does not contradict the difference in their performance: otherwise, we would only be interested in whether an optimizer converge, and the rate would be irrelevant.
>
> The impact of the lr decay is that it hides that the actual learning has mostly flattened, by making constant progress through noise reduction at the end of training. It makes the curves look like the model is still making steady progress and was interrupted early, but if we were training for twice the number of steps (and updating the lr schedule accordingly), we would get very small improvement of the final loss. The large 'slope' at the end is driven by lr decay. For a very solid evidence of this claim, please see e.g., Figure 6 (left) of [1]. Our cosine schedule corresponds to the red curve, which seems to be making consistent progress and has a fairly straight slope. However, the blue curve, which uses a constant lr for most of training, and concentrates the lr decay (and corresponding benefit of noise reduction) at the end of training, reveals what happens when we separate the decay artifact: the curve is progressively flattening and converging as we would expect.
>
> [1] Scaling Laws and Compute-Optimal Training Beyond Fixed Training Durations, A. Hagele et al., 2024
>
> [2] Symbolic Discovery of Optimization Algorithms, X. Chen, 2023

---

> > ### Author Response · Authors · 2025-08-07
> >
> > >  if the learning rate's decay toward zero were truly responsible for these patterns (as implied by the authors' cosine LR formula), we should expect the curves' slopes to approach zero
> >
> > > If your cherry-picked figures look unconvincing to me, how should I expect the precision and validation of your work?
> >
> > Please similarly refer to Figure 6 (left) of [1], or to the training curves of almost any recent work on real-world transformer pretraining for Language Modeling using cosine LR decay that provides training curves. For instance the provided training curve of nanoGPT, which has become the gold standard for research on reasonably sized transformers is nearly perfectly straight from 100k iterations to its maximum of 400k iterations. **We understand that this behaviour seems counter-intuitive, but the properties of our training curves that the referee is worried about are overwhelmingly standard in prior works for this type of setting.**
> >
> > **We kindly ask the reviewer to refer to the extensive literature on the subject, rather than what their intuition suggests our training is supposed to look like.**
> >
> > >  the possibility that there might exist multiple local minima, and these minima themselves have different values
> >
> > The reviewer is correct, in fact, even if we trained full-rot-Adam for much longer, it would realistically never reach the final loss of Adam, showing it indeeds does not converge to the same minimum. Because of the effect of noise, an optimizer that learns better from the signal will typically be able to reach a better final loss. But we find the reviewer's conclusion that in this case, it might be meaningless to compare the final loss surprising. Isn't converging to a better minima even more important than the rate of convergence? Sure, it cannot be captured purely by the rate, but here we are interested in which properties are actually aligned with and relevant to practice. **What do we care about when we actually train a neural network, if not the final loss?**.
> >
> > > Please note that your statement of the purpose of this work is to "build theoretical frameworks".
> >
> > And more precisely, to build theoretical frameworks that are relevant to practice. For this purpose, we study how different quantities supported by different theories relate to Adam's performance across bases, which allows to find which one is the driving factor.
> >
> > > A critical unanswered question persists: why can certain models effectively utilize basis functions while others cannot? This verification is crucial - without resolving this fundamental issue, how can we possibly establish a unified theoretical framework to study Adam, which is robust to different choices of models?
> >
> > Because certain models have loss landscapes that possess the properties that we confirmed Adam is utilizing, while others do not. The models for which Adam does not depend on the basis are **precisely** the models for which Adam is not superior to SGD. Imagine SGD converges on a function at rate p, and Adam converges at rate c*p, where c<=1 is a constant that depends on the model and the basis, with c=1 with large probability when sampling a full-scope random rotation of the canonical basis. If c=1 for all bases (as a property of the model), then Adam does not depend on the basis, and does not beat SGD. This is what we observe for ResNets. But if c<1 for the canonical basis, then Adam beats SGD, and its performance will align with the value of c in the basis that is used. Our finding show that c is plausibly determined by the orthogonality of Adam's update in the chosen basis. This **is** an unified theory, that applies to all models. We do not understand the concern of the referee in that aspect.
> >
> > [1] Scaling Laws and Compute-Optimal Training Beyond Fixed Training Durations, A. Hagele et al., 2024

---

> > > ### Comment · Reviewer_Jnyt · 2025-08-08
> > >
> > > Thanks for your follow-up discussions.
> > >
> > > 1. Regarding your reported training loss, is it indeed pointed out that the distinctions between the training loss are even gradually decreasing? In fact, directly comparing the offset is not rigorous; I propose such an approach just because it is comparable to your demos provided and your statement regarding the improvement in the training loss. As I stated, at least in your toy example using cosine lr, this offset is strictly increasing. In addition, your claim '...if they both converge to a minima of the same loss,' does not apply here.  You have acknowledged that in your settings, they indeed do not converge to the same minimum.
> > >
> > > 2. I can not follow your paragraph that "The impact of the lr decay is that it hides that the actual learning has mostly flattened ..." Why decayed learning rate hide the actual learning? Could you please define what is actual learning? An you said "...making steady progress and was interrupted early", I wonder why decay learning rate renders steady progress, and what the "interrupted early" indicates? Is it indicating the elbow point appearing near the 20 epochs in your Figure 1(b)? If so, my question arises: given your cosine lr decay formula, it is evident that the lr is comparable to its initialization through a relatively long phase, eg, 0.5 v.s. 1, and it only dramatically decreases to 0 at the very end stage. At least to me, I do not follow why an lr = 0.5 will 'hide' while lr=1 will not (in fact,  I don't know what 'hide' or 'interrupt' indicates, I just borrow your words).
> > >
> > > In addition, even if we put aside the disagreement on convergence analysis, and only focus on the offset between the final training loss. Why is a difference of a similar scale claimed to have no effect in ResNet?
> > >
> > > 3. I indeed checked the Figure 6 of [1]. **It is clear that at about 18000 steps, the slope gradually decreases, and it almost aligns with the horizon during the last 1000 steps.** I do not feel this result is 'counter-intuitive', and I do not feel this result can explain the *strictly positive slope* in Figure 1. It's not hard to prove that under the settings $\beta_2\geq \beta_1$, $\\|\frac{m_t}{v_t}\\|\leq c$ by Cauchy inequality, where $c$ is an absolute constant solely depending on $\beta_1, \beta_2$. Given such a result, why can loss have a steady, strictly positive decrease when lr approaches 0? All seemingly counterintuitive claims have a logical basis, and the authors are tasked with convincing readers of their truth.
> > >
> > > 4. Regarding this point, I'm very interested in your current example, and would like to discuss more. I wonder how we can incorporate your design of constant $c$ into the current theoretical analysis. Your example that SGD possesses a convergence rate $p$ and that of Adam is $c*p$ indeed provides some intuitions. However, real analysis obviously can not rely on a supposed convergence rate. Then, for example, I have an objective function $f(x)$, and I want to minimize this objective function, by SGD and Adam respectively. How could I define or design a property or whatever, to reflect their capacities to induce the advantages of basis functions according to the property or definition of this objective function? In addition, your current analysis still remains at the stage of 'whether or not', but if we indeed introduce such a design, then it seems that we have to consider what scale correctly reflects the capacities of utilizing the basis function. In addition, it's natural to include an assumption of $\ell_\infty$ norm, but what about such a design? I understand I might require too much, but these are indeed what I focus on.
> > >
> > > I have to clarify the reason that I'm not satisfied with the previous reply.  It is obvious that the model's capacity to induce the basis functions is determined by its inherent properties (loss landscape you mentioned), instead of its comparison with SGD.
> > > If I am studying Adam, why should I necessarily care about how SGD performs? Meanwhile, I want to mention your comments regarding Chen et al. For a universal theoretical framework, at least it should be able to cover linear cases.
> > >
> > > Lastly, I would like to emphasize and clarify that my evaluation is grounded in careful study of both the manuscript and the authors' responses, drawing upon my established expertise. My comments and suggestions focus specifically on substantive concerns rather than general observations applicable to any paper.  I have noticed that authors repeatedly emphasize the practical importance of their work and the inapplicability of theoretical experience in this regard. I understand people may have diverging and biased perspectives due to their distinct backgrounds. Should the authors have any concerns about the review process itself or my qualification to review this paper, they are welcome to bring these to the Area Chair's attention.

---

> > > > ### Author Response · Authors · 2025-08-08
> > > >
> > > > Thank you for your reply. We believe to have found a core cause for our disagreement: we report a running average of the loss, instead of the per-step loss which is very noisy, a standard practice in these settings. **We now realize that this smoothing, although standard, might be the cause for the referee's perplexity regarding our curves.** We will mention it in the figure's caption. See at the bottom for how it explains the difference with [1] and why our curve are not completely flat at the end.
> > > >
> > > > We agree directly comparing the offset provides limited insight. This is exacerbated by the fact that the scale of the LM loss is quite hard to interpret. It is often represented in log scale to account for the fact that a same delta of improvement becomes increasingly harder as the loss goes down, but even log scale has limitations.
> > > >
> > > > > Why decayed learning rate hide the actual learning? Could you please define what is actual learning?
> > > >
> > > > Thank you for bringing up this important distinction. We acknowledge these concepts are not rigorous. Intuitively, there are two contributions to the loss. One is what we vaguely called the 'actual learning', and encompasses knowledge, reasoning, etc. The other one is the contribution of the noise. For instance, due to the high entropy in the data, with a high lr, the model will oscillate in its logits for popular synonyms. This will hurt the loss, but not the 'underlying learning'. Although it's merely a vague intuition, there are some evidence supporting this, for instance the WSD training curve from [1] which the referee already checked: they show an almost identical final loss when the annealing is concentrated at the end of training. This suggests that the benefits from noise reduction when we anneal is *additive* with the improvement that we observe without lr decay. When we spread annealing over training with a cosine schedule, the contribution of the annealing is thus spread over training "additively". Since the 'actual learning', i.e., the improvement we see without lr decay, flattens towards the end, the contribution of lr decay ends up dominating the slope and hiding the slope induced by 'actual learning'. We agree these concepts are not rigorous though, they just give an intuition for a phenomenon that is always present in LM training. We find that [1] is very useful for visual understanding of the interplay between lr decay and learning.
> > > >
> > > > >"making steady progress and was interrupted early", I wonder why decay learning rate renders steady progress, and what the "interrupted early" indicates
> > > >
> > > > > Is it indicating the elbow point appearing near the 20 epochs in your Figure 1(b)?
> > > >
> > > > To be clear, we said it **looks** like the training was interrupted early, but it is not the case. Annealing the lr gives immediate benefit from noise reduction (again this is very visible in [1]). It looks like training was interrupted early in the sense that the curve is still going down, but this is an illusion: as the lr has been fully decayed, no further progress is possible from noise reduction. If we trained for twice the number of steps, the loss would barely improve. This has been observed in many works e.g. [2] Figure 10. So by that we refer to the fact that the curve isn't completely flat at the end, but the model has indeed converged.
> > > >
> > > > > I indeed checked the Figure 6 of [1]. It is clear that at about 18000 steps, the slope gradually decreases, and it almost aligns with the horizon during the last 1000 steps.
> > > >
> > > > This is true and we have a very clear explanation for this. First, when mentioning figure 6 of [1], we refer to the last version on ArXiv, 2405.18392 v3. The referee mentioned 18,000 steps but the figure trains for 200,000 steps, we assume it is a typo and the referee meant 180,000 steps but we want to ensure we are looking at the same figure.
> > > >
> > > > The key is that the authors of [1] are using the same smoothing factor as us to report the loss, but are training for twice longer! Because the loss is noisy, to make figures interpretable it is standard to report a running average of the loss instead of per-minibatch. **This smoothing is the reason our curve is not completely flat at the end**. It does slightly flatten, at the end of training we observe the following losses for AdamW:
> > > >
> > > > 92k: 2.90504
> > > >
> > > > 94k: 2.90212 (-0.00292)
> > > >
> > > > 96k: 2.90025 (-0.00187)
> > > >
> > > > 98k: 2.89866 (-0.00159)
> > > >
> > > > 100k: 2.89759 (-0.00107)
> > > >
> > > > The only reason it is slightly less visible in our work that in [1], is that [1] trained for twice as long (200k steps) but used the same smoothing factor as us to report the loss, meaning at the end of the training, there are twice more steps for the running average of the loss to stabilize as the lr becomes very small.
> > > >
> > > > [1] Scaling Laws and Compute-Optimal Training Beyond Fixed Training Durations, A. Hagele et al., 2024
> > > > [2] Sophia: A Scalable Stochastic Second-order Optimizer for Language Model Pre-training, H. Liu et al., 2023

---

> > > > > ### Author Response · Authors · 2025-08-08
> > > > >
> > > > > We have not saved the per-iteration losses, but we can recover them from the running average knowing the smoothing factor. If the referee wishes, we can thus compute the curves with a reduced smoothing factor, and we will obtain the same level of flattening at the end as in [1], though unfortunately we are not able to share figures this year.
> > > > >
> > > > > And thus the answer to
> > > > >
> > > > > > why can loss have a steady, strictly positive decrease when lr approaches 0?
> > > > >
> > > > > is that the small decreasing slope at the very end of training is purely driven by the running average.
> > > > >
> > > > > > How could I define or design a property or whatever, to reflect their capacities to induce the advantages of basis functions according to the property or definition of this objective function?
> > > > >
> > > > > This would be the key next step and a crucial question. The problem with update orthogonality is that it is tied to the optimizer itself, not just the objective function. What properties of the basis makes Adam's update more (or less) orthogonal? For SVD-rotation, it is easy to see, approximating Adam's behavior with sign descent indeed leads to orthogonal update. But what we're really interested in is *what makes Adam's updates more orthogonal on the canonical basis than other bases?*
> > > > >
> > > > > Our work claims that Adam's performance advantage is due to having more orthogonal updates in the canonical basis. As the referee points out, we now need to understand and quantify the geometrical properties of the loss landscape that lead to this increased orthogonality in order to advance the theory. It should be noted though that the motivation of this future research direction is heavily impacted by Muon. Since we can efficiently and explicitly orthogonalize the update, and it improves performance, we could view Adam as an inferior approximation of Muon. While making this connexion itself is important, going forward, there is no practical reason to improve the approximation versus computing explicit orthogonalization. We think it is still valuable to improve our understanding of Adam at increased depth, but in term of practical motivation, our work points toward switching consideration towards explicit orthogonalization.
> > > > >
> > > > > > If I am studying Adam, why should I necessarily care about how SGD performs?
> > > > >
> > > > > We think a key goal of theoreticians studying Adam is to understand when and why Adam outperforms SGD, hence the comparison.
> > > > >
> > > > > >  For a universal theoretical framework, at least it should be able to cover linear cases.
> > > > >
> > > > > We now understand the referee's point. We thought the referee meant different neural network architectures. Linear cases are a very different class of objective functions compared to real-world deep learning, and might require different theoretical tools. You would not use Adam to train a linear model. Our focus is very much on understanding Adam's behaviour and performance in training real-world neural networks. We believe the dynamics are too different with linear models to truly provide us applicable insights.
> > > > >
> > > > > > Should the authors have any concerns about the review process itself or my qualification to review this paper, they are welcome to bring these to the Area Chair's attention.
> > > > >
> > > > > We are not concerned about the referee's qualification to review this paper. We believe our paper lies at the intersection of theoretical and applied communities that unfortunately rarely overlap in terms of research. This makes it difficult regardless of the referee's background: applied researchers often question the motivation of building theoretical tools if they have no immediate algorithmic application, and theoreticians may not be very familiar with the interpretation of experiments results in realistic settings. We refer to prior literature to try to convince the referee that our claims are well established within a relevant field they may be less familiar with. Thanksfully, the point of the discussion period is precisely to address these type of issues, and **regardless of their assessment, as the discussion period nears its end, we are sincerely thankful that the referee willingly engaged in discussion and gave us the opportunity to defend our work.**

---

### Official Review · Reviewer_wWFQ · 2025-07-02

**Clarity:** 2
**Significance:** 3
**Originality:** 3
**Rating:** 4
**Confidence:** 4

**Summary:**

This paper shows that Adam’s well-known advantage over SGD is not a universal property of the optimizer itself but strongly depends on the coordinate basis in which training is performed. Through extensive experiments on GPT-2, ViT and ResNet-50, the authors demonstrate that random rotations of the parameter space can severely slow Adam, whereas SGD remains unaffected due to its rotation-equivariance. Conversely, a structured rotation obtained by computing a full-rank SVD of each layer’s gradient and running Adam in the SVD basis accelerates convergence beyond the standard baseline. Classic rotation-invariant theoretical assumptions and three popular rotation-dependent ones fail to predict these behaviours. The study identifies the (scaled) orthogonality of the layer updates—quantified by the coefficient of variation of their singular values—as a reliable indicator of Adam’s efficiency across bases, linking the findings to recent orthogonal-update optimizers such as Muon and SOAP.

**Questions:**

1.	Given that the success of optimizers like Muon and SOAP already highlights the importance of update orthogonality, can the authors provide a more intuitive explanation for why this property is so beneficial? And a more intuitive explanation about what is the underlying mechanism that allows SVD rotation to enhance Adam's performance?
2.	I found the discussion about SVD Rotations and Muon is interesting, could the authors further add Muon to the performance comparison in Figure 5 to show how it performs against SVD-rotated Adam?

**Ethical Concerns:**

["NO or VERY MINOR ethics concerns only"]

**Final Justification:**

My questions about the emperiments on Hessian rows was addressed. I suggest the authors add some of the explanations to the intuition and figure 6 in the next version. However, considering the paper can only give limited intuition and no experimental comparisons to Muon, I still think its technical contribution is somehow limited to the community. I will keep my score 4 unchanged.

**Limitations:**

Yes

**Quality:**

3

**Strengths And Weaknesses:**

Strength：
1.	This paper is well written and provides clear empirical demonstrations of basis sensitivity.
2.	It provides a systematic evaluation of existing assumptions, shows why three popular rotation-dependent hypotheses are inadequate.
3.	The main idea of this paper can explain the benefit of recent good optimizers (Muon, SOAP)

Weakness：
This is an overall interesting paper to deliver a good point with detailed experiments. However, the paper's conclusion lacks significant impact. Despite conducting numerous experiments, their connection to the main claims can be weak(refer to Q1). For example, the experiment on Hessian rows doesn't directly explain why the SVD-rotated space performs better. Additionally, the interesting finding in Figure 6 regarding the second-moment distributions is presented without sufficient analysis or discussion or conclusion.

---

> ### Author Rebuttal · Authors · 2025-07-31
>
> We thank the reviewer for recognizing our contributions and relevance to recent new optimizers, and we address their concerns below.
>
> ---
>
> ### **W: Significance of contribution:**
>
> Regarding the significance of impact, we claim the success of Muon and SOAP justifies more work studying how the choice of basis affects the performance of various optimization algorithms. Many optimizers are proposed and justified theoretically (for example, see Table 2 of [1]), but are rarely used in practice. We believe this is partially because the typical assumptions that are used in theory are not predictive of what makes a good deep learning optimizer. We find that the basis does matter, and therefore, many works basing their derivations on rotationally invariant assumptions are unlikely to predict good performance. In a more practical sense, recent works show that choosing a good basis can enhance performance in practice. SOAP [2], for example, “is equivalent to running Adam in a rotated space” as per the authors. We additionally note that the basis SOAP uses is different from that of the SVD rotation, while both improve performance, meaning there is likely room for further research into other bases that could lead to even superior performance.
>
> > The experiment on Hessian rows doesn't directly explain why the SVD-rotated space performs better.
>
> The reviewer is correct that the Hessian row experiments do not explain the improved performance of SVD rotation. Our purpose with the experiment on Hessian rows is to 1) verify the block-diagonal dominance of Hessian, a rotation-dependent characteristic that is mentioned in existing literature; and 2) identify whether simplifying the Hessian to a strict diagonal is a sufficient representation. Our finding shows that even though the Hessian is block-diagonal dominant, the off-block diagonal elements still play an important role and should not be ignored, hence arguing this assumption is inadequate in explaining Adam’s behaviour under rotation, including under SVD rotation.
>
> >Figure 6 regarding the second-moment distributions is presented without sufficient analysis or discussion, or conclusion
>
> Figure 6 acts as a complementary observation that global rotations concentrate Adam’s second moments, leading to less variation in effective learning rates across dimensions. This loss of adaptivity directly accounts for the performance drop observed with global rotations.  While we cannot directly interpret why the second moment values are much more widely distributed under the SVD rotation, we can see that it leads to a much more varied effective step size across dimensions. We suspect this basis is allowing the algorithm to be “more adaptive,” but leave the underlying mechanism for future work.
>
> ### **Q1: Intuition on SVD rotation enhances performance, and update orthogonality is potentially the correct metric:**
>
> A full explanation for SVD rotations enhancing performance is an active area of research, but we appeal to SVD Rotated Adam’s connections to Muon as discussed in Appendix E. Given this connection, an intuition for explaining Muon’s effectiveness should extend here. For a summary of the importance of scaled orthogonality in theoretical frameworks, which we will add to the appendix and refer to in the main paper, we refer to our reply to [reviewer z9Dc Q2](https://openreview.net/forum?id=KD4wgunbhO&noteId=24NXz18oZD), who raised a similar concern. For a detailed overview of the theoretical justifications of Muon, we refer to Jeremy Bernstein’s blog post “Deriving the Muon optimizer”. To the best of our knowledge, there is no published paper explicitly on this topic, but we will cite the blog post accordingly.
>
> ### **Q2: Add Muon to the performance comparison:**
>
> We are also interested in the performance of Muon as compared to the various Adam rotations, but our initial runs have shown subpar performance relative to Adam. We suspect a more thorough hyperparameter search is necessary. Hyperparameter choices often used in GPT speedrunning ([3]) are optimized for a slightly different model (modded-nanogpt) and dataset (fineweb10B), in contrast to our use of NanoGPT and Openwebtext. We are currently conducting this hyperparameter search and hope to be able to report results during the discussion period.
>
> [1] Robin M. Schmidt, Frank Schneider, and Philipp Hennig (2021). “Descending through a Crowded Valley - Benchmarking Deep Learning Optimizers”. In: International Conference on Machine Learning (ICML). Vol. 139, pp. 9367–9376.\
> [2] Nikhil Vyas, Depen Morwani, Rosie Zhao, Itai Shapira, David Brandfonbrener, Lucas Janson, and Sham M. Kakade. 2024. Soap: Improving and stabilizing shampoo using adam. ArXiv, abs/2409.11321.\
> [3] https://github.com/KellerJordan/modded-nanogpt/tree/master

---

### Official Review · Reviewer_Giz7 · 2025-07-03

**Clarity:** 3
**Significance:** 2
**Originality:** 2
**Rating:** 4
**Confidence:** 4

**Summary:**

This paper investigates Adam's sensitivity to different rotations of the parameter space. First, by studying the effect of different types of rotations (global, layer-wise, output-wise, input-wise) on the performance of Adam and SGD in three settings (training a GPT-2 model on the OpenWebText dataset, a ViT/S on ImageNet-1K and a ResNet50 on ImageNet-1K), it shows that broader rotation scope leads to more degradation in Adam’s performance. Output-wise rotations don’t show significant degradation, and a structured SVD-based rotation, inspired by Zhao et al. (2024) improves performance. Next, the paper studies common assumptions used in the theoretical analysis of Adam in prior work, such as $\ell_\infty$-bounded gradients, Hessian block-diagonality and $\ell_\infty$ smoothness, and shows that they are not always correlated with Adam’s performance. Based on these observations, it then proposes a coefficient of variation in the singular values of the gradient metric which measures its scaled semi-orthogonality and correlates with Adam’s performance.

**Questions:**

Please see above.

**Ethical Concerns:**

["NO or VERY MINOR ethics concerns only"]

**Final Justification:**

After going through the discussions and revisiting parts of the paper, some of my concerns were addressed and I lean toward acceptance now. I appreciate the paper's contribution (hence, learning toward accept). However, as mentioned in the comment to authors, I am unclear on the ultimate utility of this result for either theory-building or optimizer design (hence, not strongly supporting accept).

**Limitations:**

yes

**Quality:**

2

**Strengths And Weaknesses:**

In terms of strengths, the paper studies an important direction of understanding why Adam is effective via a novel perspective of looking at the effect of different types of rotation on its performance. It is well-written and easy to follow, and offers interesting insights about insufficiency of existing assumptions such as Hessian block-diagonality in analyzing Adam and proposes a new metric that measures scaled semi-orthogonality of the gradient.

On the other hand, the paper has two main weaknesses:

- The experiments are conducted with relatively smaller model sizes. Given that one of the paper's contributions is a presenting a comprehensive empirical analysis on the effect of different rotations on Adam’s performance, it’s important that the experimental section is thorough. In the three settings considered in the paper, the hyperparameters are also kept fixed and the results don’t show any error bars. This should be addressed to make the results and conclusions more convincing.
- In the current state, the work presents justification for optimizers like Muon, showing that more orthogonalized updates can lead to faster convergence. The proposed metric and its correlation with the performance looks like another way to quantify the observation that SVD-based rotations don’t cause degradations in Adam’s performance. While it can be useful for theoretical analyses for future work, it is unclear how it contributes new insights into motivating the design of other optimization algorithms. Based on the development of SOAP and Muon, it is evident that some type of orthogonalization helps speed-up convergence. So, analyzing this for Adam doesn’t seem to give new insights other than showcasing the difference and quantifying it.

---

> ### Author Rebuttal · Authors · 2025-07-31
>
> We thank the reviewer for recognizing our novelty, and we address their concerns below.
>
> ---
> ### **W1.1 Scale of experiments**
>
> While we agree that the models we train do not reach the scale used in industry, this work was conducted within the computational constraints of an academic institution. However, we argue that the scale of the models we train is sufficiently representative. Notably, the GPT2-Small (124M parameters) style model we use (NanoGPT) has been the basis for several previous and concurrent works throughout the literature on explaining the performance of Adam and related optimization methods. In particular, [1] and [2] use similar GPT2-style models while using a smaller dataset (Wikitext-103), and [3] uses the slightly smaller GPT-Neo variant. The same model is also used in [4], where the authors additionally conduct a wide array of experiments on other models that show similar results to their GPT2-Small model. The same is true of [5], as mentioned by Reviewers z9Dc and Jnyt. Given this trend of prior and concurrent work, we feel that the largest model we train is sufficient, but if the reviewer is interested in seeing any particular experimental setting, we would be happy to try to accommodate.
>
> [1] Jingzhao Zhang, Tianxing He, Suvrit Sra, and Ali Jadbabaie (2020a). “Why Gradient Clipping Accelerates Training: A Theoretical Justification for Adaptivity”. In: International Conference on Learning Representations (ICLR).\
> [2] Michael Crawshaw, Mingrui Liu, Francesco Orabona, Wei Zhang, and Zhenxun Zhuang (2022). “Robustness to Unbounded Smoothness of Generalized SignSGD”. In: Neural Information Processing Systems (NeurIPS).\
> [3] Yan Pan and Yuanzhi Li (2023). Toward Understanding Why Adam Converges Faster Than SGD for Transformers. NeurIPS 2022 Workshop on Optimization for Machine Learning. arXiv/2306.00204.\
> [4] Frederik Kunstner, Alan Milligan, Robin Yadav, Mark Schmidt, and Alberto Bietti. Heavy-tailed class imbalance and why Adam outperforms gradient descent on language models. Advances in Neural Information Processing Systems, 2024.\
> [5] Shuo Xie, Mohamad Amin Mohamadi, and Zhiyuan Li. Adam exploits $\ell_\infty$-geometry of loss landscape via coordinate-wise adaptivity. ICLR 2025\
>
> ### **W1.2 Hyperparameters:**
>
> Within the rebuttal budget, we conducted a small set of hyperparameter sensitivity tests on the choice of $\beta_2$ as suggested by reviewer c8DY (See [“W1: Hyperparameter robustness”](https://openreview.net/forum?id=KD4wgunbhO&noteId=ZHQ8pWB4Pa) response). We regret not being able to visualize these results in the rebuttal due to the NeurIPS policy of not allowing images and PDF links. Overall, the results are consistent with what we observe in the original experiments. We will add these additional experiments to the next revision.
>
> ### **W.2 Significance of contribution:**
>  >..it is unclear how it contributes new insights into motivating the design of other optimization algorithms.
>
> Theoretical assumptions are often central to the design and analysis of new algorithms in both classical and deep learning optimization. In this work, we argue that the assumptions made in several prominent theoretical arguments on the superior performance of Adam do not correlate with that superior performance. These works often conduct theoretical analysis to justify the effectiveness of a new algorithm. This is the case in roughly half of the works we mention in Table 2, and a much more extensive list of proposed (and largely forgotten) optimization algorithms can be found in Table 2 of [1]. It is often the case that algorithms are developed based on vague intuitions of what may work in a specific setting, but lack performance guarantees and fail to generalize to other settings (for example, consider the 160 optimizers in Table 2 of [1]). Given the current scale and cost of training large models, we argue that rather than continue this “brute force” approach, newer theories and assumptions are needed.
>
> We aim to remind the community that for their theory to predict the effectiveness of a new optimization algorithm, the assumptions they use in their analysis should be able to capture the differences between gradient descent and Adam. Our findings show that many common assumptions do not capture this difference, or do not correlate with downstream performance. Our first contribution to the development of new methods is a reminder of the insufficiency of these common assumptions when conducting analysis and development, and the second is to look for some problem characteristics that correlate with performance. We conducted a preliminary investigation into update orthogonality as a characteristic that we find to correlate. We additionally note that it is still often the case that theoretical analysis leads to new practical optimizers, for example, Muon builds off ideas introduced in previous work, such as [2] and [3].
>
> >Based on the development of SOAP and Muon, it is evident that some type of orthogonalization helps speed up convergence. …  analyzing this for Adam doesn’t seem to give new insights…
>
> While “orthogonalization” has been shown to help in very recent optimizers like Muon, there are many important questions left unresolved. There are limited theoretical explanations for why orthogonalization is beneficial (which is itself an interesting question to theoreticians), how it improves upon Adam, whether it is the best quantity to optimize for, or in which parts of the network it is helpful, and so on. The Muon optimizer specifically suggests still using Adam’s update in embedding, prediction, and layer-norm layers; it is unclear what the correct method is for regrouping and orthogonalizing tensors beyond two dimensions, such as in convolutional networks.
>
> As Adam has been proven over time to be effective and widely used in practice, we contend there is still significant value in the study of its performance and specifically its dependency on the basis, and make the connection to the idea behind new methods like Muon. Our work highlights that the coordinate basis being used (represented through different rotations) impacts how “close” to an orthogonal update. In Appendix E, we additionally note that under simplification, AdamW with the SVD rotation is equivalent to Muon. We argue that insights like this can directly lead to the development and proper usage of new optimization methods. For example, SOAP [4] is equivalent to Adam running on a rotated basis, highlighting the importance of studying the impact of rotations on Adam.
>
> [1] Robin M. Schmidt, Frank Schneider, and Philipp Hennig (2021). “Descending through a Crowded Valley - Benchmarking Deep Learning Optimizers”. In: International Conference on Machine Learning (ICML). Vol. 139, pp. 9367–9376.\
> [2] Jeremy Bernstein and Laker Newhouse. “Old optimizer, new norm: An anthology.” arXiv preprint arXiv:2409.20325 (2024).\
> [3] Tim Large, Yang Liu, Minyoung Huh, Hyojin Bahng, Phillip Isola, and Jeremy Bernstein. Scalable optimization in the modular norm. arXiv:2405.14813, 2024.\
> [4] Nikhil Vyas, Depen Morwani, Rosie Zhao, Itai Shapira, David Brandfonbrener, Lucas Janson, and Sham M. Kakade. 2024. Soap: Improving and stabilizing shampoo using adam. ArXiv, abs/2409.11321.
>
> ---
> We kindly ask the reviewers to consider increasing their scores if they find our responses satisfactory. We remain available and happy to address any further questions during the discussion period.

---

> > ### Comment · Reviewer_Giz7 · 2025-08-08
> >
> > Thank you for your response. After going through the discussions and revisiting parts of the paper, some of my concerns were addressed and I lean toward acceptance now. I do believe the paper would benefit from some revisions clarifying the points the authors noted in the "summary of responses to all reviewers and AC". I summarize my position below.
> >
> > I appreciate the breadth of the empirical sweep, and the effort to stress-test multiple theory-backed metrics across rotations and architectures. Using rotations as a falsification tool, and showing that scaled update orthogonality is the only candidate among those tested that consistently tracks Adam’s performance constitutes an interesting contribution.
> >
> > That said, I am *still unclear on the ultimate utility of this result for either theory-building or optimizer design*. For theory aimed at building intuition, in my opinion, it can be acceptable for a metric to be basis-dependent (as long as that is made explicit), so the fact that other metrics “break” under rotation is less of a disqualifier in that context. Moreover, given that update orthogonality is defined with respect to a specific optimizer’s dynamics rather than as an intrinsic property of the loss landscape that can be computed a priori, the paper lacks some discussion on how it is envisioned to be incorporated into a formal theoretical framework. On the other hand, as mentioned earlier, for theory aimed at guiding optimizer design, with the existence of Muon and SOAP, the design implication seems largely addressed in existing work. This should be addressed by including some discussion.

---

### Official Review · Reviewer_c8DY · 2025-07-03

**Clarity:** 4
**Significance:** 3
**Originality:** 3
**Rating:** 5
**Confidence:** 4

**Summary:**

This paper studies the effect of rotations of the parameter space on the performance of AdamW. They include four types of random rotations: global (of the entire parameter as a vector), layer-wise, output- and input-wise. They demonstrate empirically that all types except output-wise slow down training of GPT-2 on OpenWebText and ViT / ResNet-50 on ImageNet-1K, with global rotation hurting the most. They also propose a type of non-random SVD-based rotation which strongly speeds up the AdamW training (as measured in terms of the speed of the train loss decrease as a function of epoch). In a degenerate case (Adam with no memory or $\epsilon$), this happens to coincide with Muon with no momentum or Shampoo with no momentum.

A possible source of this speed-up is that the layer-wise update is closer to a semi-orthogonal matrix. This is investigated by correlating a "closeness to semi-orthogonality" metric with AdamW's training speed for different rotations: the SVD type makes updates closer to semi-orthogonal, whereas the global rotation makes them the least semi-orthogonal.

In addition, there are different assumptions in the literature used to investigate AdamW's performance or prove convergence, specifically, $\|\cdot\|_\infty$-bounded gradients, small $(1, 1)$-norm of the Hessian, near block-diagonality of the Hessian. These assumptions are investigated in the context of rotations.

**Questions:**

1. Why is "Epoch" replaced with "Iteration (k)" in Fig. 13? (Was it intended?)
2. When comparing, say, $A A^T$ with a rescaled identity matrix (e.g. in Appendix C.4), is $A$ assumed to be wide and full-rank? If so, why is it a reasonable assumption for the matrices whose near-semi-orthogonality is checked?

**Ethical Concerns:**

["NO or VERY MINOR ethics concerns only"]

**Final Justification:**

I think this paper provide a significant empirical contribution connecting to theoretical work. I further commented on the discussion with other reviewers in my response to the authors.

**Limitations:**

Although limitations are discussed and referenced in the relevant section of the checklist, I would suggest adding them in the paper.

**Paper Formatting Concerns:**

I am unconcerned.

**Quality:**

3

**Strengths And Weaknesses:**

## Strengths

* I am not aware of such a fine-grained analysis of the effect of rotations on AdamW's training speed, and rotation-dependent assumptions used in the literature. The study includes rotations of different scopes, and both language and vision tasks are included.
* Existing rotation-dependent assumptions are challenged, and a new metric is introduced which is the variability of the singular values of layer-wise updates (how close these updates are to semi-orthogonal matrices). This new metric appears original and promising.
* The paper is clearly written.

## Weaknesses

* Only the typical hyperparameters of Adam, and one choice of batch size, are included. It would be interesting to see how robust the results are with respect to the $\beta_2$ parameter. I understand that it may be unconventional to change $\epsilon$, but technically, just as an illustration, the results should NOT be robust to $\epsilon$ because changing it can transform Adam into SGD (e.g., https://arxiv.org/pdf/1910.05446). Maybe tuning $\epsilon$ is too much, but looking at a couple of $\beta_2$ values and batch sizes would not be unreasonable. Relatedly, statistical significance of the training speed comparisons is not really clear.
* The performance of Adam is evaluated only in terms of the training speed: how quickly the training loss decreases as a function of the iteration number. The outperformance of Muon is typically reported in terms of time until reaching a certain validation loss [Jordan et al., 2024]
* I think more evidence is needed to claim that better typical performance of SGD compared to Adam when training ResNets is explained by reduced sensitivity to rotations, whereas typical outperformance of Adam when training Transformers is explained by using the features of the standard basis. (At least it would be nice to carefully compare to SGD.)
* The paper would benefit from careful proofreading. For example, I doubt that a matrix is called semi-orthogonal if all singular values are 1 or 0. (E.g., the zero matrix is not usually called semi-orthogonal.) The third picture in Figures 13 and 14 seems exactly the same which is likely a small bug.

---

> ### Author Rebuttal · Authors · 2025-07-31
>
> We thank the reviewer for recognizing our contributions, and we address their concerns below.
>
> ---
>
> ### **W1: Hyperparameter robustness:**
>
> Regarding robustness to the $\beta_2$ hyperparameter, we have conducted an additional sensitivity analysis on the base AdamW run and under global rotation. We fixed all other hyperparameters but $\beta_2$, with the additional runs using $\beta_2=0.9$ and $\beta_2 = 0.99$ to compare to the original $\beta_2= 0.95$.
>
> In the case of AdamW under global rotation, both increasing and decreasing $\beta_2$ lead to inferior performance of the optimizer. This indicates that our original choice of $\beta_2$ was near optimal with all other hyperparameters fixed for the global rotation.
> Specifically, the globally rotated AdamW with $\beta_2=0.95$  completed training with training loss 2.912, whereas with $\beta_2$ being set to $0.9$ and $0.99$, the training completed with loss 2.9165 and 2.9167, respectively.
> For the base AdamW, we found that decreasing $\beta_2$  led to worse performance than the original run, while increasing it led to superior performance. While this does mean that base AdamW could have been tuned more, in all configurations of $\beta_2$, the base AdamW showed superior performance to all configurations of the globally rotated AdamW. From our experimentation, we would expect further hyperparameter tuning to only widen this gap.
> Specifically, our original run with $\beta_2=0.95$ ended with a loss $2.896$, whereas $\beta_2=0.9$ and $\beta_2=0.99$ led to a loss of $2.905$ and $2.888$, respectively.
> We regret not being able to visualize these results in the rebuttal due to Neurips' policy of not allowing images and PDF links, but the ordering of these loss values remains consistent throughout the 100 epochs. We plan to add these visualizations in future revisions.
>
> For the rebuttal period, we prioritize budget for searching over $\beta_2$ as we felt it would be more likely to alter performance. Regarding batch size, we refer the reviewer to other recent works [1] and [2] that primarily study the impact of batch size on the performance gap between Adam and SGD. For our focus on Adam’s performance under different coordinate bases, we do not expect the change of gradient variance to have a strong interaction with the rotations. However, it is possible that a larger batch size beyond a certain threshold would better recover to full batch properties that are basis dependent. We will investigate the impact of batch size in future revisions.
>
> [1] Teodora Srećković, Jonas Geiping, Antonio Orvieto. (2025) Is your batch size the problem? Revisiting the Adam-SGD gap in language modeling. arxiv/2506.12543\
> [2] Martin Marek, Sanae Lotfi, Aditya Somasundaram, Andrew Gordon Wilson, Micah Goldblum. (2025) Small Batch Size Training for Language Models: When Vanilla SGD Works, and Why Gradient Accumulation Is Wasteful. arxiv/2507.07101
>
> ### **W2: Comparison metrics**
>
> Our aim in comparing Adam under various rotations was strictly to understand Adam’s dependence on the basis. While our SVD Rotated Adam algorithm does offer some improvements as a function of iteration number, we are not proposing this as an effective replacement for Adam or a competitor to Muon (although the two are connected, as we note in Appendix E). As we mentioned in our response to the reviewer wWFQ, we are planning to add a direct comparison to Muon for further clarity, but again are focusing on comparing training speed rather than wall clock time. As a quick note on the distinction between training and validation loss, the GPT experiments are conducted by doing a single pass over the training set, so the validation and training loss are equivalent in theory. That said, for completeness, we do observe the same patterns in the validation loss, where the final validation loss of global rotation is $2.929$ and the base run is $2.909$.
>
> ### **W3: Re: ..more evidence is needed to claim that better typical performance of SGD compared to Adam when training ResNets is explained by reduced sensitivity to rotations…**
>
> We agree that more evidence would be required if one were to make the strong claim that the only factor leading to a gap or lack of gap in optimization performance is sensitivity to the basis being used. Instead, we claim that the basis can have a significant impact, as shown in our experimentation. While it is certainly possible and likely that other factors, such as properties of the datasets and architectures, would also impact optimization performance differences, we feel our results showing that the choice of basis can both degrade and improve performance make a meaningful contribution.
>
> ### **Mistakes/edit suggestions:**
>
> We thank the reviewer for their careful read. We will fix the following in the next revision:
> * W4: The zero matrix should not be considered as semi-orthogonal,
> * W4: There is a mistake in Figure 13/14 where we uploaded the wrong figures for ViT validation loss and accuracy. Although Figure 13 (left) is correct, Figure 13 (mid) representing the validation loss is wrong, and Figure 13 (right) is just a duplicate of Figure 14 (right), which is for ResNet50, not ViT. Additionally, the "Epoch" replaced with "Iteration (k)" in Fig. 13 is also part of this mistake. We invite the reviewer to read our response to [Reviewer Jnyt](https://openreview.net/forum?id=KD4wgunbhO&noteId=QTuhcsNZQg) for a more detailed summary correcting this mistake,  and we will certainly replace the incorrect figures in future revisions.
> * Limitation 1: We will also move the limitation discussion to the main text and further discuss the possibility of other factors contributing to optimization performance differences.
>
> ### **Q2: Re: Is A assumed to be wide and full-rank?**
>
> We don’t make any assumptions on the matrix shape or rank, given that the parameter matrices vary in shape and the training process cannot guarantee any rank structure.

---

> ### Comment · Reviewer_c8DY · 2025-08-07
>
> Thank you for these answers. I appreciate the additional hyperparameter sweeps.
>
> I have read the discussion with other reviewers and found it useful. I agree with the authors that the gap in the training loss is significant, and I take their point that for the GPT experiment validation and training loss are equivalent. I understand that the contribution is somewhat incremental compared to Xie et al. [3], although this empirical study is more extensive, and using the criterion the AC gave, I would say it would in fact be interesting to see both papers side by side. I also agree with the authors that comparing training speed is informative and is standard in the literature; in general, I find that the authors defended their paper well in the discussion with Reviewer Jnyt. I would suggest replicating some of the large-scale experiments more times with different seeds, to alleviate the concerns about the difference being an accident.
>
> Taking into account the authors' response to my and other reviews, I will raise my score to 5.

---

> > ### Author Response · Authors · 2025-08-08
> >
> > We want to sincerely thank the referee for the time they spent reviewing our work, their consideration of our rebuttal, and their positive assessment.

---

### Official Review · Reviewer_z9Dc · 2025-07-03

**Clarity:** 2
**Significance:** 2
**Originality:** 2
**Rating:** 4
**Confidence:** 3

**Summary:**

This paper empirically studies how different types of parameter space rotations—such as global, output-wise, and SVD-based rotations—affect the performance of Adam. Motivated by these observations, the authors revisit several rotation-dependent assumptions from prior work and demonstrate their limitations in explaining Adam’s behavior under various rotations. To address this, they propose a new metric: the scaled (semi)orthogonality of the update. They show that this quantity aligns well with Adam’s performance across different rotation settings.

**Questions:**

1. For the SVD-based rotation, it appears that running SGD with momentum in the rotated space is equivalent to the Muon update rule. Is this interpretation correct? Additionally, how does running Adam in the SVD-rotated space compare to Muon in terms of performance?
2. Could the authors provide theoretical motivation or intuition for why scaled orthogonality should be considered a meaningful metric for understanding Adam’s behavior under rotation?

**Ethical Concerns:**

["NO or VERY MINOR ethics concerns only"]

**Final Justification:**

My original concerns were the limited novelty compared to Xie et al. (ICLR 2025) and the lack of theoretical development. After the authors communicated with the PC/SAC, I was informed to treat this work as concurrent to Xie et al., so I do not consider this limitation in my final evaluation. During the rebuttal, the authors provided additional discussion on the connection between Adam on an SVD-rotated subspace and Muon, which offers some theoretical insight but does not introduce new theory. Overall, the empirical observations are well controlled and interesting. However, I agree with Reviewer Giz7 that the ultimate utility of these results for theory-building or optimizer design remains unclear, and I therefore lean toward a borderline acceptance.

**Limitations:**

yes

**Quality:**

3

**Strengths And Weaknesses:**

**Strengths:**
The paper conducts systematic and well-controlled experiments examining how different types of rotations—including but not limited to global rotation—affect Adam’s performance. This broad empirical scope is a valuable contribution.

**Weaknesses**
1. **Limited Novelty:** The core findings largely overlap with prior work, especially Xie et al. (2025), which shows that global rotation degrades Adam’s performance and provides a theoretical explanation via $ \ell_\infty $-geometry and convergence analysis. While the observation that SVD-based rotation improves Adam is interesting, it is unsurprising given that it effectively reproduces Muon’s update rule.
2. **Lack of Theoretical Justification:** The proposed metric, scaled orthogonality, is presented as a better predictor of Adam’s behavior under rotation than existing metrics such as $ \ell_\infty $-smoothness or the $ (1,1) $-norm. However, this claim lacks theoretical support. In contrast to existing metrics, scaled orthogonality is not accompanied by any convergence analysis or formal guarantees on optimization performance.

---
**Reference**

Xie et al., Adam Exploits $ \ell_\infty $-geometry of Loss Landscape via Coordinate-wise Adaptivity, ICLR 2025.

---

> ### Author Rebuttal · Authors · 2025-07-31
>
> We thank the referee for their consideration of our work. We would like to bring up some crucial points in response to the referee’s concerns:
>
> ---
>
> ### **W1: Novelty with respect to [Xie et al., 2025]:**
>
> We have communicated to the PCs evidence showing that our work could be treated as concurrent work to the ICLR 2025 reference (and their previous arXiv versions), and thus our novelty evaluation should not be negatively impacted by their paper. To stay compliant with the anonymity requirements, we leave it to the PCs to continue communicating on the topic with the reviewer and AC.
>
> Besides the above, the referee is concerned that our core findings overlap with [Xie et al., 2025]. On the contrary, we would like to point out that our work directly contradicts the findings from [Xie et al., 2025]. Indeed, their main claim is “the exploitation of nice $L_\infty$-geometry is the key advantage of Adam over SGD”. They support their claim empirically by showing in [Table 1] that Adam’s performance is better when the (1,1) norm is lower. Although we find their work very valuable, we show experimentally that this correlation doesn’t generalize to other bases than the ones they considered, with random rotations by output neurons increasing the (1,1) norm while simultaneously improving Adam’s performance. Our work investigates an alternative quantity to link Adam’s performance to the properties of the basis.
>
> The novelty of our work includes not only the study of Adam’s behavior under SVD-based rotations, but also the study of Adam’s behavior under different scopes of random rotation (layer wise, input wise, output wise), which brings significant differences (described in great details in Figures 18 to 33), and also the evaluation of several existing theoretical quantities under each of these rotations, leading to different conclusions from prior works.
>
> ### **Q1: Link between Adam in SVD-based rotations and Muon:**
>
> The closest existing method to running Adam in SVD-based rotations like we do is GaLore, but without rank compression. Although there are indeed some links between Muon and our approach in that they both tend to orthogonalize the update (similarly to SOAP and Shampoo), there are significant differences:
> Muon orthogonalizes the first moment at each step, while our approach updates SVD every T steps
> Muon uses Newton-Schulz (an approximative method) to compute UV, whereas we use SVD to compute U and V separately so that we can run Adam in the latent space.
> Muon orthogonalizes the first moment, whereas we compute SVD and apply it to the gradients
> Using Adam in the SVD rotated space only orthogonalizes the update if you assume Adam behaves exactly like sign descent
>
> ### **W2/Q2: Theoretical motivation for scaled orthogonality:**
>
> Regarding the theoretical motivation for scaled orthogonality, we agree with the referee that it should be explicitly mentioned in the main paper. We discuss the connection with Muon, but do not explicitly remind or point to Muon’s theoretical justification, which directly justifies scaled orthogonality as a powerful theoretical tool for studying the optimization of neural networks. We will update our submission to explicitly refer to the theoretical origins of Muon (and therefore the justification for scaled orthogonality), and remind them in an appendix section. For a detailed overview of the theoretical justifications of Muon, we refer to Jeremy Bernstein’s blog post “Deriving the Muon optimizer”. To the best of our knowledge, there is no published paper explicitly on this topic, but we will cite the blog post accordingly. Here is a quick summary of the justifications:
>
> * Due to the sequential nature of operations in Deep Learning, the function value and the gradient with respect to higher layers only depend on the output of layer $L$, not directly on the weights of layer $L$. Thus, when updating the weights $W$ of layer $L$, one should control how much the update $dW$ impacts the possible outputs, rather than the weights themselves. In other words, instead of constraining the Frobenius norm of dW (like SGD does), we should seek to control the $L_2$ norm of the output shift $dy=(W+dW)x - Wx = dWx$. This is achieved by constraining the operator norm of dW instead of its Frobenius norm. It turns out that if the gradient $G$ has SVD decomposition USV, minimizing $dW^\top G$ (i.e., the linear approximation of the loss) under a constraint on the operator norm of $dW$ has a closed form solution: $-\eta UV$, which is the scaled orthogonalization of the gradient. It is the update that maximizes the reduction of the loss (under linear approximation) while constraining the worst-case shift of output.
>
> This gives a theoretical justification to orthogonalizing the gradient like Muon does, and directly relates the scaled orthogonality of the update to optimization performance. Since Adam’s update orthogonality depends on the basis, it could be an explanatory factor of Adam’s variation of performance across bases, which we find to be empirically supported.
>
> ---
>
> In light of the above clarifications and changes, we hope the referee will reconsider their assessment of our work’s significance and novelty.

---

> > ### Comment · Reviewer_z9Dc · 2025-08-03
> >
> > Thank you for the clarifications.
> >
> > I acknowledge that this work could be considered concurrent with [Xie et al., 2025]. That said, if [Xie et al., 2025] is treated as prior work, I find this paper somewhat incremental. The main idea—that Adam’s performance depends on the “rotated” space—is already established, even if this work provides additional empirical evidence by exploring other rotations. Nevertheless, I am open to changing my assessment if this work is deemed concurrent with [Xie et al., 2025], and I leave this judgment to the AC.
> >
> > One follow-up point: the main message of the paper appears to be that “Adam’s performance is intricately tied to the choice of basis, a relationship that existing theoretical frameworks struggle to capture.” I believe the contribution would be stronger if the paper went beyond empirical observations and suggested a theoretical framework that can capture why SVD-based rotations, in particular, are beneficial.

---

> > > ### Author Response · Authors · 2025-08-05
> > >
> > > Thank you for your response. The SAC has been made aware of the concurrency of our work, and we will notify the AC accordingly.
> > >
> > > On the follow up comment, we agree with the reviewer. We mentioned we would add an explicit reference to the theoretical background of Muon and a new section in appendix, "theoretical background for orthogonalized updates" which justifies the superiority of orthogonalized updates using Muon's theoretical insights (see W2/Q2 in our rebuttal for a quick summary). So we have theoretical justifications for the advantages of orthogonalized updates, and empirical observations that Adam under SVD rotations results in updates with higher orthogonality than under the canonical basis, which in turns lead to better performance.
> > >
> > > Perhaps the last missing component here that the reviewer wishes for is to understand *why* Adam under SVD rotations leads to more orthogonal updates than under the canonical basis. There is also a clear theoretical justification for this, that will be mentioned in the main paper and discussed in the new appendix section:
> > >
> > > Given the SVD decomposition of the gradient EMA (first moment) G=USV.t for a given layer, and assume G square for simplicity (same reasoning applies to rectangular matrices), the SVD-rotated first moment will be G' = S (We update SVD rotations every 200 steps, so it is an approximation because the U and V used for rotation might not match the ones of the current gradient, but GaLore remarked that this approximation holds remarkably well as long as SVD is updated every few hundred steps as in our case). Adam's behavior is often compared to a continuous version of sign decent, because its second moment division behaves as a normalization of the scale (but due to the epsilon constant, near-zero values will still result in negligible updates, hence the continuity). Under this second approximation, since S is diagonal, Adam's update in the latent space will normalize diagonal entries and lead to - cst * identity. When rotating the update back in the original space to apply it, we will obtain - cst * U * identity * V.t = - cst * UV.t, ie., exactly the orthogonalized gradient EMA! Because of the two approximations made here, this method is not as efficient as Muon, which *explicitly* approximates the orthogonalized gradient EMA. But it still justifies theoretically why SVD-rotated Adam has more orthogonal updates than regular Adam, which in turn justifies theoretically why it has superior performance.
> > >
> > > We hope this gives a more complete picture of theoretical justifications behind SVD rotations, through the lens of orthogonality of updates. Our work shows that on the contrary of other theoretical tools, update orthogonality *does* align with empirical findings, suggesting it might be the right key to capture the relationship between Adam and the basis, which is crucial to understand Adam's advantages.

---

> > > > ### Comment · Reviewer_z9Dc · 2025-08-06
> > > >
> > > > Thank you for the clarifications. As I understand it, Adam on an SVD-rotated subspace is essentially similar to Muon (ignoring Newton–Schulz iteration and assuming idealized orthogonalization), and the theoretical background you provide is already drawn from Muon’s insights. This explains why Adam under SVD rotations performs well. However, it remains unclear what the new takeaway message of this paper is. If the theoretical foundation is already established by Muon, it is not evident what novel insights or contributions this work adds beyond that. Could you clarify what new message this paper aims to deliver?

---

> > > > > ### Author Response · Authors · 2025-08-07
> > > > >
> > > > > Absolutely!
> > > > >
> > > > > Of course there is the study of the link between Adam's performance and the choice of basis. Besides this, while Muon provides theoretical justifications for using orthonormal updates, this was never linked to a property of the basis for Adam, i.e., to the best of our knowledge, it was never suggested that Adam's advantage are tied to the orthogonality of its updates, nor that this orthogonality depends of the choice of basis, thus underlining the necessity to address these properties in theoretical frameworks studying Adam.
> > > > >
> > > > > **Moreover, one of our essential contribution is to determine, among several candidate quantities that are all supported by theoretical results, which ones actually drive Adam's performance across bases.**
> > > > >
> > > > > All of the quantities we studied had theoretical backings, just like update orthogonality is backed by Muon's theory. [Xie et al., 2025] explicitly suggested that $l_\infty$-smoothness is the factor accounting for Adam's performance across bases, and especially it's superior performance in the canonical basis which has favorable $l_\infty$ constant. They provided convincing theoretical justification in the form of convergence guarantees -- but while plausible, we provide new evidence suggesting that this quantity is probably not the one driving performance.
> > > > >
> > > > > In contrast, we find that only update orthogonality aligns with empirical performance and is suitable to study Adam's behaviour across bases. **We provide a crucial step between theory and practice, which we believe to be essential to building impactful theoretical frameworks: Muon's theory shows that update orthogonality *could* explain Adam's performance. Our work shows that it empirically *does* -- and that other similar theories do not.**

---

> > > > > > ### Comment · Reviewer_z9Dc · 2025-08-07
> > > > > >
> > > > > > Thank you for the detailed response. I now better appreciate the novelty in drawing a connection between Adam’s behavior on SVD-rotated subspaces and Muon optimizer. The clarification regarding the role of update orthogonality—both in theory (via Muon) and in empirical performance—helped contextualize the contribution. Based on the additional insights provided, I believe the paper would be strengthened, and I’m now leaning toward acceptance. I have no further questions.

---

> > > > > > > ### Author Response · Authors · 2025-08-08
> > > > > > >
> > > > > > > We want to express our sincere gratitude to the referee for the time they spent reviewing our work, and for their consideration of our rebuttal.

---

### Author Response · Authors · 2025-08-07
**Summary of responses to all reviewers and AC**

We thank the reviewers for taking the time to prepare their feedback. We summarize the strengths and weaknesses from reviewers and our rebuttals.

---
Reviewer z9Dc, c8DY recognize the empirical scope of our experiments to be comprehensive and valuable;\
Reviewer c8DY, Giz7, Jnyt recognize our novelty and contribution in challenging existing assumptions and identifying a new promising metric;\
Reviewer c8DY, Giz7, wWFQ finds our work well written.

We summarize the main contributions of our work:
1. Comprehensive analysis of Adam’s rotation sensitivity in practical neural network training: We found random rotations degrade performance, while SVD rotation improves performance;
2. Assessment of rotation-dependent assumptions in the literature: We find that existing theoretical frameworks fall short in capturing Adam’s beneficial properties and offer guidance on what a suitable metric should reflect;
3. Verifying a better rotation-dependent quantity: we find the (semi-)orthogonality of layer updates to be a strong predictor of Adam's performance.

---

We have attempted to address the reviewer's questions individually. Here, we make a few general comments on the main points that emerged across the reviews.


### Comment 1: Concurrency to [Xie. et. al, 2025]
Reviewer z9Dc raises the concern of our novelty regarding [Xie. et. al, 2025]. We raise two points:
1. We have communicated to the PCs evidence showing that our work could be treated as concurrent work to the ICLR 2025 reference (and their previous arXiv versions), and thus our novelty evaluation should not be negatively impacted by their paper;
2. Besides the concurrency, our work disagrees with [Xie et al., 2025]'s claim. We show experimentally that the correlation between Adam’s performance and the (1,1) norm doesn’t generalize to other bases than the ones they considered.

### Comment 2: New experiments on hyperparameter robustness
Reviewer c8DY suggested it would be interesting to see how robust the results are concerning other hyperparameters such as $\beta_2$.

Within the rebuttal budget, we conducted a small set of hyperparameter sensitivity tests on $\beta_2$. Overall, the results are consistent with what we observe in the original experiments.

### Comment 3: Importance of the contribution
3.1: Reviewer z9Dc, Giz7 questions the importance of our results given the existence of Muon, a new optimizer that employs a similar idea to our SVD-rotation.

We point out that

1. The theoretical understanding of Muon (made public only in December 2024), such as why orthogonalization is beneficial and what convergence guarantee it achieves, is still an ongoing research area. Many other important questions are also left unresolved, such as how it improves upon Adam, whether it is the best quantity to optimize for, or in which parts of the network it is helpful, and so on; This was not linked to Adam's performance before, and most importantly no prior works has established a relationship between the orthogonality of Adam's update, the choice of basis, and Adam's performance;
2. Muon provides support for the advantages of orthogonal updates, and by proxy running Adam under SVD rotations; our work examines several properties similarly supported by theory, including $L_\infty$ smoothness, which [Xie et al., 2025] explicitly claimed was the driving factor behind Adam's performance across bases. All of those theoretical results provide quantities that **plausibly** drive Adam's performance, and our work determines which ones **actually do empirically**. This is a very crucial step between theory and practice to establish which frameworks are not just sound, but also relevant to real-world settings.

3.2: Reviewer Jnyt questions the significance of the convergence rate gap since "training loss appears almost parallel".

We provide evidence and detailed explanations that
1. This is a natural artifact of empirical training of transformer models at this scale, that all reasonable optimizers in realistic settings for LM training will display this behaviour;
2. The gap in our training loss is considered significant at the scale of our experiments.

---
Reference

Xie et al., Adam Exploits $l_\infty$-geometry of Loss Landscape via Coordinate-wise Adaptivity, ICLR 2025

---

### Decision · Program_Chairs · 2025-09-17

**Decision:**

Accept (poster)

**Comment:**

The paper considers the problem of understanding the performance of Adam. Given the popularity of Adam and a lack of rigor around it, the problem is clearly very important and of interest.

The paper shows two results:

R1. Across different architectures, Adam's performance is dependent on the basis.
R2. Across different metrics that have been proposed to understand Adam, orthogonality of updates has the most promise to explain the speedup that Adam provides relative to SGD.

Strengths:

1. The paper does a fine-grained and thorough empirical analysis of Adam.
2. While Adam being basis-dependent is known, the analysis points out various subtleties such as the type of rotation.

I will now go through some of the concerns that were raised:

1. The paper uses the final loss to compare setting, the rate of convergence appears similar for different settings: During discussion phase, reviewers were mostly satisfied with this, since it is standard practice in LLM optimization (even though there is some gap from usual theoretical analysis). Therefore, I will mainly ignore this for decision making.

2. Comparison with prior work: Xie. et al, 2025 was regarded as concurrent work. However, there is still the point about comparison with Muon. Perhaps the most important takeaway of the paper, about the orthogonality of the updates, is already incorporated in Muon, and is the main motivation for Muon.

3. A more rigorous link between rotations and Adam's performance: The claim that "the models for which Adam does not depend on the basis are precisely the models for which Adam is not superior to SGD" is not that well-supported. There seems to be a correlation, but a general claim about a model class (esp. about Adam vs SGD) that does not take other factors such as the properties of the data (such as heavy tailed behavior) into account can potentially be misleading. Therefore result R1 could have been better supported.

Overall, I feel this paper is borderline, though given the current lack of understanding of Adam and the potential to inspire future work, I lean in favor of acceptance.